# Accelerating Adaptive Federated Optimization with Local Gossip Communications

**Yujia Wang**
Pennsylvania State University
`yjw5427@psu.edu`

**Pei Fang**
Tongji University
`greilfang@gmail.com`

**Jinghui Chen**[*]
Pennsylvania State University
`jzc5917@psu.edu`

## Abstract

Recently, adaptive federated optimization methods, such as FedAdam and FedAMS-Grad, have gained increasing attention for their fast convergence and stable performance, especially in training models with heavy-tail stochastic gradient distributions. However, these adaptive federated methods suffer from the *dilemma of local steps*, i.e., the convergence rate gets worse as the number of local steps increases in partial participation settings, making it challenging to further improve the efficiency of adaptive federated optimization. In this paper, we propose a novel method to accelerate adaptive federated optimization with local gossip communications when data is heterogeneous. Particularly, we aim to lower the impact of data dissimilarity by gathering clients into disjoint clusters inside which they are connected with local client-to-client links and are able to conduct local gossip communications. We show that our proposed algorithm achieves a faster convergence rate as the local steps increase thus solving the *dilemma of local steps*. Specifically, our solution improves the convergence rate from $\mathcal{O}(\sqrt{\tau}/\sqrt{TM})$ in FedAMSGrad to $\mathcal{O}(1/\sqrt{T\tau M})$ in partial participation scenarios for nonconvex stochastic setting. Extensive experiments and ablation studies demonstrate the effectiveness and broad applicability of our proposed method.

## 1 Introduction

Federated Learning [22, 29] has become a crucial large-scale machine learning paradigm where multiple clients jointly train a machine learning model coordinated by a central server. Unlike traditional centralized training, where data is stored in a single central server, in federated learning, training data are stored on each client and only the local trained models are iteratively exchanged and synchronized to the central server. FedAvg [29] (also known as Local SGD [33]) has become one of the most popular federated optimization methods, where each client locally performs multiple steps of SGD updates then aggregates together for the global model update. Aside from the advantage of data privacy protection, the design of multiple local update steps also intends to reduce the communication between the server and clients. Compared with distributed learning [29, 33] where each local update step is followed by server aggregation, federated learning can further reduce the communication rounds. Recently, as the booming interests in training large-scale models such as BERT [8], GPT-3 [3] and ViT[9], adaptive federated optimization methods such as FedAdam [30], FedAGM [36] and FedAMS [41] has also been proposed and attracted a lot of attention. Specifically, adaptive federated optimization retains the multiple steps of SGD update on local clients but changes the global update of FedAvg from one-step SGD to one-step adaptive gradient methods update. By introducing adaptivity into federated learning, it achieves fast convergence, especially for heavy-tail stochastic gradient noise distributions.

---

[*]Corresponding author.

Workshop on Federated Learning: Recent Advances and New Challenges, in Conjunction with NeurIPS 2022 (FL-NeurIPS'22). This workshop does not have official proceedings and this paper is non-archival.

While various adaptive federated optimization algorithms have been proposed, there still exist several key bottlenecks in applying adaptive federated optimization in practice, such as (1) *large client-to-server communication overhead* due to the limited bandwidth and repetitive transmission between the server and clients; (2) *intense sensitivity on data heterogeneity* since nonidentical data distribution on different clients introduce extra variance between clients and slow down the training process of federated learning. What's even worse, these two objectives may conflict with each other: while increasing the number of local training steps and using partial participation strategies can certainly save the communication costs between the server and clients, it has been shown that the variance overhead term grows as the number of local steps increases in partial participation settings, which leads to worse convergence rate in adaptive federated optimization [30, 41]. Such worse convergence result is largely due to data heterogeneity, as in the i.i.d setting, the increasing of local steps can indeed lead to a better convergence rate. In this work, we refer this problem as the *dilemma of local steps*. Similar issues have also been shown in FedAvg that a larger number of local SGD steps may cause over-fitting on local clients, also known as client-drift, which slows down the convergence or leads to an unstable result [19]. This motivates us to study the following question:

*Can we resolve the **dilemma of local steps** for adaptive federated optimizations? i.e., achieving a faster convergence rate as the number of local steps increases under the non-i.i.d. setting?*

Note that previous studies have shown that traditional variance reduction techniques [17, 10] can help reduce the client-drift and improve the convergence rate in FedAvg by additionally computing and communicating a control variate or a full-batch gradient [18, 19]. However, it still remains an open problem how to apply such variance reduction techniques to adaptive federated optimization as it requires precise characterization of each local SGD iteration, which is incompatible with adaptive federated optimization, whose current analysis can only give the characterization of cumulative gradient estimators between two communication rounds. Therefore, we take a different route here to solve the *dilemma of local steps* in adaptive federated optimization: since the core idea of variance reduction is to lower the impact of data dissimilarity between clients, we could obtain a similar effect by enabling the local client-to-client communications similar to gossip averaging in decentralized training [2, 26, 25] for reducing the dissimilarity variance between clients. Specifically, in this paper, we propose a novel hybrid adaptive federated optimization method, HA-Fed, which benefits from both adaptive federated optimization [30, 36, 41] and techniques in decentralized training [26, 21, 25]. HA-Fed is structured by partitioning a global network into disjoint network clusters, where clients in the same cluster are connected via locally gossip communication links. These locally communications are fast and frequent, which incurs neglectable extra communication overhead compared with client-to-server communication links.

Our contributions can be summarized as follows:

1. We propose a new hybrid adaptive federated optimization method, HA-Fed, which benefits from the frequently local gossip communications to resolve the *dilemma of local steps* in adaptive federated optimization methods. i.e., achieves faster convergence rate as the local steps increases.

2. We show the theoretical convergence improvements for our proposed HA-Fed in the stochastic nonconvex optimization settings. Specifically, we prove that HA-Fed achieves a faster convergence rate than FedAMSGrad [2] on the non-dominant term in full participation scenarios. Moreover, we show that in the more practical partial participation setting, HA-Fed improves the convergence rate (dominant term) from $\mathcal{O}(\sqrt{\tau}/\sqrt{TM})$ to $\mathcal{O}(1/\sqrt{T\tau M})$ w.r.t. global communication rounds $T$, local update steps $\tau$ and the number of participation clients $M$.

3. Extensive experiments are conducted on several benchmarks dataset and show that our proposed HA-Fed effectively saves the client-to-server communication overhead while achieving faster convergence with heterogeneous data. Extensive ablation studies also show the broad applicability of our proposed method.

**Notation:** We consider column vectors throughout this paper except special explanations. For $\mathbf{x}, \mathbf{y} \in \mathbb{R}^d$, denote $\sqrt{\mathbf{x}}, \mathbf{x}^2, \mathbf{x}/\mathbf{y}$ as the element-wise square root, square, and division of the vectors. For vector $\mathbf{x}$ and matrix $A$, $\| \cdot \|$ abbreviates the $\ell_2$ norm of the vector and Frobenius norm of the

---

[2]The convergence rate of FedAMSGrad is obtained from the convergence analysis for FedAMS [41], where FedAMSGrad gets a similar convergence to FedAMS. FedAMSGrad is also included in [36].

matrix, i.e., $\|\mathbf{x}\| = \|\mathbf{x}\|_2$ and $\|A\| = \|A\|_F$, and $\|A\|_2$ denotes the spectral norm of matrix $A$. We denote $\mathbf{1}$ as vector with all elements equal to 1 with appropriate dimension, and $\mathbf{I}$ as the identity matrix with appropriate dimension.

## 2 Related Work

**Federated learning:** Federated learning [22] has attracted growing interest recently due to the demand for training models locally at edge devices and the requirements of privacy protection. Federated optimization methods such as SGD-based optimization algorithm, FedAvg [29], also known as Local SGD [33], have been widely used in federated learning. Aside from FedAvg, since adaptive gradient methods such as Adam [20] and its variant AMSGrad [31] overcame the sensitivity to parameters and slow to convergence issue of SGD, adaptive federated optimizations such as FedAdam[30], FedAGM [36] and FedAMS [41] studied the corresponding adaptive optimization algorithms in federated learning. Moreover, several works [14, 11, 19, 24, 43] addressed and focused on the data heterogeneity issues of federated learning, where [19] proposed a federated learning variance reduction method that overcomes the data heterogeneity, but it requires extra communication costs for variance reduction operations. [12] considered heterogeneous communications for modern communication networks that improve communication efficiency. Hierarchical federated learning algorithms [27, 1, 4] are developed by aggregating client models to edge servers first before synchronizing them to the central server.

**Decentralized learning and other frameworks:** Decentralized learning is a large-scale machine learning paradigm without a central server. It has been firstly studied from gossip averaging techniques [38, 2]. Decentralized (gossip) SGD algorithms [26, 25, 2, 34] are then proposed that consider client-to-client communications after each step of SGD update on the client. [28] proved a tight lower bound for decentralized training under the nonconvex setting. [35] proposes a leader-distributed SGD algorithm that pulls workers to the currently best-performing model among all models, which also utilizes inexpensive gossip communication. Moreover, recent studies generalized various distributed SGD algorithms under unified frameworks [39, 21], where [39] included reducing communication costs and decentralized training in i.i.d. settings, and [21] studied a general network topology-changing gossip SGD methods that summarize several algorithms in distributed and federated learning.

**Communication-efficient federated learning:** In terms of reducing the communication overhead in federated learning, one of the common approaches is to save the communication bits when synchronizing, such as the compressed and quantized FedAvg-based methods [32, 16, 15, 6]. Note that the bit compression strategy is orthogonal to our hybrid adaptive federated learning framework and can potentially be combined to further reduce communication overheads.

## 3 Preliminaries on Adaptive Federated Optimization

Firstly, let's begin with the general federated learning problem under nonconvex stochastic optimization settings. Suppose we have $N$ local clients, and our goal is to minimize the following objective:

$$\min_{\mathbf{x} \in \mathbb{R}^d} f(\mathbf{x}) := \frac{1}{N} \sum_{i=1}^{N} f_i(\mathbf{x}), \tag{3.1}$$

where $\mathbf{x}$ denotes the model parameters, $d$ denotes the dimension of the model parameters $\mathbf{x}$, $f_i(\mathbf{x}) = \mathbb{E}_{\xi \sim \mathcal{D}_i} f_i(\mathbf{x}, \xi_i)$ is the local nonconvex loss function corresponding to client $i$, and $\mathcal{D}_i$ is the local data distribution associated with client $i$. FedAvg [29] is a popular optimization algorithm to solve Eq. 3.1, with the sequential implementation of local SGD updates and global averaging.

Adaptive federated optimization is then proposed to incorporate adaptivity in federated optimization methods by replacing the global averaging in FedAvg with one-step adaptive gradient optimization. For example, FedAMSGrad is designed with multi-steps of local SGD updates and followed by one step of global AMSGrad [31] update. Specifically, at global round $t$, the server broadcasts the model $\mathbf{x}_t$ to selected clients. Each client $i$ conducts $\tau$ steps of local SGD updates with local learning rate $\eta_l$ and obtains the local model $\mathbf{x}_{t,\tau}^i$. The model difference $\Delta_t^i = \mathbf{x}_{t,\tau}^i - \mathbf{x}_t$ for each client is aggregated to the server and averaged to $\Delta_t$. The server updates the global model $\mathbf{x}_{t+1}$ by taking $\Delta_t$ as a pseudo

gradient for calculating momentum $\mathbf{m}_t$ and variance $\mathbf{v}_t$ for AMSGrad optimizer, and performs one step AMSGrad update with global learning rate $\eta$, i.e.,

$$\mathbf{m}_t = \beta_1 \mathbf{m}_{t-1} + (1 - \beta_1)\Delta_t, \mathbf{v}_t = \beta_2 \mathbf{v}_{t-1} + (1 - \beta_2)\Delta_t^2,$$
$$\widehat{\mathbf{v}}_t = \max\{\widehat{\mathbf{v}}_{t-1}, \mathbf{v}_t\}, \mathbf{x}_{t+1} = \mathbf{x}_t + \eta \frac{\mathbf{m}_t}{\sqrt{\widehat{\mathbf{v}}_t} + \epsilon}, \tag{3.2}$$

the server obtains model $\mathbf{x}_{t+1}$ after one global round. Besides FedAdam and FedAMSGrad, there are several adaptive federated optimization methods with slightly changes in update formulas, e.g., FedAdagrad and FedYogi [30], FedAGM [36] and FedAMS [41].

The convergence of FedAMSGrad is affected by several factors such as the number of local steps $\tau$, global rounds $T$, and the number of participating clients $M$. In full participation settings, where $M$ is equal to the total number of clients $N$, FedAMSGrad enjoys a convergence rate of $\mathcal{O}(1/\sqrt{T\tau N})$. This suggests that even for heterogeneous data, a larger number of local steps $\tau$ can help save the client-to-server communication rounds and lead to faster convergence. However, previous study shows that under more practical partial participation settings, FedAMSGrad only achieves a convergence rate of $\mathcal{O}(\sqrt{\tau}/\sqrt{TM})$ with heterogeneous data. This suggests that while larger $\tau$ can reduce communication frequency, it scarifies the convergence rate and requires more communication rounds to converge. We refer to this problem as the *dilemma of local steps*.

The *dilemma of local steps* arises in partial participation settings since the heterogeneous data induces a large variance term in the final convergence result, which is proportional to the number of local steps $\tau$ and thus leads to a worse convergence rate. For full participation settings, it is fortunate that this variance overhead only appears on the non-dominant term, thus it does not slow down the overall convergence. While for partial participation settings, the larger $\tau$ amplifies the over-fitting issue on local clients as fewer clients participate in each round of global training and becomes a dominant term in the convergence result. Although variance reduction techniques [17, 10] can help reduce the client-drift (or the *dilemma of local steps*) in the local iterations of FedAvg [19, 18], the success of applying variance reduction techniques to FedAvg rely on the precise characteristic of each local SGD iteration. However, as shown in Eq. 3.2, the global adaptive optimizer updates via the cumulative model difference $\Delta_t$ between two communication rounds, which makes how to apply iterative variance reduction bounds to adaptive federated optimization an open problem. In the following, we will present our attempt to resolve the *dilemma of local steps* by a new hybrid adaptive federated optimization method.

## 4   Proposed Method

In this paper, we propose a hybrid adaptive federated optimization method (HA-Fed) where the clients are partitioned into disjoint clusters inside which they can communicate by fast client-to-client links, and clusters communicate with the central server with client-to-server communication links. Specifically, assuming we have one central server and $K$ disjoint clusters, each of which contains $n$ local clients and there are connected by client-to-client links (denoted by the adjacency matrix $W_k$). Let's denote the total number of clients as $N = Kn$. Our goal is to solve the following optimization problem:

$$\min_{x \in \mathbb{R}^d} f(\mathbf{x}) := \frac{1}{N} \sum_{i=1}^{N} f_i(\mathbf{x}) = \frac{1}{K} \sum_{k=1}^{K} \bar{f}_k(\mathbf{x}), \tag{4.1}$$

where $f_i(\mathbf{x}) = \mathbb{E}_{\xi \sim \mathcal{D}_i} f_i(\mathbf{x}, \xi_i)$ is the nonconvex loss function for the $i$-th client, and $\bar{f}_k(\mathbf{x}) := \frac{1}{n} \sum_{i \in \mathcal{V}_k} f_i(\mathbf{x})$ is the average loss on cluster $k$. We consider $\mathcal{V}_k$ as the set of local clients in the cluster $k$, and clients in cluster $k$ are linked by a connected graph $\mathcal{G}_k$[3].

In order to accelerate FedAMSGrad under heterogeneous data settings, our HA-Fed starts from FedAMSGrad and introduces intra-cluster gossip communications. Gossip communication is designed for clients in a network to communicate with their neighbors without a central server, and it has been a popular approach in decentralized learning [26, 21, 7]. Our proposed HA-Fed adds frequent client-to-client gossip communication inside each cluster to leverage the over-fitting issue within the

---

[3]The connected graph implies there is a path from any client to any other client in the graph.

cluster. These gossip communications rely on inexpensive local client-to-client communications without incurring extra client-to-server communication rounds, but at the same time, prevent over-fitting on local clients since the model on each client sufficiently communicates with their neighbors.

Algorithm 1 summarizes the proposed HA-Fed in full participation scenarios. The major difference between HA-Fed and FedAMSGrad lies in the local update step within each cluster (Line 9 in Algorithm 1): at the $s$-th step of intra-cluster training for cluster $k$, after client $i$ finishes their local update and obtains $\mathbf{x}^i_{t,s+\frac{1}{2}}$ by one step SGD, we conduct one gossip averaging step within the cluster, i.e., let each client communicate with its neighbors $\mathcal{N}^i_k$ and aggregate the nearby local models with a weighted matrix $W_k$. The rest part of the algorithm is similar to FedAMSGrad.

In order to further reduce client-to-server communication rounds, we also adopt partial participation setting for HA-Fed[4]. Generally, in partial participation settings, the server samples a subset of $m$ clients in each cluster before each round starts and only broadcasts the current model to these $m$ selected clients and the selected clients will broadcast the received model to other clients within the same cluster with client-to-client links. For global model updates, all selected clients send the model difference $\Delta^i_t$ to the central server, and the server aggregates them to $\Delta_t$. The rest of the partial participation update is the same as the full participation scenarios.

---

**Algorithm 1** HA-Fed:full participation

**Input:** initial point $\mathbf{x}_1$, global step size $\eta$, local step size $\eta_l$, $\beta_1, \beta_2, \epsilon$, weighting matrix $W_k$ for all clusters $k \in [K]$

1: $\mathbf{m}_0 \leftarrow 0, \mathbf{v}_0 \leftarrow 0$
2: **for** $t = 1$ to $T$ **do**
3:     **for** each cluster $k \in [K]$ in parallel **do**
4:         **for** each client $i \in \mathcal{V}_k$ in parallel **do**
5:             Receive model from the server: $\mathbf{x}^i_{t,0} = \mathbf{x}_t$
6:             **for** $s = 0, ..., \tau - 1$ **do**
7:                 Compute local stochastic gradient: $\mathbf{g}^i_{t,s} = \nabla F_i(\mathbf{x}^i_{t,s}; \xi^i_{t,s})$
8:                 Local update: $\mathbf{x}^i_{t,s+\frac{1}{2}} = \mathbf{x}^i_{t,s} - \eta_l \mathbf{g}^i_{t,s}$
9:                 Gossip communication: $\mathbf{x}^i_{t,s+1} = \sum_{j \in \mathcal{N}^i_k} (W_k)_{i,j} \mathbf{x}^j_{t,s+\frac{1}{2}}$
10:             **end for**
11:             Get the model difference: $\Delta^i_t = \mathbf{x}^i_{t,\tau} - \mathbf{x}_t$
12:         **end for**
13:     **end for**
14:     Server gets model difference: $\Delta_t = \frac{1}{K} \sum_{k \in [K]} \frac{1}{n} \sum_{i \in \mathcal{V}_k} \Delta^i_t$
15:     Update: $\mathbf{m}_t = \beta_1 \mathbf{m}_{t-1} + (1 - \beta_1) \Delta_t$
16:     Update: $\mathbf{v}_t = \beta_2 \mathbf{v}_{t-1} + (1 - \beta_2) \Delta_t^2$
17:     $\widehat{\mathbf{v}}_t = \max(\widehat{\mathbf{v}}_{t-1}, \mathbf{v}_t)$ and $\widehat{\mathbf{V}}_t = \mathrm{diag}(\widehat{\mathbf{v}}_t + \epsilon)$
18:     Server updates $\mathbf{x}_{t+1} = \mathbf{x}_t + \eta \frac{\mathbf{m}_t}{\sqrt{\widehat{\mathbf{v}}_t} + \epsilon}$
19: **end for**

---

In a nutshell, HA-Fed takes advantage of decentralized training to resolve the *dilemma of local steps* in adaptive federated optimization while preserving the benefit of adaptive optimizations: The server aggregation rule and update schemes follow standard adaptive federated optimization, which enjoys nice convergence properties, especially for heavy-tail stochastic gradient noise distributions. Meanwhile, the local gossip communications alleviate the impact of data dissimilarity between clients on the final convergence rate. Of course, this design requires all clients within each cluster to stay active and perform gossip communications. Yet we also want to emphasize that HA-Fed can also be compatible with scenarios where not all clients are active at each iteration by simply adapting the frequency of local gossip communications. We refer interested readers to Appendix F.3 for more details.

---

[4]Due to the space limit, see details in Algorithm 2 in the Appendix.

# 5 Convergence Analysis

In this section, we provide the theoretical convergence analysis of the proposed HA-Fed method. Before starting with the main theoretical results, let us first state the following assumptions:

**Assumption 5.1** (Smoothness). Each loss function on the $i$-th client $f_i(\mathbf{x})$ is $L$-smooth, i.e., $\forall \mathbf{x}, \mathbf{y} \in \mathbb{R}^d$, $\left| f_i(\mathbf{x}) - f_i(\mathbf{y}) - \langle \nabla f_i(\mathbf{y}), \mathbf{x} - \mathbf{y} \rangle \right| \leq \frac{L}{2} \|\mathbf{x} - \mathbf{y}\|^2$.

**Assumption 5.2** (Bounded Gradient). Each loss function on the $i$-th client $f_i(\mathbf{x})$ has $G$-bounded stochastic gradient on $\ell_2$, i.e., for all $\xi$, we have $\|\nabla f_i(\mathbf{x}, \xi)\| \leq G$.

**Assumption 5.3** (Bounded Stochastic Variance). Each stochastic gradient on the $i$-th client has a bounded local variance, i.e., for all $\mathbf{x}, i \in [m]$, we have $\mathbb{E}\left[\|\nabla f_i(\mathbf{x}, \xi) - \nabla f_i(\mathbf{x})\|^2\right] \leq \sigma^2$.

Assumption 5.1 also implies the $L$-gradient Lipschitz condition, i.e., $\|\nabla f_i(\mathbf{x}) - \nabla f_i(\mathbf{y})\| \leq L\|\mathbf{x} - \mathbf{y}\|$, it is a standard assumption in nonconvex optimization problems [20, 31, 24, 43]. Assumption 5.2 is usually adopted in studying adaptive gradient methods [20, 31, 45, 5]. Assumption 5.3 is frequently stated in studying distributed and federated learning optimization problems [30, 43, 7, 40].

**Assumption 5.4** (Bounded Inter-Client Variances). The variance between local client's objective function and the objective function on the corresponding cluster is bounded, i.e., for all $\mathbf{x}, k \in [K]$, we have $\frac{1}{n} \sum_{i \in \mathcal{V}_k} \|\nabla f_i(\mathbf{x}) - \nabla \bar{f}_k(\mathbf{x})\|^2 \leq \sigma_k^2$. The objective function on each cluster and the global function has a bounded variance: for $\alpha \geq 1$ and $\sigma_g \geq 0$, there is $\frac{1}{K} \sum_{k \in [K]} \|\nabla \bar{f}_k(\mathbf{x})\|^2 \leq \alpha^2 \|\nabla f(\mathbf{x})\|^2 + \sigma_g^2$.

Assumption 5.4 represents the data heterogeneity in a cluster and between clusters. The similar data heterogeneity assumption, which considers the variance between local clients, is common in federated learning [30, 43] and decentralized learning [26, 25, 21].

**Assumption 5.5** (Gossip Weighting Matrix). The local clients in cluster $k$ are connected in the graph $\mathcal{G}_k$, and the corresponding weighting matrix $W_k$ is a doubly stochastic matrix with the fact: $W_k \in [0,1]^{n \times n}$, $W_k \mathbf{1} = \mathbf{1}$, $\mathbf{1}^\top W_k = \mathbf{1}^\top$ and null$(\mathbf{I} - W_k) = \text{span}(\mathbf{1})$. We further assume the spectral gap $\rho_k$: there exists $\rho_k \in [0, 1)$ such that $\|W_k - \frac{1}{n}\mathbf{1}\mathbf{1}^\top\|_2 \leq \rho_k$.

Assumption 5.5 is usually assumed for decentralized learning framework [21, 7, 12]. Specifically, $\rho_k = 0$ means the matrix $W_k$ with all elements $\frac{1}{n}$, corresponding to a fully connected graph $\mathcal{G}_k$ and $\rho_k \to 1$ means the matrix $W_k$ tends to be elements with either 0 or 1, corresponding to a graph that is nearly disconnected. Several works [26, 25] alternatively assume the spectral gap $\rho$ of a weighting matrix $W$ as the second largest eigenvalue of a doubly stochastic matrix $W$, i.e., $\rho = |\lambda_2(W)|$, and this spectral gap holds the same role for revealing the connectivity of the graph.

## 5.1 Convergence Analysis for HA-Fed: Full Participation

We first study the convergence behaviour of HA-Fed under full participation scenarios.

**Theorem 5.6** (HA-Fed full participation). Under Assumptions 5.1-5.5, if the local learning rate satisfies $\eta_l \leq \min\left\{\frac{\sqrt[4]{\epsilon}}{\alpha\sqrt{CC_0\tau(\tau + \rho_{\max}^2 D_{\tau,\rho})}}, \frac{\epsilon}{2\tau C_0 C_{\beta,\eta}}\right\}$, then the iterates of Algorithm 1 satisfy

$$\min_{t \in [T]} \mathbb{E}[\|\nabla f(\mathbf{x}_t)\|^2] \leq 8(\beta_2 \eta_l^2 \tau^2 G^2 + \epsilon)^{\frac{1}{2}} \left\{ \frac{f_0 - f_*}{\eta \eta_l \tau T} + \frac{\Psi}{T} + \Phi_1 + \Phi_2 \right\}, \quad (5.1)$$

where $\Psi = \frac{C_\beta G^2 d}{\sqrt{\epsilon}} + \frac{2C_\beta^2 \eta \eta_l \tau LG^2 d}{\epsilon}$, $\Phi_1 = \frac{CL^2 \eta_l^2}{4\sqrt{\epsilon}}\left[\tau^2 \sigma_g^2 + \tau \rho_{\max}^2 D_{\tau,\rho} \bar{\sigma}_L^2 + \tau \sigma^2\left(\frac{1}{n} + \rho_{\max}^2\right)\right]$, $\Phi_2 = C_{\beta,\eta} \frac{\eta_l}{2\epsilon N} \sigma^2$, where $C_\beta = \frac{\beta_1}{1 - \beta_1}$, $C_{\beta,\eta} = \left((C_\beta^2 + 3)\eta L + 2\sqrt{1 - \beta_2}G\right)$, where $C$ and $C_0$ are numerical constants that are irrelevant to parameters, $\rho_{\max} = \max_{k \in [K]} \rho_k$ is the maximum spectral gap of all $K$ clusters, $D_{\tau,\rho} = \min\left\{\frac{1}{1 - \rho_{\max}}, \tau\right\}$ describes the density and connectivity of clusters, and $\bar{\sigma}_L^2 = \frac{1}{K} \sum_{k=1}^K \sigma_k^2$ is the average dissimilarity between local clients in the same cluster.

**Remark 5.7.** The convergence rate Eq. 5.1 is composed of four terms. The first and second terms are related to $T$ and vanish as $T$ increases. The third term $\Phi_1$ represents the variance overhead introduced by both stochastic and inter-client variances. The last term $\Phi_2$ represents the stochastic variance from all $N$ clients. Note that only $\Phi_1$ is related to the cluster connectivity $\rho_{\max}$ while the other three

terms are identical to the corresponding term in the convergence rate of $N$-clients FedAMSGrad. Specifically, the dependency of $\Phi_1$ for HA-Fed is $\Phi_1 = \mathcal{O}\big(\eta_l^2\tau^2\sigma_g^2 + \eta_l^2\rho_{\max}^2\tau^2\bar\sigma_L^2 + \eta_l^2\big(\frac{1}{n} + \rho_{\max}^2\big)\tau\sigma^2\big)$, while the corresponding term $\widetilde\Phi_1$ for FedAMSGrad is $\mathcal{O}(\eta_l^2\tau^2\widetilde\sigma_g^2 + \eta_l^2\tau\sigma^2)$. When $\rho_{\max} = 0$, $\Phi_1$ in HA-Fed becomes $\mathcal{O}\big(\eta_l^2\tau^2\sigma_g^2 + \eta_l^2\tau\frac{\sigma^2}{n}\big)$, which is better than that of FedAMSGrad. And when $\rho_{\max} \to 1$, $\Phi_1$ in HA-Fed becomes $\mathcal{O}\big(\eta_l^2\tau^2(\sigma_g^2 + \bar\sigma_L^2) + \eta_l^2\tau\sigma^2\big)$, which matches the results in FedAMSGrad[5]. In terms of the overall convergence rate, since $\Phi_1$ in HA-Fed has the same order of dependency w.r.t. $\tau$ and $\eta_l$ as in FedAMSGrad, suppose we pick the learning rates $\eta = \Theta(\sqrt{\tau N})$ and $\eta_l = \Theta(1/\sqrt{T\tau^2})$ and when $T$ is sufficient large, i.e., $T > \tau N$, HA-Fed achieves the same convergence rate of $O(1/\sqrt{T\tau N})$ as FedAMSGrad [41] and also same as other general federated nonconvex optimization methods such as FedAvg [44, 43] and FedAdam [30].

## 5.2 Convergence Analysis for HA-Fed: Partial Participation

In such settings, we assume that only selected clients participate in each round of global synchronization. We assume the sampling strategy is random sampling without replacement in each cluster. Generally, at the beginning of global iteration $t$, the server samples a subset $\mathcal{S}_t^k$ for cluster $k$ that contains $m$ clients, these $M = Km$ clients receive the model from the server and synchronize their model difference for the global update.

**Theorem 5.8** (HA-Fed partial participation). Under Assumptions 5.1-5.5, if the local learning rate satisfies $\eta_l \leq \min\big\{\frac{1}{4C_0C_{\beta,\eta}(\tau-1)}, \frac{\sqrt[4]{\epsilon}}{2\alpha L\sqrt{\widetilde{C}C_0\tau(\tau+\rho_{\max}^2 D_{\tau,\rho})}}, \frac{1}{128\alpha^2\widetilde{C}C_0C_{\beta,\eta}\rho_{\max}^2 D_{\tau,\rho}}\big(\frac{n-m}{m(n-1)} + \frac{1}{\tau^2}\big)^{-1}\big\}$, then the iterates of Algorithm 1 in partial participation scenarios satisfy

$$\min_{t\in[T]}\mathbb{E}[\|\nabla f(\mathbf{x}_t)\|^2] \leq 8(\beta_2\eta_l^2\tau^2 G^2 + \epsilon)^{\frac{1}{2}}\Big\{\frac{f_0 - f_*}{\eta\eta_l\tau T} + \frac{\Psi}{T} + \Phi_1 + \Phi_2 + \Phi_3 + \Phi_4\Big\}, \qquad (5.2)$$

where $\Psi = \frac{C_\beta G^2 d}{\sqrt{\epsilon}} + \frac{2C_\beta^2\eta\eta_l\tau LG^2 d}{\epsilon}$, $\Phi_1 = \frac{CL^2}{4\sqrt{\epsilon}}\eta_l^2\big[\tau^2\sigma_g^2 + \tau\rho_{\max}^2 D_{\tau,\rho}\bar\sigma_L^2 + \tau\sigma^2\big(\frac{1}{n} + \rho_{\max}^2\big)\big]$, $\Phi_2 = C_{\beta,\eta}\big[1 + \big(\frac{n-m}{m}\big)\rho_{\max}^2\big]\frac{\eta_l}{\epsilon N}\sigma^2$, $\Phi_3 = \widetilde{C}C_{\beta,\eta}\cdot\frac{n-m}{m(n-1)}\eta_l D_{\tau,\rho}\rho_{\max}^2\big[\sigma_g^2 + \bar\sigma_L^2 + \sigma^2 + D_{\tau,\rho}\frac{\sigma^2}{\tau^2 n}\big]$, $\Phi_4 = \widetilde{C}C_{\beta,\eta}\cdot\eta_l^3 L^2 D_{\tau,\rho}\rho_{\max}^2\big[\sigma_g^2 + \bar\sigma_L^2 + \sigma^2 + D_{\tau,\rho}\frac{\sigma^2}{\tau^2 n}\big]$, where $C$, $\widetilde{C}$ and $C_0$ are numerical constants that are irrelevant to parameters, and $\rho_{\max}, D_{\tau,\rho}, \bar\sigma_L^2, C_{\beta,\eta}, C_\beta$ are same defined as Theorem 5.6.

**Remark 5.9.** When $\rho_{\max} = 0$, i.e., clients in each cluster are fully connected, in such case, there are $\Phi_1 = \mathcal{O}\big(\eta_l^2\tau^2\sigma_g^2 + \eta_l^2\tau\frac{\sigma^2}{n}\big)$, $\Phi_2 = \mathcal{O}\big(\frac{\eta_l\sigma^2}{N}\max\{\eta,1\}\big)$ and $\Phi_3 = \Phi_4 = 0$ in Eq. 5.2, which matches the result of fully participated HA-Fed with $\rho_{\max} = 0$. It is worth noting that although partially participated HA-Fed aggregates $M$ client models in each global round, since clients are fully connected inside the clusters, picking a part of the clients (inside each cluster) for global aggregation is the same as picking all the clients. Therefore, partially participated HA-Fed recovers to fully participated HA-Fed under such a setting.

**Remark 5.10.** When $\rho_{\max} \to 1$ and $K = 1$, i.e., all clients are tending to disconnected, HA-Fed will reduce to partial participated FedAMSGrad with $M$ clients. Under such cases, we have $D_{\tau,\rho} = \min\big\{\frac{1}{1-\rho_{\max}}, \tau\big\} = \tau$. By choosing same learning rates $\eta = \Theta(\sqrt{\tau M})$ and $\eta_l = \Theta(1/\sqrt{T\tau^2})$ as in FedAMSGrad, $\Phi_3 = \mathcal{O}\big(\frac{\sqrt\tau}{\sqrt{TM}}\big)$ dominates the convergence rate of HA-Fed, which recovers the convergence of partially participated FedAMSGrad.

Remark 5.9 and 5.10 implies that when clients are sparsely connected, the convergence of partial participated HA-Fed still suffers from *dilemma of local steps* as in FedAMSGrad, while HA-Fed indeed resolves the dilemma when clients are densely connected. Therefore, it is crucial to investigate how cluster connectivity helps solve the *dilemma of local steps*. The following corollary gives a precise characterization on condition of $\rho_{\max}$ needed for solving the *dilemma of local steps*.

**Corollary 5.11.** Suppose all clusters satisfies $\rho_{\max} \leq \frac{1}{2\sqrt{n-m}}$ and $K < n$, then by choosing the global learning rate $\eta = \Theta(\sqrt{\tau M})$ and local learning rate $\eta_l = \Theta(\frac{1}{\sqrt{T\tau}})$, when $T$ is sufficient large, i.e., $T > \tau M$, then the convergence rate for HA-Fed in partial participation settings satisfies $\min_{t\in[T]}\mathbb{E}[\|\nabla f(\mathbf{x}_t)\|^2] = \mathcal{O}\big(\frac{1}{\sqrt{T\tau M}}\big)$.

---

[5]$\widetilde\sigma_g^2$ is the global variance obtaining by a similar assumption on clients' loss function , i.e., the loss function on each client of FedAMSGard satisfies $\frac{1}{N}\sum_{i=1}^N\|\nabla f_i(\mathbf{x}) - \nabla f(\mathbf{x})\|^2 \leq \widetilde\sigma_g^2$.

**Remark 5.12.** Corollary 5.11 shows that HA-Fed successfully resolves the *dilemma of local steps*: larger number of local steps $\tau$ can now achieve a faster convergence rate if clusters satisfy certain constraints. Note that when $m = n$, i.e., in the full participation setting, this $\rho_{\max} \leq \frac{1}{2\sqrt{n-m}}$ condition imposes no actual constraint on $\rho_{\max}$. When $m$ becomes smaller, the requirements for $\rho_{\max}$ also get stronger, i.e., the local cluster needs to be more densely connected. Also, for a given number of total clients $N$, the condition $K < n$ implies the number of clients in each cluster is larger than the number of clusters in the network, which ensures that each cluster has enough clients for local gossip communications and thus can reduce the variance and resolve the *dilemma of local steps* in the partial participation settings.

## 6 Experiments

In this section, we present the empirical evaluations for the HA-Fed algorithm. We mainly compare HA-Fed with the adaptive federated optimization counterpart, FedAMSGrad, and also conduct several ablation studies related to the algorithm framework and the intra-cluster topology.

**Experimental Setup:** We compare our proposed HA-Fed with FedAMSGrad, on CIFAR-10/CIFAR-100 [23] using (1) ResNet-18 [13] model, and (2) ConvMixer[6] model [37], and Fashion MNIST [42] datasets using (1) ConvMixer model and (2) CNN model[7]. For HA-Fed, the global network topology is set up with 32 total clients, and they are equally divided into 4 clusters where each cluster contains 8 clients. We set the default partial participation ratio as $p = 0.25$, i.e., 2 clients participated per cluster per round. We adopt ring topology for all clusters by default with maximum spectral gap $\rho_{\max} = 0.805$. For FedAMSGrad, we set the number of clients and the partial participation ratio the same, i.e., 32 clients in total and 8 clients synchronize to the central server in each round. For both methods, we conduct $\tau = 48$ steps of local training with a batch size of 50. We search for the best training hyper-parameter for both models. Due to the space limit, we leave the CIFAR-10 and Fashion MNIST experiments as well as the other experimental details in Appendix F.

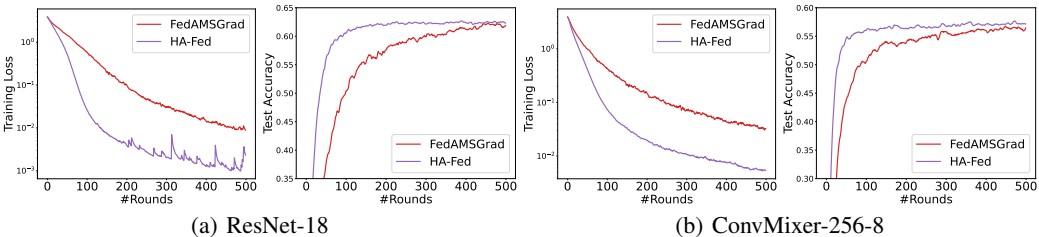

(a) ResNet-18         (b) ConvMixer-256-8

Figure 1: The learning curves for HA-Fed and FedAMSGrad in training CIFAR-100 data on (a) ResNet-18 model and (b) ConvMixer-256-8 model using ring topology for local communications.

Figure 1 shows the convergence result of HA-Fed and FedAMSGrad on training CIFAR-100 with ResNet-18 and ConvMixer-256-8 model. We compare the training loss and test accuracy against global rounds for both models. For the ResNet-18 model, HA-Fed achieves faster convergence than FedAMSGrad in reducing training loss, and HA-Fed grows rapidly to obtain an overall higher test accuracy. For the ConvMixer-256-8 model, HA-Fed again shows its faster convergence speed on training loss; in the meantime, HA-Fed still holds a higher test accuracy compared to FedAMSGrad under the same settings.

Now we study how the participation ratio $p$ and network connectivity $\rho_{\max}$ would affect the convergence of our proposed HA-Fed algorithm. Figure 2(a) illustrates the ablation study on the participation ratio $p$. Specifically, we test various values of $p$ from $p = \{0.125, 0.25, 0.5, 1.0\}$. From Figure 2(a), we observe that a larger participation ratio $p$ slightly improves the convergence on training loss. This is consistent with our theoretical convergence rate that increasing the number of participating clients improves the convergence rate, but the improvement is slight compared to a large number of global round $T$ and local steps $\tau$. Figure 2(b) then shows ablation study on clusters' maximum spectral gap $\rho_{\max}$. Specifically, we compare various of $\rho_{\max}$ from $\rho_{\max} = \{0, 0.125, 0.599, 0.805\}$ calculated

---

[6]ConvMixer shares similar ideas to vision transformer [9] to use patch embeddings to preserve locality and similarly, and it is trained via adaptive gradient methods by default.

[7]See details for the CNN model in Appendix F.3.

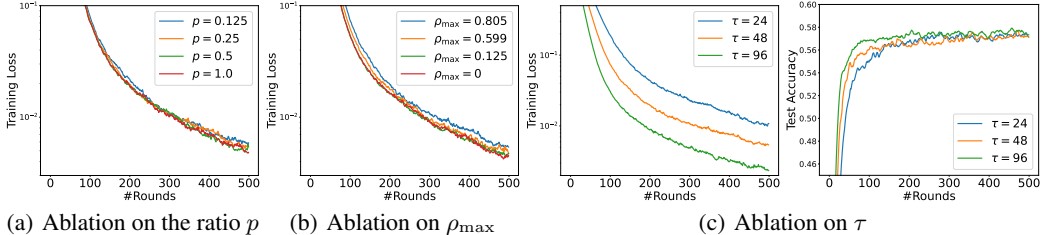

| (a) Ablation on the ratio $p$ | (b) Ablation on $\rho_{\max}$ | (c) Ablation on $\tau$ |

Figure 2: The learning curves with (a) different participating ratio $p$, (b) different maximum spectral gap $\rho_{\max}$ of clusters in training CIFAR-100 data on ConvMixer-256-8 model and (c) different numbers of local steps $\tau$ in training CIFAR-100 on ConvMixer-256-8 model.

by different network typologies. From Figure 2(b), we can observe that smaller $\rho_{\max}$ contributes to a faster convergence on training loss, which is shown as the red and green lines achieve faster convergence on training loss than the orange and blue lines. This result matches the theoretical result that $\rho_{\max}$ holds the non-dominant term in the convergence of HA-Fed even for partial participation scenarios. This suggests that without a dense network topology, HA-Fed can still take the benefit of gossip communication to achieve the expected convergence result.

We further study how the number of local update steps $\tau$ would affect the convergence of our proposed HA-Fed algorithm. Figure 2(c) shows the ablation study about the number of local steps $\tau$, we compare different $\tau$ from $\tau = \{24, 48, 96\}$. We observe that a larger number of local steps $\tau$ indeed helps accelerate convergence on training loss, as the green line ($\tau = 96$) in the left plot keeps the smallest training loss. From the right plot in Figure 2(c), larger $\tau$ generally achieves better generalization performance with higher test accuracy. This result backup our theory and show that HA-Fed achieves a faster convergence as the number of local steps increases, and HA-Fed indeed resolves the *dilemma of local steps*.

## 7  Conclusions

In this paper, we propose a novel hybrid adaptive federated optimization algorithm, HA-Fed, that overcomes the *dilemma of local steps* and achieves a faster convergence rate as the local training step increases. HA-Fed mitigates the impact of data heterogeneity by adding inexpensive client-to-client communications hence resolving the *dilemma of local steps* without extra client-to-server communications. We present a completed theoretical convergence analysis for the proposed HA-Fed. We prove that HA-Fed achieves a faster convergence rate than the previous adaptive federated optimization method for both full and partial participation scenarios with heterogeneous data under nonconvex stochastic settings. Experiments on several benchmarks and ablation studies verify our theory.

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

## A HA-Fed Algorithm for Partial Participation

In the following, we summarize the proposed HA-Fed in partial participation settings.

---
**Algorithm 2** HA-Fed:partial participation
---
**Input:** initial point $\mathbf{x}_1$, global step size $\eta$, local step size $\eta_l$, $\beta_1, \beta_2, \epsilon$, weighting matrix $W_k$ for all clusters $k \in [K]$

1: $\mathbf{m}_0 \leftarrow 0, \mathbf{v}_0 \leftarrow 0$
2: **for** $t = 1$ to $T$ **do**
3:     **for** each cluster $k \in [K]$ in parallel **do**
4:         Random sample a subset $\mathcal{S}_t^k$ of clients with $|\mathcal{S}_t^k| = m$
5:         **for** each client $i \in \mathcal{S}_t^k$ in parallel **do**
6:             Receive model from the server $\mathbf{x}_{t,0}^i = \mathbf{x}_t$
7:             Intra-cluster broadcast: $\mathbf{x}_{t,0}^j = \mathbf{x}_t, j \in \mathcal{V}_k$
8:         **end for**
9:         **for** each client $i \in \mathcal{V}_k$ in parallel **do**
10:            **for** $s = 0, ..., \tau - 1$ **do**
11:               Compute local stochastic gradient: $\mathbf{g}_{t,s}^i = \nabla F_i(\mathbf{x}_{t,s}^i; \xi_{t,s}^i)$
12:               $\mathbf{x}_{t,s+\frac{1}{2}}^i = \mathbf{x}_{t,s}^i - \eta_l \mathbf{g}_{t,s}^i$
13:               $\mathbf{x}_{t,s+1}^i = \sum_{j \in \mathcal{N}_k^i}(W_k)_{i,j}\mathbf{x}_{t,s+\frac{1}{2}}^j$
14:            **end for**
15:            Get the model difference: $\Delta_t^i = \mathbf{x}_{t,\tau}^i - \mathbf{x}_t$
16:         **end for**
17:     **end for**
18:     Collect model differences from selected clients: $\Delta_t = \frac{1}{K}\sum_{k \in [K]}\frac{1}{m}\sum_{i \in S_t^k}\Delta_t^i$
19:     $\mathbf{m}_t = \beta_1\mathbf{m}_{t-1} + (1 - \beta_1)\Delta_t$
20:     $\mathbf{v}_t = \beta_2\mathbf{v}_{t-1} + (1 - \beta_2)\Delta_t^2$
21:     $\widehat{\mathbf{v}}_t = \max(\widehat{\mathbf{v}}_{t-1}, \mathbf{v}_t)$ and $\widehat{\mathbf{V}}_t = \text{diag}(\widehat{\mathbf{v}}_t + \epsilon)$
22:     Server update $\mathbf{x}_{t+1} = \mathbf{x}_t + \eta\frac{\mathbf{m}_t}{\sqrt{\widehat{\mathbf{v}}_t}+\epsilon}$
23: **end for**

---

The proposed HA-Fed for partial participation settings is similar to full participation except for the broadcast and synchronization steps (Line 5-8 and Line 18 in Algorithm 2). For the partial participation setting, the server sends the current model $\mathbf{x}_t$ to $m$ selected clients in each cluster for

the broadcast step. Then these clients who received the model from the server will further broadcast the model $\mathbf{x}_t$ to their neighbors in the same cluster via fast client-to-client communication links. For example, if 8 clients within one cluster are grouped with ring topology, and two clients are sampled in each round, i.e., $m = 2$, then the partial participation setting needs 2 server-to-client broadcast rounds and 6 much cheaper client-to-client broadcast rounds. In contrast, the full participation setting needs 8 server-to-client broadcast rounds for the same cluster grouping strategy which is much more expensive. Moreover, only the selected clients send the model difference $\Delta_t^i$ to the server (Line 18 in Algorithm 2), which also reduces the client-to-server communication overhead. The rest of the algorithm is the same as full participation scenarios.

## B  Preliminaries

We define the following auxiliary sequences, w.r.t. $\mathbf{x}_{t,s}$, $\mathbf{x}_{t,s}^k$. Firstly, we denote the average model on cluster $k$ as

$$\bar{\mathbf{x}}_{t,s+1}^k = \bar{\mathbf{x}}_{t,s}^k - \eta_l \bar{\mathbf{g}}_{t,s}^k, \tag{B.1}$$

where $\bar{\mathbf{g}}_{t,s}^k = \frac{1}{n} \sum_{i \in \mathcal{V}_k} \mathbf{g}_{t,s}^i$. We also define the global average model

$$\bar{\mathbf{x}}_{t,s+1} = \bar{\mathbf{x}}_{t,s} - \eta_l \frac{1}{N} \sum_{i=1}^N \mathbf{g}_{t,s}^i. \tag{B.2}$$

We next define sequences related to model differences, we denote the average model difference on cluster $k$ as $\bar{\Delta}_t^k$, and the average global model difference $\bar{\Delta}_t$ without sampling consideration.

$$\bar{\Delta}_t^k = \frac{1}{n} \sum_{i \in \mathcal{V}_k} \Delta_t^i = \frac{1}{n} \sum_{i \in \mathcal{V}_k} (\mathbf{x}_{t,\tau}^i - \mathbf{x}_t) = \bar{\mathbf{x}}_{t,\tau}^k - \mathbf{x}_t = \bar{\mathbf{x}}_{t,0}^k - \eta_l \sum_{s=0}^{\tau-1} \bar{\mathbf{g}}_{t,s}^k - \mathbf{x}_t = -\eta_l \sum_{s=0}^{\tau-1} \bar{\mathbf{g}}_{t,s}^k$$

$$\bar{\Delta}_t = \frac{1}{K} \sum_{k \in [K]} \frac{1}{n} \sum_{i \in \mathcal{V}_k} (\mathbf{x}_{t,\tau}^i - \mathbf{x}_t) = \frac{1}{K} \sum_{k \in [K]} \Delta_t^k = -\eta_l \frac{1}{K} \frac{1}{n} \sum_{s=0}^{\tau-1} \sum_{k \in [K]} \sum_{i \in \mathcal{V}_k} \mathbf{g}_{t,s}^i, \tag{B.3}$$

recall the definition of $\Delta_t$

$$\Delta_t = \frac{1}{K} \sum_{k \in [K]} \frac{1}{m} \sum_{i \in S_t^k} \Delta_t^i = \frac{1}{K} \frac{1}{m} \sum_{k \in [K]} \sum_{i \in S_t^k} \mathbf{x}_{t,\tau}^i - \mathbf{x}_t, \tag{B.4}$$

note that for $\bar{\Delta}_t$, we have the following result, which shows that $\Delta_t$ in the algorithm is the unbiased estimation of global average model difference $\bar{\Delta}_t$.

$$\mathbb{E}_{S_t}[\Delta_t] = \bar{\Delta}_t = \frac{1}{K} \sum_{k \in [K]} \frac{1}{n} \sum_{i=1}^n \mathbf{x}_{t,\tau}^i - \mathbf{x}_t = \frac{1}{N} \sum_{i=1}^N \mathbf{x}_{t,\tau}^i - \mathbf{x}_t. \tag{B.5}$$

## C  Proof of Theorem 5.6: HA-Fed full participation

*Proof of Theorem 5.6.* For full participation cases, we have $\bar{\Delta}_t = \Delta_t$. Similar to previous works about adaptive methods [45, 5], we introduce a Lyapunov sequence $\mathbf{z}_t$: assume $\mathbf{x}_0 = \mathbf{x}_1$, for each $t \geq 1$, we have

$$\mathbf{z}_t = \mathbf{x}_t + \frac{\beta_1}{1 - \beta_1}(\mathbf{x}_t - \mathbf{x}_{t-1}) = \frac{1}{1 - \beta_1}\mathbf{x}_t - \frac{\beta_1}{1 - \beta_1}\mathbf{x}_{t-1}. \tag{C.1}$$

For the difference of two adjacent element in sequence $\mathbf{z}_t$, we have

$$\begin{aligned}
\mathbf{z}_{t+1} - \mathbf{z}_t &= \frac{1}{\beta_1}(\mathbf{x}_{t+1} - \mathbf{x}_t) - \frac{\beta_1}{1 - \beta_1}(\mathbf{x}_t - \mathbf{x}_{t-1}) \\
&= \frac{1}{1 - \beta_1}(\eta \widehat{\mathbf{V}}_t^{-1/2}\mathbf{m}_t) - \frac{\beta_1}{1 - \beta_1}\eta \widehat{\mathbf{V}}_{t-1}^{-1/2}\mathbf{m}_{t-1} \\
&= \frac{1}{1 - \beta_1}\eta \widehat{\mathbf{V}}_t^{-1/2}\left[\beta_1 \mathbf{m}_{t-1} + (1 - \beta_1)\Delta_t\right] - \frac{\beta_1}{1 - \beta_1}\eta \widehat{\mathbf{V}}_{t-1}^{-1/2}\mathbf{m}_{t-1} \\
&= \eta \widehat{\mathbf{V}}_t^{-1/2}\Delta_t - \eta \frac{\beta_1}{1 - \beta_1}\left(\widehat{\mathbf{V}}_{t-1}^{-1/2} - \widehat{\mathbf{V}}_t^{-1/2}\right)\mathbf{m}_{t-1}.
\end{aligned}$$

By Assumption 5.1, since $f$ is $L$-smooth, taking conditional expectation at time $t$, we have

$$\mathbb{E}[f(\mathbf{z}_{t+1})] - f(\mathbf{z}_t)$$

$$\leq \mathbb{E}[\langle \nabla f(\mathbf{z}_t), \mathbf{z}_{t+1} - \mathbf{z}_t \rangle] + \frac{L}{2} \mathbb{E}[\|\mathbf{z}_{t+1} - \mathbf{z}_t\|^2]$$

$$\leq \eta \mathbb{E}\left[\left\langle \nabla f(\mathbf{z}_t), \widehat{\mathbf{V}}_t^{-1/2} \Delta_t \right\rangle\right] - \eta \mathbb{E}\left[\left\langle \nabla f(\mathbf{z}_t), \frac{\beta_1}{1-\beta_1}\left(\widehat{\mathbf{V}}_{t-1}^{-1/2} - \widehat{\mathbf{V}}_t^{-1/2}\right)\mathbf{m}_{t-1} \right\rangle\right]$$

$$+ \frac{\eta^2 L}{2} \mathbb{E}\left[\left\|\widehat{\mathbf{V}}_t^{-1/2}\Delta_t - \frac{\beta_1}{1-\beta_1}\left(\widehat{\mathbf{V}}_{t-1}^{-1/2} - \widehat{\mathbf{V}}_t^{-1/2}\right)\mathbf{m}_{t-1}\right\|^2\right]$$

$$= \underbrace{\eta \mathbb{E}\left[\left\langle \nabla f(\mathbf{x}_t), \widehat{\mathbf{V}}_t^{-1/2}\Delta_t \right\rangle\right]}_{I_1} \underbrace{-\eta \mathbb{E}\left[\left\langle \nabla f(\mathbf{z}_t), \frac{\beta_1}{1-\beta_1}\left(\widehat{\mathbf{V}}_{t-1}^{-1/2} - \widehat{\mathbf{V}}_t^{-1/2}\right)\mathbf{m}_{t-1} \right\rangle\right]}_{I_2}$$

$$+ \underbrace{\frac{\eta^2 L}{2} \mathbb{E}\left[\left\|\widehat{\mathbf{V}}_t^{-1/2}\Delta_t - \frac{\beta_1}{1-\beta_1}\left(\widehat{\mathbf{V}}_{t-1}^{-1/2} - \widehat{\mathbf{V}}_t^{-1/2}\right)\mathbf{m}_{t-1}\right\|^2\right]}_{I_3} +$$

$$+ \quad \underbrace{\eta \mathbb{E}\left[\left\langle \nabla f(\mathbf{z}_t) - \nabla f(\mathbf{x}_t), \widehat{\mathbf{V}}_t^{-1/2}\Delta_t \right\rangle\right]}_{I_4}, \tag{C.2}$$

## C.1   Bounding $I_1$

We have

$$I_1 = \eta \mathbb{E}\left[\left\langle \nabla f(\mathbf{x}_t), \frac{\Delta_t}{\sqrt{\widehat{\mathbf{v}}_t} + \epsilon} \right\rangle\right]$$

$$= \eta \mathbb{E}\left[\left\langle \nabla f(\mathbf{x}_t), \frac{\Delta_t}{\sqrt{\beta_2 \widehat{\mathbf{v}}_{t-1}} + \epsilon} \right\rangle\right] + \eta \mathbb{E}\left[\left\langle \nabla f(\mathbf{x}_t), \frac{\Delta_t}{\sqrt{\widehat{\mathbf{v}}_t} + \epsilon} - \frac{\Delta_t}{\sqrt{\beta_2 \widehat{\mathbf{v}}_{t-1}} + \epsilon} \right\rangle\right]. \tag{C.3}$$

For the second term in Eq. C.3, we have

$$\eta \mathbb{E}\left[\left\langle \nabla f(\mathbf{x}_t), \frac{\Delta_t}{\sqrt{\widehat{\mathbf{v}}_t} + \epsilon} - \frac{\Delta_t}{\sqrt{\beta_2 \widehat{\mathbf{v}}_{t-1}} + \epsilon} \right\rangle\right]$$

$$\leq \eta \|\nabla f(\mathbf{x}_t)\| \mathbb{E}\left[\left\|\frac{1}{\sqrt{\widehat{\mathbf{v}}_t} + \epsilon} - \frac{1}{\sqrt{\beta_2 \widehat{\mathbf{v}}_{t-1}} + \epsilon}\right\| \cdot \|\Delta_t\|\right]$$

$$\leq \frac{\eta \sqrt{1-\beta_2} G}{\epsilon} \mathbb{E}[\|\Delta_t\|^2], \tag{C.4}$$

where the second inequality holds by Assumption 5.2, Lemma E.8 and Lemma E.10. For the first term in Eq. C.3, recall that $\Delta_t = -\frac{\eta_l}{N} \sum_{i=1}^{N} \sum_{s=0}^{\tau-1} \mathbf{g}_{t,s}^i$, we have

$$\eta \mathbb{E}\left[\left\langle \nabla f(\mathbf{x}_t), \frac{\Delta_t}{\sqrt{\beta_2 \widehat{\mathbf{v}}_{t-1}} + \epsilon} \right\rangle\right] = \eta \mathbb{E}\left[\left\langle \frac{\nabla f(\mathbf{x}_t)}{\sqrt{\beta_2 \widehat{\mathbf{v}}_{t-1}} + \epsilon}, \Delta_t \right\rangle\right]$$

$$= -\eta \eta_l \sum_{s=0}^{\tau-1} \mathbb{E}\left[\left\langle \frac{\nabla f(\mathbf{x}_t)}{\sqrt{\beta_2 \widehat{\mathbf{v}}_{t-1}} + \epsilon}, \frac{1}{N} \sum_{i=1}^{N} \mathbf{g}_{t,s}^i \right\rangle\right]$$

$$= -\eta \eta_l \sum_{s=0}^{\tau-1} \mathbb{E}\left[\left\langle \frac{\nabla f(\mathbf{x}_t)}{\sqrt{\beta_2 \widehat{\mathbf{v}}_{t-1}} + \epsilon}, \frac{1}{N} \sum_{i=1}^{N} \nabla f_i(\mathbf{x}_{t,s}^i) \right\rangle\right], \tag{C.5}$$

where we have

$$-\mathbb{E}\left[\left\langle \frac{\nabla f(\mathbf{x}_t)}{\sqrt{\beta_2\widehat{\mathbf{v}}_{t-1}+\epsilon}}, \frac{1}{N}\sum_{i=1}^{N}\nabla f_i(\mathbf{x}_{t,s}^i)\right\rangle\right]$$

$$=-\frac{1}{2}\mathbb{E}\left[\left\langle \frac{\nabla f(\mathbf{x}_t)}{\sqrt{\beta_2\widehat{\mathbf{v}}_{t-1}+\epsilon}}, \frac{1}{N}\sum_{i=1}^{N}\nabla f_i(\mathbf{x}_{t,s}^i)\right\rangle\right]$$

$$-\frac{1}{2}\mathbb{E}\left[\left\langle \frac{\nabla f(\mathbf{x}_t)}{\sqrt{\beta_2\widehat{\mathbf{v}}_{t-1}+\epsilon}}, \frac{1}{N}\sum_{i=1}^{N}\nabla f_i(\mathbf{x}_{t,s}^i)\pm\frac{1}{K}\sum_{k=1}^{K}\nabla\bar{f}_k(\bar{\mathbf{x}}_{t,s}^k)\right\rangle\right]$$

$$=-\frac{1}{2}\mathbb{E}\left[\left\langle \frac{\nabla f(\mathbf{x}_t)}{\sqrt[4]{\beta_2\widehat{\mathbf{v}}_{t-1}+\epsilon}}, \frac{1}{\sqrt[4]{\beta_2\widehat{\mathbf{v}}_{t-1}+\epsilon}}\frac{1}{N}\sum_{i=1}^{N}\nabla f_i(\mathbf{x}_{t,s}^i)\right\rangle\right]$$

$$-\frac{1}{2}\mathbb{E}\left[\left\langle \frac{\nabla f(\mathbf{x}_t)}{\sqrt[4]{\beta_2\widehat{\mathbf{v}}_{t-1}+\epsilon}}, \frac{1}{\sqrt[4]{\beta_2\widehat{\mathbf{v}}_{t-1}+\epsilon}}\left(\frac{1}{N}\sum_{i=1}^{N}\nabla f_i(\mathbf{x}_{t,s}^i)\pm\frac{1}{K}\sum_{k=1}^{K}\nabla\bar{f}_k(\bar{\mathbf{x}}_{t,s}^k)\right)\right\rangle\right]. \quad \text{(C.6)}$$

Since we have the following inequalities, $\langle \mathbf{a},\mathbf{b}\rangle = \|\mathbf{a}\|^2+\|\mathbf{b}\|^2-\|\mathbf{a}-\mathbf{b}\|^2$ and $\langle \mathbf{a},\mathbf{b}\rangle \le \frac{1}{2}\|\mathbf{a}\|^2+\frac{1}{2}\|\mathbf{b}\|^2$, then we have

$$-\mathbb{E}\left[\left\langle \frac{\nabla f(\mathbf{x}_t)}{\sqrt{\beta_2\widehat{\mathbf{v}}_{t-1}+\epsilon}}, \frac{1}{N}\sum_{i=1}^{N}\nabla f_i(\mathbf{x}_{t,s}^i)\right\rangle\right]$$

$$\le -\frac{1}{4}\mathbb{E}\left[\left\|\frac{\nabla f(\mathbf{x}_t)}{\sqrt[4]{\beta_2\widehat{\mathbf{v}}_{t-1}+\epsilon}}\right\|^2+\left\|\frac{1}{\sqrt[4]{\beta_2\widehat{\mathbf{v}}_{t-1}+\epsilon}}\frac{1}{N}\sum_{i=1}^{N}\nabla f_i(\mathbf{x}_{t,s}^i)\right\|^2\right.$$

$$-\left\|\frac{1}{\sqrt[4]{\beta_2\widehat{\mathbf{v}}_{t-1}+\epsilon}}\left(\nabla f(\mathbf{x}_t)-\frac{1}{N}\sum_{i=1}^{N}\nabla f_i(\mathbf{x}_{t,s}^i)\right)\right\|^2\right]-\frac{1}{4}\mathbb{E}\left[\left\|\frac{\nabla f(\mathbf{x}_t)}{\sqrt[4]{\beta_2\widehat{\mathbf{v}}_{t-1}+\epsilon}}\right\|^2\right.$$

$$+\left\|\frac{1}{\sqrt[4]{\beta_2\widehat{\mathbf{v}}_{t-1}+\epsilon}}\frac{1}{K}\sum_{k=1}^{K}\nabla\bar{f}_k(\bar{\mathbf{x}}_{t,s}^k)\right\|^2-\left\|\frac{1}{\sqrt[4]{\beta_2\widehat{\mathbf{v}}_{t-1}+\epsilon}}\left(\nabla f(\mathbf{x}_t)-\frac{1}{K}\sum_{k=1}^{K}\nabla\bar{f}_k(\bar{\mathbf{x}}_{t,s}^k)\right)\right\|^2\right]$$

$$+\frac{1}{4}\mathbb{E}\left[\left\|\frac{\nabla f(\mathbf{x}_t)}{\sqrt[4]{\beta_2\widehat{\mathbf{v}}_{t-1}+\epsilon}}\right\|^2\right]+\frac{1}{4}\mathbb{E}\left[\left\|\frac{1}{\sqrt[4]{\beta_2\widehat{\mathbf{v}}_{t-1}+\epsilon}}\left(\frac{1}{N}\sum_{i=1}^{N}\nabla f_i(\mathbf{x}_{t,s}^i)-\frac{1}{K}\sum_{k=1}^{K}\nabla\bar{f}_k(\bar{\mathbf{x}}_{t,s}^k)\right)\right\|^2\right]$$

$$\le -\frac{1}{4C_0}\mathbb{E}[\|\nabla f(\mathbf{x}_t)\|^2]-\frac{1}{4C_0}\mathbb{E}\left[\left\|\frac{1}{N}\sum_{i=1}^{N}\nabla f_i(\mathbf{x}_{t,s}^i)\right\|^2\right]-\frac{1}{4C_0}\mathbb{E}\left[\left\|\frac{1}{K}\sum_{k=1}^{K}\nabla\bar{f}_k(\bar{\mathbf{x}}_{t,s}^k)\right\|^2\right]$$

$$+\frac{1}{4\sqrt{\epsilon}}\mathbb{E}\left[\left\|\nabla f(\mathbf{x}_t)-\frac{1}{N}\sum_{i=1}^{N}\nabla f_i(\mathbf{x}_{t,s}^i)\right\|^2\right]+\frac{1}{4\sqrt{\epsilon}}\mathbb{E}\left[\left\|\nabla f(\mathbf{x}_t)-\frac{1}{K}\sum_{k=1}^{K}\nabla\bar{f}_k(\bar{\mathbf{x}}_{t,s}^k)\right\|^2\right]$$

$$+\frac{1}{4\sqrt{\epsilon}}\mathbb{E}\left[\left\|\frac{1}{N}\sum_{i=1}^{N}\nabla f_i(\mathbf{x}_{t,s}^i)-\frac{1}{K}\sum_{k=1}^{K}\nabla\bar{f}_k(\bar{\mathbf{x}}_{t,s}^k)\right\|^2\right], \quad \text{(C.7)}$$

where the second inequality holds by $\frac{\|\mathbf{x}\|^2}{C_0} \le \frac{\|\mathbf{x}\|^2}{\sqrt{\beta_2\widehat{\mathbf{v}}_{t-1}+\epsilon}} \le \frac{\|\mathbf{x}\|^2}{\sqrt{\epsilon}}$. Bounding the last three terms above are equal to bound the inter-cluster consensus error $\|\mathbf{x}_t - \bar{\mathbf{x}}_{t,s}^k\|$ and intra-cluster consensus error $\|\bar{\mathbf{x}}_{t,s}^k - \mathbf{x}_{t,s}^i\|$.

Merging pieces together, we can finally bound $I_1$ here.

$$
\begin{aligned}
I_1 &= \eta\mathbb{E}\left[\left\langle\nabla f(\mathbf{x}_t), \frac{\Delta_t}{\sqrt{\beta_2\widehat{\mathbf{v}}_{t-1}+\epsilon}}\right\rangle\right] + \eta\mathbb{E}\left[\left\langle\nabla f(\mathbf{x}_t), \frac{\Delta_t}{\sqrt{\widehat{\mathbf{v}}_t+\epsilon}} - \frac{\Delta_t}{\sqrt{\beta_2\widehat{\mathbf{v}}_{t-1}+\epsilon}}\right\rangle\right] \\
&\leq \frac{\eta\eta_l}{4C_0}\sum_{s=0}^{\tau-1}\left[-\mathbb{E}[\|\nabla f(\mathbf{x}_t)\|^2] - \mathbb{E}\left\|\frac{1}{N}\sum_{i=1}^N\nabla f_i(\mathbf{x}_{t,s}^i)\right\|^2 - \left\|\frac{1}{K}\sum_{k=1}^K\nabla\bar{f}_k(\bar{\mathbf{x}}_{t,s}^k)\right\|^2\right] \\
&\quad + \frac{\eta\eta_l}{4\sqrt{\epsilon}}\sum_{s=0}^{\tau-1}\left\{\mathbb{E}\left[\left\|\nabla f(\mathbf{x}_t) - \frac{1}{N}\sum_{i=1}^N\nabla f_i(\mathbf{x}_{t,s}^i)\right\|^2\right] + \mathbb{E}\left[\left\|\nabla f(\mathbf{x}_t) - \frac{1}{K}\sum_{k=1}^K\nabla\bar{f}_k(\bar{\mathbf{x}}_{t,s}^k)\right\|^2\right]\right. \\
&\quad \left. + \mathbb{E}\left[\left\|\frac{1}{N}\sum_{i=1}^N\nabla f_i(\mathbf{x}_{t,s}^i) - \frac{1}{K}\sum_{k=1}^K\nabla\bar{f}_k(\bar{\mathbf{x}}_{t,s}^k)\right\|^2\right]\right\} + \frac{\eta\sqrt{1-\beta_2}G}{\epsilon}\mathbb{E}[\|\Delta_t\|^2] \\
&\leq \frac{\eta\eta_l}{4C_0}\sum_{s=0}^{\tau-1}\left[-\mathbb{E}[\|\nabla f(\mathbf{x}_t)\|^2] - \mathbb{E}\left\|\frac{1}{N}\sum_{i=1}^N\nabla f_i(\mathbf{x}_{t,s}^i)\right\|^2 - \left\|\frac{1}{K}\sum_{k=1}^K\nabla\bar{f}_k(\bar{\mathbf{x}}_{t,s}^k)\right\|^2\right] \\
&\quad + \frac{\eta\eta_l}{4\sqrt{\epsilon}}\sum_{s=0}^{\tau-1}\left\{\mathbb{E}\left[\left\|\nabla f(\mathbf{x}_t) \pm \frac{1}{K}\sum_{k=1}^K\nabla\bar{f}_k(\bar{\mathbf{x}}_{t,s}^k) - \frac{1}{N}\sum_{i=1}^N\nabla f_i(\mathbf{x}_{t,s}^i)\right\|^2\right]\right. \\
&\quad + \frac{\eta\sqrt{1-\beta_2}G}{\epsilon}\mathbb{E}[\|\Delta_t\|^2] + \mathbb{E}\left[\left\|\nabla f(\mathbf{x}_t) - \frac{1}{K}\sum_{k=1}^K\nabla\bar{f}_k(\bar{\mathbf{x}}_{t,s}^k)\right\|^2\right] \\
&\quad \left. + \mathbb{E}\left[\left\|\frac{1}{N}\sum_{i=1}^N\nabla f_i(\mathbf{x}_{t,s}^i) - \frac{1}{K}\sum_{k=1}^K\nabla\bar{f}_k(\bar{\mathbf{x}}_{t,s}^k)\right\|^2\right]\right\} \\
&\leq \frac{\eta\eta_l}{4C_0}\sum_{s=0}^{\tau-1}\left[-\mathbb{E}[\|\nabla f(\mathbf{x}_t)\|^2] - \mathbb{E}\left\|\frac{1}{N}\sum_{i=1}^N\nabla f_i(\mathbf{x}_{t,s}^i)\right\|^2 - \left\|\frac{1}{K}\sum_{k=1}^K\nabla\bar{f}_k(\bar{\mathbf{x}}_{t,s}^k)\right\|^2\right] \\
&\quad + \frac{\eta\eta_l}{4\sqrt{\epsilon}}\sum_{s=0}^{\tau-1}\left\{\frac{2L^2}{K}\sum_{k=1}^K\mathbb{E}[\|\mathbf{x}_t - \bar{\mathbf{x}}_{t,s}^k\|^2] + \frac{2L^2}{N}\sum_{k=1}^K\sum_{i\in\mathcal{V}_k}\mathbb{E}[\|\bar{\mathbf{x}}_{t,s}^k - \mathbf{x}_{t,s}^i\|^2]\right. \\
&\quad \left. + \frac{L^2}{K}\sum_{k=1}^K\mathbb{E}[\|\mathbf{x}_t - \bar{\mathbf{x}}_{t,s}^k\|^2] + \frac{L^2}{N}\sum_{k=1}^K\sum_{i\in\mathcal{V}_k}\mathbb{E}[\|\bar{\mathbf{x}}_{t,s}^k - \mathbf{x}_{t,s}^i\|^2]\right\} + \frac{\eta\sqrt{1-\beta_2}G}{\epsilon}\mathbb{E}[\|\Delta_t\|^2] \\
&\leq \frac{\eta\eta_l}{4C_0}\sum_{s=0}^{\tau-1}\left[-\mathbb{E}[\|\nabla f(\mathbf{x}_t)\|^2] - \mathbb{E}\left\|\frac{1}{N}\sum_{i=1}^N\nabla f_i(\mathbf{x}_{t,s}^i)\right\|^2 - \left\|\frac{1}{K}\sum_{k=1}^K\nabla\bar{f}_k(\bar{\mathbf{x}}_{t,s}^k)\right\|^2\right] \\
&\quad + \frac{\eta\eta_l}{4\sqrt{\epsilon}}\sum_{s=0}^{\tau-1}\left\{\frac{3L^2}{K}\sum_{k=1}^K\mathbb{E}[\|\mathbf{x}_t - \bar{\mathbf{x}}_{t,s}^k\|^2] + \frac{3L^2}{N}\sum_{k=1}^K\sum_{i\in\mathcal{V}_k}\mathbb{E}[\|\bar{\mathbf{x}}_{t,s}^k - \mathbf{x}_{t,s}^i\|^2]\right\} \\
&\quad + \frac{\eta\sqrt{1-\beta_2}G}{\epsilon}\mathbb{E}[\|\Delta_t\|^2] \\
&\leq \frac{\eta\eta_l}{4C_0}\sum_{s=0}^{\tau-1}\left[-\mathbb{E}[\|\nabla f(\mathbf{x}_t)\|^2] - \mathbb{E}\left\|\frac{1}{N}\sum_{i=1}^N\nabla f_i(\mathbf{x}_{t,s}^i)\right\|^2 - \left\|\frac{1}{K}\sum_{k=1}^K\nabla\bar{f}_k(\bar{\mathbf{x}}_{t,s}^k)\right\|^2\right] \\
&\quad + \frac{\eta\eta_l}{4\sqrt{\epsilon}}\left[3L^2\tau^2C_1\eta_l^2(\tau + \rho_{\max}^2 D_{\tau,\rho})(\alpha^2\mathbb{E}\|\nabla f(\mathbf{x}_t)\|^2 + \sigma_g^2) + 3L^2\tau^2C_1\rho_{\max}^2 D_{\tau,\rho}\eta_l^2\bar{\sigma}_L^2\right. \\
&\quad \left. + 3L^2\tau^2C_1\eta_l^2\sigma^2\rho_{\max}^2 + 3L^2C_1\left(\tau^2 + D_{\tau,\rho}^2\cdot\rho_{\max}^2\right)\eta_l^2\frac{\sigma^2}{n}\right] + \frac{\eta\sqrt{1-\beta_2}G}{\epsilon}\mathbb{E}[\|\Delta_t\|^2], \quad\text{(C.8)}
\end{aligned}
$$

where the last inequality holds by Lemma E.1 and E.2.

## C.2   Bounding $I_2$

The bound for $I_2$ mainly follows by the update rule and definition of virtual sequence $\mathbf{z}_t$,

$$I_2 = -\eta \mathbb{E}\left[\left\langle \nabla f(\mathbf{z}_t), \frac{\beta_1}{1-\beta_1}(\widehat{\mathbf{V}}_{t-1}^{-1/2} - \widehat{\mathbf{V}}_t^{-1/2})\mathbf{m}_{t-1}\right\rangle\right]$$

$$= -\eta \mathbb{E}\left[\left\langle \nabla f(\mathbf{z}_t) - \nabla f(\mathbf{x}_t) + \nabla f(\mathbf{x}_t), \frac{\beta_1}{1-\beta_1}(\widehat{\mathbf{V}}_{t-1}^{-1/2} - \widehat{\mathbf{V}}_t^{-1/2})\mathbf{m}_{t-1}\right\rangle\right], \qquad \text{(C.9)}$$

then by Assumption 5.1, we have

$$I_2 \leq \eta \mathbb{E}\left[\|\nabla f(\mathbf{x}_t)\|\left\|\frac{\beta_1}{1-\beta_1}(\widehat{\mathbf{V}}_{t-1}^{-1/2} - \widehat{\mathbf{V}}_t^{-1/2})\mathbf{m}_{t-1}\right\|\right]$$

$$+ \eta L \mathbb{E}\left[\|\mathbf{z}_t - \mathbf{x}_t\|\left\|\frac{\beta_1}{1-\beta_1}(\widehat{\mathbf{V}}_{t-1}^{-1/2} - \widehat{\mathbf{V}}_t^{-1/2})\mathbf{m}_{t-1}\right\|\right]$$

$$= \eta \mathbb{E}\left[\|\nabla f(\mathbf{x}_t)\|\left\|\frac{\beta_1}{1-\beta_1}(\widehat{\mathbf{V}}_{t-1}^{-1/2} - \widehat{\mathbf{V}}_t^{-1/2})\mathbf{m}_{t-1}\right\|\right]$$

$$+ \eta^2 L \mathbb{E}\left[\left\|\frac{1}{\sqrt{\widehat{\mathbf{v}}_{t-1}} + \epsilon}\frac{\beta_1}{1-\beta_1}\mathbf{m}_{t-1}\right\|\left\|\frac{\beta_1}{1-\beta_1}(\widehat{\mathbf{V}}_{t-1}^{-1/2} - \widehat{\mathbf{V}}_t^{-1/2})\mathbf{m}_{t-1}\right\|\right]$$

$$\leq \eta \frac{\beta_1}{1-\beta_1}\eta_l \tau G^2 \mathbb{E}\left[\|\widehat{\mathbf{V}}_{t-1}^{-1/2} - \widehat{\mathbf{V}}_t^{-1/2}\|_1\right] + \eta^2 \frac{\beta_1^2}{(1-\beta_1)^2} L \eta_l^2 \tau^2 G^2 \epsilon^{-1/2} \mathbb{E}\left[\|\widehat{\mathbf{V}}_{t-1}^{-1/2} - \widehat{\mathbf{V}}_t^{-1/2}\|_1\right],$$

$$\text{(C.10)}$$

where the last inequality holds by Assumption 5.2 and Lemma E.8 about bounding $\nabla f(\mathbf{x}_t)$ and $\mathbf{m}_t$.

## C.3   Bounding $I_3$

We have the following result for bounding $I_3$

$$I_3 = \frac{\eta^2 L}{2}\mathbb{E}\left[\left\|\widehat{\mathbf{V}}_t^{-1/2}\Delta_t + \frac{\beta_1}{1-\beta_1}(\widehat{\mathbf{V}}_{t-1}^{-1/2} - \widehat{\mathbf{V}}_t^{-1/2})\mathbf{m}_{t-1}\right\|^2\right]$$

$$\leq \eta^2 L \mathbb{E}\left[\|\widehat{\mathbf{V}}_t^{-1/2}\Delta_t\|^2\right] + \eta^2 L \mathbb{E}\left[\left\|\frac{\beta_1}{1-\beta_1}(\widehat{\mathbf{V}}_{t-1}^{-1/2} - \widehat{\mathbf{V}}_t^{-1/2})\mathbf{m}_{t-1}\right\|^2\right]$$

$$\leq \frac{\eta^2 L}{\epsilon}\mathbb{E}[\|\Delta_t\|^2] + \eta^2 L \frac{\beta_1^2}{(1-\beta_1)^2}\eta_l^2 \tau^2 G^2 \mathbb{E}\left[\|\widehat{\mathbf{V}}_{t-1}^{-1/2} - \widehat{\mathbf{V}}_t^{-1/2}\|^2\right], \qquad \text{(C.11)}$$

where the first inequality follows by Cauchy-Schwarz inequality, and the second one follows by Assumption 5.2 and Lemma E.8 about bounding $\nabla f(\mathbf{x}_t)$ and $\mathbf{m}_t$.

## C.4   Bounding $I_4$

$$I_4 = \mathbb{E}\left[\left\langle \nabla f(\mathbf{z}_t) - \nabla f(\mathbf{x}_t), \eta \widehat{\mathbf{V}}_t^{-1/2}\Delta_t\right\rangle\right]$$

$$\leq \mathbb{E}\left[\|\nabla f(\mathbf{z}_t) - \nabla f(\mathbf{x}_t)\|\left\|\eta \widehat{\mathbf{V}}_t^{-1/2}\Delta_t\right\|\right]$$

$$\leq L \mathbb{E}\left[\|\mathbf{z}_t - \mathbf{x}_t\|\left\|\eta \widehat{\mathbf{V}}_t^{-1/2}\Delta_t\right\|\right]$$

$$\leq \frac{\eta^2 L}{2}\mathbb{E}\left[\left\|\frac{\beta_1}{1-\beta_1}\widehat{\mathbf{V}}_t^{-1/2}\mathbf{m}_t\right\|^2\right] + \frac{\eta^2 L}{2}\mathbb{E}\left[\left\|\widehat{\mathbf{V}}_t^{-1/2}\Delta_t\right\|^2\right]$$

$$\leq \frac{\eta^2 L}{2\epsilon}\frac{\beta_1^2}{(1-\beta_1)^2}\mathbb{E}[\|\mathbf{m}_t\|^2] + \frac{\eta^2 L}{2\epsilon}\mathbb{E}[\|\Delta_t\|^2], \qquad \text{(C.12)}$$

where the first inequality holds due to Young's inequality, and the second one follows from Assumption 5.1 and the definition of virtual sequence $\mathbf{z}_t$. By Lemma E.7, we have

$$\sum_{t=1}^T \mathbb{E}[\|\mathbf{m}_t\|^2] \leq \sum_{t=1}^T \mathbb{E}[\|\Delta_t\|^2]. \qquad \text{(C.13)}$$

Therefore, the summation of $I_4$ term is bounded by

$$\sum_{t=1}^{T} I_4 \leq \left( \frac{\eta^2 L}{2\epsilon} \frac{\beta_1^2}{(1-\beta_1)^2} + \frac{\eta^2 L}{2\epsilon} \right) \sum_{t=1}^{T} \mathbb{E}[\|\Delta_t\|^2]. \tag{C.14}$$

## C.5 Merging pieces together

Summing $I_1$ to $I_4$ from $t = 1$ to $T$, we have

$$\mathbb{E}[f(\mathbf{z}_{T+1})] - f(\mathbf{z}_1) = \sum_{t=1}^{T}[I_1 + I_2 + I_3 + I_4]$$

$$\leq \frac{\eta\eta_l}{4C_0} \sum_{t=1}^{T} \left[ -\tau\mathbb{E}[\|\nabla f(\mathbf{x}_t)\|^2] - \sum_{s=0}^{\tau-1} \mathbb{E}\left\| \frac{1}{N}\sum_{i=1}^{N} \nabla f_i(\mathbf{x}_{t,s}^i) \right\|^2 - \sum_{s=0}^{\tau-1} \left\| \frac{1}{K}\sum_{k=1}^{K} \nabla \bar{f}_k(\bar{\mathbf{x}}_{t,s}^k) \right\|^2 \right]$$

$$+ \frac{\eta\eta_l}{4\sqrt{\epsilon}} \sum_{t=1}^{T} \left\{ 3L^2\tau^2 C_1\eta_l^2(\tau + \rho_{\max}^2 D_{\tau,\rho})(\alpha^2\mathbb{E}\|\nabla f(\mathbf{x}_t)\|^2 + \sigma_g^2) + 3L^2\tau^2 C_1\rho_{\max}^2 D_{\tau,\rho}\eta_l^2\bar{\sigma}_L^2 \right.$$

$$+ 3L^2\tau^2 C_1\eta_l^2\sigma^2\rho_{\max}^2 + 3L^2 C_1\left(\tau^2 + D_{\tau,\rho}^2 \cdot \rho_{\max}^2\right)\eta_l^2\frac{\sigma^2}{n} \right\} + \frac{\eta\sqrt{1-\beta_2}G}{\epsilon} \sum_{t=1}^{T} \mathbb{E}[\|\Delta_t\|^2]$$

$$+ \left( \eta\frac{\beta_1}{1-\beta_1}\eta_l\tau G^2 + \eta^2\frac{\beta_1^2}{(1-\beta_1)^2}L\eta_l^2\tau^2 G^2\epsilon^{-1/2} \right) \sum_{t=1}^{T} \mathbb{E}\left[ \left\| \widehat{\mathbf{V}}_{t-1}^{-1/2} - \widehat{\mathbf{V}}_t^{-1/2} \right\|_1 \right]$$

$$+ \frac{\eta^2 L}{\epsilon} \sum_{t=1}^{T} \mathbb{E}[\|\Delta_t\|^2] + \eta^2 L\frac{\beta_1^2}{(1-\beta_1)^2}\eta_l^2\tau^2 G^2 \sum_{t=1}^{T} \mathbb{E}\left[ \left\| \widehat{\mathbf{V}}_{t-1}^{-1/2} - \widehat{\mathbf{V}}_t^{-1/2} \right\|^2 \right]$$

$$+ \left( \frac{\eta^2 L}{2\epsilon} \frac{\beta_1^2}{(1-\beta_1)^2} + \frac{\eta^2 L}{2\epsilon} \right) \sum_{t=1}^{T} \mathbb{E}[\|\Delta_t\|^2],$$

Merge the similar pieces, then we have

$$\mathbb{E}[f(\mathbf{z}_{T+1})] - f(\mathbf{z}_1)$$

$$\leq \frac{\eta\eta_l}{4C_0} \sum_{t=1}^{T} \left[ -\tau\mathbb{E}[\|\nabla f(\mathbf{x}_t)\|^2] - \sum_{s=0}^{\tau-1} \mathbb{E}\left\| \frac{1}{N}\sum_{i=1}^{N} \nabla f_i(\mathbf{x}_{t,s}^i) \right\|^2 - \sum_{s=0}^{\tau-1} \left\| \frac{1}{K}\sum_{k=1}^{K} \nabla \bar{f}_k(\bar{\mathbf{x}}_{t,s}^k) \right\|^2 \right]$$

$$+ \frac{\eta\eta_l}{4\sqrt{\epsilon}} \sum_{t=1}^{T} \left\{ 3L^2\tau^2 C_1\eta_l^2(\tau + \rho_{\max}^2 D_{\tau,\rho})(\alpha^2\mathbb{E}\|\nabla f(\mathbf{x}_t)\|^2 + \sigma_g^2) + 3L^2\tau^2 C_1\rho_{\max}^2 D_{\tau,\rho}\eta_l^2\bar{\sigma}_L^2 \right.$$

$$+ 3L^2\tau^2 C_1\eta_l^2\sigma^2\rho_{\max}^2 + 3L^2 C_1\left(\tau^2 + D_{\tau,\rho}^2 \cdot \rho_{\max}^2\right)\eta_l^2\frac{\sigma^2}{n} \right\}$$

$$+ \left( \eta\frac{\beta_1}{1-\beta_1}\eta_l\tau G^2 + \eta^2\frac{\beta_1^2}{(1-\beta_1)^2}L\eta_l^2\tau^2 G^2\epsilon^{-1/2} \right) \sum_{t=1}^{T} \mathbb{E}\left[ \left\| \widehat{\mathbf{V}}_{t-1}^{-1/2} - \widehat{\mathbf{V}}_t^{-1/2} \right\|_1 \right]$$

$$+ \eta^2 L\frac{\beta_1^2}{(1-\beta_1)^2}\eta_l^2\tau^2 G^2 \sum_{t=1}^{T} \mathbb{E}\left[ \left\| \widehat{\mathbf{V}}_{t-1}^{-1/2} - \widehat{\mathbf{V}}_t^{-1/2} \right\|^2 \right]$$

$$+ \left( \frac{\eta^2 L}{2\epsilon} \frac{\beta_1^2}{(1-\beta_1)^2} + \frac{\eta^2 L}{2\epsilon} + \frac{\eta^2 L}{\epsilon} + \frac{\eta\sqrt{1-\beta_2}G}{\epsilon} \right) \sum_{t=1}^{T} \mathbb{E}[\|\Delta_t\|^2],$$

Further apply Lemma E.9 and Lemma E.5, we have

$$\mathbb{E}[f(\mathbf{z}_{T+1})] - f(\mathbf{z}_1)$$

$$\leq \frac{\eta\eta_l}{4C_0} \sum_{t=1}^{T} \left[ -\tau\mathbb{E}[\|\nabla f(\mathbf{x}_t)\|^2] - \sum_{s=0}^{\tau-1} \mathbb{E}\left\| \frac{1}{N}\sum_{i=1}^{N} \nabla f_i(\mathbf{x}_{t,s}^i) \right\|^2 - \sum_{s=0}^{\tau-1} \left\| \frac{1}{K}\sum_{k=1}^{K} \nabla \bar{f}_k(\bar{\mathbf{x}}_{t,s}^k) \right\|^2 \right]$$

$$+ \frac{\eta\eta_l}{4\sqrt{\epsilon}} \sum_{t=1}^{T} \left\{ 3L^2\tau^2 C_1\eta_l^2(\tau + \rho_{\max}^2 D_{\tau,\rho})(\alpha^2\mathbb{E}\|\nabla f(\mathbf{x}_t)\|^2 + \sigma_g^2) + 3L^2\tau^2 C_1\rho_{\max}^2 D_{\tau,\rho}\eta_l^2\bar{\sigma}_L^2 \right.$$

$$+ 3L^2\tau^2 C_1\eta_l^2\sigma^2\rho_{\max}^2 + 3L^2 C_1(\tau^2 + D_{\tau,\rho}^2 \cdot \rho_{\max}^2)\eta_l^2\frac{\sigma^2}{n} \bigg\}$$

$$+ \left( \eta\frac{\beta_1}{1-\beta_1}\eta_l\tau G^2 + \eta^2\frac{\beta_1^2}{(1-\beta_1)^2}L\eta_l^2\tau^2 G^2\epsilon^{-1/2} \right)\frac{d}{\sqrt{\epsilon}} + \eta^2 L\frac{\beta_1^2}{(1-\beta_1)^2}\eta_l^2\tau^2 G^2 \cdot \frac{d}{\epsilon}$$

$$+ \left( \frac{\eta^2 L}{2\epsilon}\frac{\beta_1^2}{(1-\beta_1)^2} + \frac{\eta^2 L}{2\epsilon} + \frac{\eta^2 L}{\epsilon} + \frac{\eta\sqrt{1-\beta_2}G}{\epsilon} \right)\sum_{t=1}^{T}\left\{ \frac{\eta_l^2\tau}{N}\sigma^2 \right.$$

$$+ \eta_l^2\tau\sum_{s=0}^{\tau-1}\mathbb{E}\left[ \left\| \frac{1}{N}\sum_{i=1}^{N}\nabla f_i(\mathbf{x}_{t,s}^i) \right\|^2 \right] \bigg\}, \tag{C.15}$$

drop extra terms in the second line with the following condition on learning rate,

$$\eta\eta_l^2\tau\frac{C_{\beta,\eta}}{2\epsilon} \leq \frac{\eta\eta_l}{4C_0} \Rightarrow \eta \leq \frac{\epsilon}{2\tau C_0 C_{\beta,\eta}}, \tag{C.16}$$

where $C_{\beta,\eta} = \left( \eta L\frac{\beta_1^2}{(1-\beta_1)^2} + 3\eta L + 2\sqrt{1-\beta_2}G \right) = \mathcal{O}(\max\{\eta, 1\})$, merge the similar items,

$$\mathbb{E}[f(\mathbf{z}_{T+1})] - f(\mathbf{z}_1)$$

$$\leq -\left[ \frac{\eta\eta_l\tau}{4C_0} - \frac{\eta\eta_l}{4\sqrt{\epsilon}}3L^2\tau^2 C_1\eta_l^2(\tau + \rho_{\max}^2 D_{\tau,\rho})\alpha^2 \right]\sum_{t=1}^{T}\mathbb{E}[\|\nabla f(\mathbf{x}_t)\|^2]$$

$$+ \frac{\eta\eta_l}{4\sqrt{\epsilon}}\sum_{t=1}^{T}\left\{ 3L^2\tau^2 C_1\eta_l^2(\tau + \rho_{\max}^2 D_{\tau,\rho})\sigma_g^2 + 3L^2\tau^2 C_1\rho_{\max}^2 D_{\tau,\rho}\eta_l^2\bar{\sigma}_L^2 \right.$$

$$+ 3L^2\tau^2 C_1\eta_l^2\sigma^2\rho_{\max}^2 + 3L^2 C_1(\tau^2 + D_{\tau,\rho}^2 \cdot \rho_{\max}^2)\eta_l^2\frac{\sigma^2}{n} \bigg\}$$

$$+ \eta\frac{\beta_1}{1-\beta_1}\eta_l\tau G^2\frac{d}{\sqrt{\epsilon}} + 2\eta^2 L\frac{\beta_1^2}{(1-\beta_1)^2}\eta_l^2\tau^2 G^2 \cdot \frac{d}{\epsilon}$$

$$+ \left( \frac{\eta^2 L}{2\epsilon}\frac{\beta_1^2}{(1-\beta_1)^2} + \frac{\eta^2 L}{2\epsilon} + \frac{\eta^2 L}{\epsilon} + \frac{\eta\sqrt{1-\beta_2}G}{\epsilon} \right)\frac{T\eta_l^2\tau}{N}\sigma^2, \tag{C.17}$$

note that we need the following requirement for local learning rate $\eta_l$:

$$\eta_l \leq \frac{\sqrt[4]{\epsilon}}{\alpha\sqrt{6C_0 C_1\tau(\tau + \rho_{\max}^2 D_{\tau,\rho})}}. \tag{C.18}$$

Thus we have

$$\frac{\eta\eta_l\tau}{8C_0 T}\sum_{t=1}^{T}\mathbb{E}[\|\nabla f(\mathbf{x}_t)\|^2]$$

$$\leq \frac{\mathbb{E}[f(\mathbf{z}_{T+1})] - f(\mathbf{z}_T)}{T} + \eta\frac{\beta_1}{1-\beta_1}\eta_l\tau G^2\frac{d}{T\sqrt{\epsilon}} + 2\eta^2 L\frac{\beta_1^2}{(1-\beta_1)^2}\eta_l^2\tau^2 G^2 \cdot \frac{d}{T\epsilon}$$

$$+ \frac{\eta\eta_l}{4\sqrt{\epsilon}}\left( 3L^2\tau^2 C_1\eta_l^2(\tau + \rho_{\max}^2 D_{\tau,\rho})\sigma_g^2 \right.$$

$$+ 3L^2\tau^2 C_1\rho_{\max}^2 D_{\tau,\rho}\eta_l^2\bar{\sigma}_L^2 + 3L^2\tau^2 C_1\eta_l^2\sigma^2\rho_{\max}^2 + 3L^2 C_1(\tau^2 + D_{\tau,\rho}^2 \cdot \rho_{\max}^2)\eta_l^2\frac{\sigma^2}{n} \bigg)$$

$$+ \left( \frac{\eta^2 L}{2\epsilon}\frac{\beta_1^2}{(1-\beta_1)^2} + \frac{\eta^2 L}{2\epsilon} + \frac{\eta^2 L}{\epsilon} + \frac{\eta\sqrt{1-\beta_2}G}{\epsilon} \right)\frac{\eta_l^2\tau}{N}\sigma^2, \tag{C.19}$$

since we have $D_{\tau,\rho} \le \tau$ and the maximum spectral gap satisfies $\rho_{\max} \le 1$,

$$3L^2\tau^2 C_1 \rho_{\max}^2 D_{\tau,\rho} \eta_l^2 \bar\sigma_L^2 + 3L^2\tau^2 C_1 \eta_l^2 \sigma^2 \rho_{\max}^2 + 3L^2 C_1 (\tau^2 + D_{\tau,\rho}^2 \rho_{\max}^2)\eta_l^2 \frac{\sigma^2}{n}$$

$$= 3C_1 L^2 \eta_l^2 \left( \tau^2 \rho_{\max}^2 (D_{\tau,\rho}\bar\sigma_L^2 + \sigma^2) + (\tau^2 + D_{\tau,\rho}^2\rho_{\max}^2)\frac{\sigma^2}{n} \right)$$

$$\le 3C_1 L^2 \eta_l^2 \left( \tau^2 \rho_{\max}^2 D_{\tau,\rho}\bar\sigma_L^2 + \tau^2\rho_{\max}^2\sigma^2 + (\tau^2 + \tau^2\rho_{\max}^2)\frac{\sigma^2}{n} \right)$$

$$\le 6C_1 L^2 \eta_l^2 \left( \tau^2 \rho_{\max}^2 D_{\tau,\rho}\bar\sigma_L^2 + \tau^2\sigma^2 \left( \frac{1}{n} + \rho_{\max}^2 \right) \right),$$

and

$$3L^2\tau C_1 \eta_l^2(\tau + \rho_{\max}^2 D_{\tau,\rho})\sigma_g^2 \le 6L^2\tau^2 C_1 \eta_l^2 \sigma_g^2,$$

hence with a universal constant $C$, we have the following derivation for iterations,

$$\frac{1}{T}\sum_{t=1}^{T}\mathbb{E}[\|\nabla f(\mathbf{x}_t)\|^2]$$

$$\le 8C_0 \Bigg\{ \frac{\mathbb{E}[f(\mathbf{z}_{T+1})] - f(\mathbf{z}_T)}{\eta\eta_l\tau T} + \frac{1}{T}\left( \frac{C_\beta G^2 d}{\sqrt{\epsilon}} + \frac{2C_\beta^2 \eta\eta_l\tau L G^2 d}{\epsilon} \right)$$

$$+ \frac{CL^2\eta_l^2}{4\sqrt{\epsilon}}\left[ \tau^2\sigma_g^2 + \left( \tau\rho_{\max}^2 D_{\tau,\rho}\bar\sigma_L^2 + \tau\sigma^2\left(\frac{1}{n} + \rho_{\max}^2\right) \right) \right] + C_\beta \frac{\eta_l}{2\epsilon N}\sigma^2 \Bigg\}, \qquad \text{(C.20)}$$

$C$ is a constant irrelevant to parameters $\rho_{\max} = \max_{k\in[K]}\rho_k$, $D_{\tau,\rho} = \min\left\{\frac{1}{1-\rho_{\max}},\tau\right\}$, $\bar\sigma_L^2 = \frac{1}{K}\sum_{k=1}^{K}\sigma_k^2$ $C_\beta = \frac{\beta_1}{1-\beta_1}$ and $C_{\beta,\eta} = \left(\eta L\frac{\beta_1^2}{(1-\beta_1)^2} + 3\eta L + 2\sqrt{1-\beta_2}G\right) = \mathcal{O}(\max\{\eta,1\})$. $\qquad\square$

## D  Proof of Theorem 5.8: HA-Fed partial participation

*Proof of Theorem 5.8.* From Section B, we know that $\Delta_t$ is an unbiased estimation of $\bar\Delta_t$, thus the main difference between full participation and partial participation lies in the second order momentum estimation of model difference $\Delta_t$, i.e., $\mathbb{E}[\|\Delta_t\|^2]$. Hence with a different bounded $\mathbb{E}[\|\Delta_t\|^2]$ in Section E.4 for partial participation, we have the following result starting from $L$-smooth expansion like Eq. C.2,

$$\mathbb{E}[f(\mathbf{z}_{t+1})] - f(\mathbf{z}_t) \le I_1' + I_2' + I_3' + I_4', \qquad \text{(D.1)}$$

where $I_i'$ corresponds to the similar term of $I_i$ in Eq. C.2.

By the result of Lemma E.1 and E.2, and the bound for model difference $\Delta_t$ in partial participation settings in Section E.4, we can obtain the bound for $I_1'$. The bound for $I_2'$ $I_3'$ and $I_4'$ is obtained by the similar approach, and the derivation for them is similar to full participation settings, here we omit the derivation for bounding $I_2'$ $I_3'$ and $I_4'$.

Merging $I'_1$ to $I'_4$ together, we obtain the following result,

$$\mathbb{E}[f(\mathbf{z}_{T+1})] - f(\mathbf{z}_1)$$

$$\leq \frac{\eta\eta_l}{4C_0} \sum_{t=1}^{T} \left\{ -\tau\mathbb{E}[\|\nabla f(\mathbf{x}_t)\|^2] - \sum_{s=0}^{\tau-1} \mathbb{E}\left[\left\|\frac{1}{N}\sum_{i=1}^{N}\nabla f_i(\mathbf{x}_{t,s}^i)\right\|^2\right] - \sum_{s=0}^{\tau-1}\mathbb{E}\left[\left\|\frac{1}{K}\sum_{k=1}^{K}\nabla\bar{f}_k(\bar{\mathbf{x}}_{t,s}^k)\right\|^2\right] \right\}$$

$$+ \frac{\eta\eta_l}{4\sqrt{\epsilon}} \sum_{t=1}^{T} \left[ 3L^2\tau^2 C_1\eta_l^2(\tau + \rho_{\max}^2 D_{\tau,\rho})(\alpha^2\mathbb{E}\|\nabla f(\mathbf{x}_t)\|^2 + \sigma_g^2) + 3L^2\tau^2 C_1\rho_{\max}^2 D_{\tau,\rho}\eta_l^2\bar{\sigma}_L^2 \right.$$

$$\left. + 3L^2\tau^2 C_1\eta_l^2\sigma^2\rho_{\max}^2 + 3L^2 C_1(\tau^2 + D_{\tau,\rho}^2 \cdot \rho_{\max}^2)\eta_l^2\frac{\sigma^2}{n} \right]$$

$$+ \left( \eta\frac{\beta_1}{1-\beta_1}\eta_l\tau G^2 + \eta^2\frac{\beta_1^2}{(1-\beta_1)^2}L\eta_l^2\tau^2 G^2\epsilon^{-1/2} \right) \sum_{t=1}^{T}\mathbb{E}[\|\widehat{\mathbf{V}}_{t-1}^{-1/2} - \widehat{\mathbf{V}}_t^{-1/2}\|_1]$$

$$+ \eta^2 L\frac{\beta_1^2}{(1-\beta_1)^2}\eta_l^2\tau^2 G^2 \sum_{t=1}^{T}\mathbb{E}[\|\widehat{\mathbf{V}}_{t-1}^{-1/2} - \widehat{\mathbf{V}}_t^{-1/2}\|^2]$$

$$+ \left( \frac{\eta^2 L}{2\epsilon}\frac{\beta_1^2}{(1-\beta_1)^2} + \frac{\eta^2 L}{2\epsilon} + \frac{\eta^2 L}{\epsilon} + \frac{\eta\sqrt{1-\beta_2}G}{\epsilon} \right) \sum_{t=1}^{T}\mathbb{E}[\|\Delta_t\|^2],$$

then substituting the bound of $\Delta_t$ in Lemma E.6, we have

$$\mathbb{E}[f(\mathbf{z}_{T+1})] - f(\mathbf{z}_1)$$

$$\leq -\frac{\eta\eta_l\tau}{4C_0}\sum_{t=1}^{T}\mathbb{E}[\|\nabla f(\mathbf{x}_t)\|^2] - \frac{\eta\eta_l}{4C_0}\sum_{t=1}^{T}\sum_{s=0}^{\tau-1}\mathbb{E}\left[\left\|\frac{1}{N}\sum_{i=1}^{N}\nabla f_i(\mathbf{x}_{t,s}^i)\right\|^2\right]$$

$$- \frac{\eta\eta_l}{4C_0}\sum_{t=1}^{T}\sum_{s=0}^{\tau-1}\mathbb{E}\left[\left\|\frac{1}{K}\sum_{k=1}^{K}\nabla\bar{f}_k(\bar{\mathbf{x}}_{t,s}^k)\right\|^2\right]$$

$$+ \frac{\eta\eta_l}{4\sqrt{\epsilon}}\sum_{t=1}^{T}\left[ 3L^2\tau^2 C_1\eta_l^2(\tau + \rho_{\max}^2 D_{\tau,\rho})(\alpha^2\mathbb{E}\|\nabla f(\mathbf{x}_t)\|^2 + \sigma_g^2) + 3L^2\tau^2 C_1\rho_{\max}^2 D_{\tau,\rho}\eta_l^2\bar{\sigma}_L^2 \right.$$

$$\left. + 3L^2\tau^2 C_1\eta_l^2\sigma^2\rho_{\max}^2 + 3L^2 C_1(\tau^2 + D_{\tau,\rho}^2 \cdot \rho_{\max}^2)\eta_l^2\frac{\sigma^2}{n} \right]$$

$$+ \left( \eta\frac{\beta_1}{1-\beta_1}\eta_l\tau G^2 + \eta^2\frac{\beta_1^2}{(1-\beta_1)^2}L\eta_l^2\tau^2 G^2\epsilon^{-1/2} \right) \sum_{t=1}^{T}\mathbb{E}[\|\widehat{\mathbf{V}}_{t-1}^{-1/2} - \widehat{\mathbf{V}}_t^{-1/2}\|_1]$$

$$+ \eta^2 L\frac{\beta_1^2}{(1-\beta_1)^2}\eta_l^2\tau^2 G^2 \sum_{t=1}^{T}\mathbb{E}[\|\widehat{\mathbf{V}}_{t-1}^{-1/2} - \widehat{\mathbf{V}}_t^{-1/2}\|^2]$$

$$+ \left( \frac{\eta^2 L}{2\epsilon}\frac{\beta_1^2}{(1-\beta_1)^2} + \frac{\eta^2 L}{2\epsilon} + \frac{\eta^2 L}{\epsilon} + \frac{\eta\sqrt{1-\beta_2}G}{\epsilon} \right) \sum_{t=1}^{T}\left\{ \frac{2\eta_l^2\tau}{N}\sigma^2 \right.$$

$$+ 2\eta_l^2(\tau-1)\sum_{s=0}^{\tau-2}\mathbb{E}\left[\left\|\frac{1}{N}\sum_{i=1}^{N}\nabla f_i(\mathbf{x}_{t,s}^i)\right\|^2\right] + 4\eta_l^2\mathbb{E}\left[\left\|\frac{1}{K}\sum_{k=1}^{K}\nabla\bar{f}_k(\bar{\mathbf{x}}_{t,\tau-1}^k)\right\|^2\right]$$

$$\left. + 8\left( \frac{n-m}{m(n-1)} + \eta_l^2 L^2 \right)\left( \frac{1}{N}\sum_{k=1}^{K}\mathbb{E}[\|X_{t,\tau-1}^{k,\perp}\|^2] \right) + \frac{2\eta_l^2\sigma^2}{N}\left( \frac{n-m}{m} \cdot \rho_{\max}^2 \right) \right\}, \qquad \text{(D.2)}$$

we need the following constraint on local learning rate $\eta_l$

$$2C_{\beta,\eta}\eta\eta_l^2(\tau-1) \leq \frac{\eta\eta_l}{4C_0}, \quad 4C_{\beta,\eta}\eta\eta_l^2 \leq \frac{\eta\eta_l}{4C_0}$$

$$\Rightarrow \eta \leq \frac{1}{8C_0 C_{\beta,\eta}(\tau-1)}, \quad \eta \leq \frac{1}{16C_0 C_{\beta,\eta}}, \qquad \text{(D.3)}$$

where $C_{\beta,\eta} = \left(\frac{\eta L}{2\epsilon}\frac{\beta_1^2}{(1-\beta_1)^2} + \frac{\eta L}{2\epsilon} + \frac{\eta L}{\epsilon} + \frac{\sqrt{1-\beta_2}G}{\epsilon}\right) = \mathcal{O}(\max\{\eta, 1\})$, and by applying Lemma E.4, then we have

$$
\mathbb{E}[f(\mathbf{z}_{T+1})] - f(\mathbf{z}_1)
$$

$$
\leq -\frac{\eta\eta_l\tau}{4C_0}\sum_{t=1}^{T}\mathbb{E}[\|\nabla f(\mathbf{x}_t)\|^2] + \frac{\eta\eta_l}{4\sqrt{\epsilon}}3L^2\tau^2 C_1\eta_l^2(\tau + \rho_{\max}^2 D_{\tau,\rho})\alpha^2\sum_{t=1}^{T}\mathbb{E}[\|\nabla f(\mathbf{x}_t)\|^2]
$$

$$
+ \frac{\eta\eta_l}{4\sqrt{\epsilon}}\sum_{t=1}^{T}\Big\{3L^2\tau^2 C_1\eta_l^2(\tau + \rho_{\max}^2 D_{\tau,\rho})\sigma_g^2
$$

$$
+ 3L^2\tau^2 C_1\rho_{\max}^2 D_{\tau,\rho}\eta_l^2\bar\sigma_L^2 + 3L^2\tau^2 C_1\eta_l^2\sigma^2\rho_{\max}^2 + 3L^2 C_1\big(\tau^2 + D_{\tau,\rho}^2\cdot\rho_{\max}^2\big)\eta_l^2\frac{\sigma^2}{n}\Big\}
$$

$$
+ \eta\frac{\beta_1}{1-\beta_1}\eta_l\tau G^2\frac{d}{\sqrt{\epsilon}} + 2\eta^2 L\frac{\beta_1^2}{(1-\beta_1)^2}\eta_l^2\tau^2 G^2\cdot\frac{d}{\epsilon}
$$

$$
+ \left(\frac{\eta^2 L}{2\epsilon}\frac{\beta_1^2}{(1-\beta_1)^2} + \frac{\eta^2 L}{2\epsilon} + \frac{\eta^2 L}{\epsilon} + \frac{\eta\sqrt{1-\beta_2}G}{\epsilon}\right)\left[\frac{2T\eta_l^2\tau}{N}\sigma^2 + \frac{2T\eta_l^2\sigma^2}{N}\left(\frac{n-m}{m}\rho_{\max}^2\right)\right]
$$

$$
+ 8T\left(\frac{n-m}{m(n-1)} + \eta_l^2 L^2\right)\left(\frac{\eta^2 L}{2\epsilon}\frac{\beta_1^2}{(1-\beta_1)^2} + \frac{\eta^2 L}{2\epsilon} + \frac{\eta^2 L}{\epsilon} + \frac{\eta\sqrt{1-\beta_2}G}{\epsilon}\right)
$$

$$
\cdot\Big[\tau C_1\eta_l^2 D_{\tau,\rho}\rho_{\max}^2(\alpha^2\mathbb{E}\|\nabla f(\mathbf{x}_t)\|^2 + \sigma_g^2)
$$

$$
+ \tau C_1\eta_l^2 D_{\tau,\rho}\rho_{\max}^2\bar\sigma_L^2 + \tau C_1\eta_l^2\rho_{\max}^2\sigma^2 + C_1 D_{\tau,\rho}^2\tau^{-1}\rho_{\max}^2\eta_l^2\frac{\sigma^2}{n}\Big], \tag{D.4}
$$

we further need the requirement of $\eta_l$, which is same as the requirement in full participation settings

$$
\frac{\eta\eta_l}{4\sqrt{\epsilon}}3L^2\tau^2 C_1\eta_l^2(\tau + \rho_{\max}^2 D_{\tau,\rho})\alpha^2 + 8\left(\frac{n-m}{m(n-1)} + \eta_l^2 L^2\right)C_{\beta,\eta}\eta\tau C_1\eta_l^2 D_{\tau,\rho}\rho_{\max}^2\alpha^2 \leq \frac{\eta\eta_l\tau}{8C_0},
$$

$$
\Rightarrow \frac{1}{4\sqrt{\epsilon}}3L^2\tau C_1\eta_l^2(\tau + \rho_{\max}^2 D_{\tau,\rho})\alpha^2 + 8\left(\frac{n-m}{m(n-1)} + \frac{1}{\tau^2}\right)C_{\beta,\eta}C_1\eta_l D_{\tau,\rho}\rho_{\max}^2\alpha^2 \leq \frac{1}{8C_0},
$$

$$
\Rightarrow \eta_l \leq \frac{\sqrt[4]{\epsilon}}{\sqrt{12L^2 C_0 C_1\tau(\tau + \rho_{\max}^2 D_{\tau,\rho})\alpha^2}}, \eta_l \leq \frac{1}{128 C_0 C_{\beta,\eta}C_1 D_{\tau,\rho}\rho_{\max}^2\alpha^2}\left(\frac{n-m}{m(n-1)} + \frac{1}{\tau^2}\right)^{-1}
$$

$$
\tag{D.5}
$$

thus we have

$$
\frac{\eta\eta_l\tau}{8C_0 T}\sum_{t=1}^{T}\mathbb{E}[\|\nabla f(\mathbf{x}_t)\|^2]
$$

$$
\leq \frac{\mathbb{E}[f(\mathbf{z}_{T+1})] - f(\mathbf{z}_T)}{T} + \eta\frac{\beta_1}{1-\beta_1}\eta_l\tau G^2\frac{d}{T\sqrt{\epsilon}} + 2\eta^2 L\frac{\beta_1^2}{(1-\beta_1)^2}\eta_l^2\tau^2 G^2\cdot\frac{d}{T\epsilon}
$$

$$
+ \frac{\eta\eta_l}{4\sqrt{\epsilon}}\cdot\sum_{t=1}^{T}\Big\{3L^2\tau^2 C_1\eta_l^2(\tau + \rho_{\max}^2 D_{\tau,\rho})\sigma_g^2
$$

$$
+ 3L^2\tau^2 C_1\rho_{\max}^2 D_{\tau,\rho}\eta_l^2\bar\sigma_L^2 + 3L^2\tau^2 C_1\eta_l^2\sigma^2\rho_{\max}^2 + 3L^2 C_1\big(\tau^2 + D_{\tau,\rho}^2\cdot\rho_{\max}^2\big)\eta_l^2\frac{\sigma^2}{n}\Big)
$$

$$
+ \left(\frac{\eta^2 L}{2\epsilon}\frac{\beta_1^2}{(1-\beta_1)^2} + \frac{\eta^2 L}{2\epsilon} + \frac{\eta^2 L}{\epsilon} + \frac{\eta\sqrt{1-\beta_2}G}{\epsilon}\right)\left[\frac{2\eta_l^2\tau}{N}\sigma^2 + \frac{2\eta_l^2\sigma^2}{N}\left(\frac{n-m}{m}\rho_{\max}^2\right)\right]
$$

$$
+ 8\left(\frac{n-m}{m(n-1)} + \eta_l^2 L^2\right)\left(\frac{\eta^2 L}{2\epsilon}\frac{\beta_1^2}{(1-\beta_1)^2} + \frac{\eta^2 L}{2\epsilon} + \frac{\eta^2 L}{\epsilon} + \frac{\eta\sqrt{1-\beta_2}G}{\epsilon}\right)
$$

$$
\cdot\Big[\tau C_1\eta_l^2 D_{\tau,\rho}\rho_{\max}^2\sigma_g^2 + \tau C_1\eta_l^2 D_{\tau,\rho}\rho_{\max}^2\bar\sigma_L^2 + \tau C_1\eta_l^2\rho_{\max}^2\sigma^2 + C_1 D_{\tau,\rho}^2\tau^{-1}\rho_{\max}^2\eta_l^2\frac{\sigma^2}{n}\Big],
$$

$$
\tag{D.6}
$$

since there is $D_{\tau,\rho} \leq \tau$ and $\rho_{\max} \leq 1$, thus we have

$$\frac{1}{T}\sum_{t=1}^{T}\mathbb{E}[\|\nabla f(\mathbf{x}_t)\|^2]$$

$$\leq 8C_0\bigg\{\frac{\mathbb{E}[f(\mathbf{z}_{T+1})]-f(\mathbf{z}_T)}{\eta\eta_l\tau T} + \frac{1}{T}\bigg(\frac{C_\beta G^2 d}{\sqrt{\epsilon}} + \frac{2C_\beta^2\eta\eta_l\tau LG^2 d}{\epsilon}\bigg)$$

$$+ \frac{1}{4\sqrt{\epsilon}}\bigg[C\cdot L^2\tau(\tau+D_{\tau,\rho})\eta_l^2\sigma_g^2 + C\cdot L^2\eta_l^2\bigg(\tau\rho_{\max}^2 D_{\tau,\rho}\bar{\sigma}_L^2 + \tau\sigma^2\bigg(\frac{1}{n}+\rho_{\max}^2\bigg)\bigg)\bigg]$$

$$+ \bigg(\frac{\eta L}{\epsilon}\frac{\beta_1^2}{(1-\beta_1)^2} + \frac{3\eta L}{\epsilon} + \frac{2\sqrt{1-\beta_2}G}{\epsilon}\bigg)\bigg[\frac{\eta_l}{N}\sigma^2 + \frac{\eta_l\sigma^2}{N}\bigg(\frac{n-m}{m}\rho_{\max}^2\bigg)\bigg]$$

$$+ 4\bigg(\frac{n-m}{m(n-1)}+\eta_l^2 L^2\bigg)\bigg(\frac{\eta L}{\epsilon}\frac{\beta_1^2}{(1-\beta_1)^2} + \frac{3\eta L}{\epsilon} + \frac{2\sqrt{1-\beta_2}G}{\epsilon}\bigg)$$

$$\cdot\bigg[C_1\eta_l D_{\tau,\rho}\rho_{\max}^2\sigma_g^2 + C_1\eta_l D_{\tau,\rho}\rho_{\max}^2\bar{\sigma}_L^2 + C_1\eta_l\rho_{\max}^2\sigma^2 + C_1 D_{\tau,\rho}^2\tau^{-2}\rho_{\max}^2\eta_l\frac{\sigma^2}{n}\bigg]\bigg\}, \quad \text{(D.7)}$$

where $C$ is a constant irrelevant to parameters and $\rho_{\max} = \max_{k\in[K]}\rho_k$, $D_{\tau,\rho} = \min\big\{\frac{1}{1-\rho_{\max}},\tau\big\}$, $C_\beta = \frac{\beta_1}{1-\beta_1}$ and $\bar{\sigma}_L^2 = \frac{1}{K}\sum_{k=1}^{K}\sigma_k^2$. This concludes the proof. $\qquad\square$

*Proof of Corollary 5.11.* Further apply the constraint of

$$\frac{n-m}{m(n-1)}D_{\tau,\rho}\rho_{\max}^2 \leq \frac{1}{N}, \tag{D.8}$$

where the condition Eq. D.8 implies that the spectral gap $\rho_{\max}$ satisfies

$$\rho_{\max}^2 \leq \frac{1}{4(n-m)}. \tag{D.9}$$

With the condition of Eq. D.9, there is $\rho_{\max} \leq \frac{1}{2}$ hence when $\tau \geq 2$, there is $\frac{1}{1-\rho_{\max}} \leq \tau$, also assume $K \leq n-1$,

$$\frac{n-m}{m(n-1)}D_{\tau,\rho}\rho_{\max}^2 = \frac{n-m}{m(n-1)}\frac{1}{1-\rho_{\max}}\rho_{\max}^2$$

$$\leq 2\frac{n-m}{m(n-1)}\rho_{\max}^2$$

$$\leq 2\frac{n-m}{m(n-1)}\frac{1}{4(n-m)}$$

$$\leq \frac{1}{M}. \tag{D.10}$$

Also by choosing a constant $\widetilde{C}$, we have

$$\frac{1}{T}\sum_{t=1}^{T}\mathbb{E}[\|\nabla f(\mathbf{x}_t)\|^2]$$

$$\leq 8C_0\bigg\{\frac{\mathbb{E}[f(\mathbf{z}_{T+1})]-f(\mathbf{z}_T)}{\eta\eta_l\tau T} + \frac{1}{T}\bigg(\frac{C_\beta G^2 d}{\sqrt{\epsilon}} + \frac{2C_\beta^2\eta\eta_l\tau LG^2 d}{\epsilon}\bigg)$$

$$+ \frac{CL^2\eta_l^2}{4\sqrt{\epsilon}}\bigg[\tau(\tau+D_{\tau,\rho})\sigma_g^2 + \bigg(\tau\rho_{\max}^2 D_{\tau,\rho}\bar{\sigma}_L^2 + \tau\sigma^2\bigg(\frac{1}{n}+\rho_{\max}^2\bigg)\bigg)\bigg]$$

$$+ \bigg(\frac{\eta L}{\epsilon}\frac{\beta_1^2}{(1-\beta_1)^2} + \frac{3\eta L}{\epsilon} + \frac{2\sqrt{1-\beta_2}G}{\epsilon}\bigg)\bigg[\frac{\eta_l}{N}\sigma^2 + \frac{\eta_l\sigma^2}{N}\bigg(\frac{n-m}{m}\rho_{\max}^2\bigg)\bigg]$$

$$+ \frac{1}{N}\bigg(\frac{\eta L}{\epsilon}\frac{\beta_1^2}{(1-\beta_1)^2} + \frac{3\eta L}{\epsilon} + \frac{2\sqrt{1-\beta_2}G}{\epsilon}\bigg)\widetilde{C}\eta_l\bigg[\sigma_g^2 + \bar{\sigma}_L^2 + \sigma^2 + D_{\tau,\rho}\frac{\sigma^2}{\tau^2 n}\bigg]$$

$$+ \widetilde{C}L^2 D_{\tau,\rho}\rho_{\max}^2\bigg(\frac{\eta L}{\epsilon}\frac{\beta_1^2}{(1-\beta_1)^2} + \frac{3\eta L}{\epsilon} + \frac{2\sqrt{1-\beta_2}G}{\epsilon}\bigg)\eta_l^3\bigg[\sigma_g^2 + \bar{\sigma}_L^2 + \sigma^2 + D_{\tau,\rho}\frac{\sigma^2}{\tau^2 n}\bigg]\bigg\}$$

$$\text{(D.11)}$$

By adopting learning rates $\eta = \Theta(\sqrt{\tau M})$, $\eta_l = \Theta\left(\frac{1}{\sqrt{T}\tau}\right)$ then we concludes the proof. $\qquad\square$

# E   Supporting Lemmas

### E.1   Lemma for inter-cluster consensus error

**Lemma E.1.** For local learning rate which satisfying the condition $\eta_l \leq \frac{1}{8\tau L}$, denote $C_\tau = 1 + \frac{3}{2} \cdot \frac{1}{4\tau - 1}$, recall the definition for $\bar{\mathbf{x}}$ in Eq. B.1, the inter-cluster model difference after $s$ local steps satisfies

$$
\frac{1}{K}\sum_{k=1}^{K}\mathbb{E}\|\bar{\mathbf{x}}_{t,s+1}^k - \mathbf{x}_t\|^2
$$

$$
\leq C_\tau \frac{1}{K}\sum_{k=1}^{K}\mathbb{E}\|\bar{\mathbf{x}}_{t,s}^k - \mathbf{x}_t\|^2 + 8\tau\eta_l^2(\alpha^2\mathbb{E}[\|\nabla f(\mathbf{x}_t)\|^2] + \sigma_g^2) + \eta_l^2\frac{\sigma^2}{n}. \tag{E.1}
$$

*Proof.* Note that the following proof is similar to Lemma 3 in [30].

$$
\begin{aligned}
\mathbb{E}\|\bar{\mathbf{x}}_{t,s+1}^k - \mathbf{x}_t\|^2 &= \mathbb{E}\|\bar{\mathbf{x}}_{t,s}^k - \mathbf{x}_t - \eta_l\bar{\mathbf{g}}_{t,s}^k\|^2 \\
&= \mathbb{E}\|\bar{\mathbf{x}}_{t,s}^k - \mathbf{x}_t - \eta_l(\bar{\mathbf{g}}_{t,s}^k - \nabla\bar{f}_k(\bar{\mathbf{x}}_{t,s}^k) + \nabla\bar{f}_k(\bar{\mathbf{x}}_{t,s}^k) - \nabla\bar{f}_k(\mathbf{x}_t) + \nabla\bar{f}_k(\mathbf{x}_t))\|^2 \\
&\leq (1+\gamma)\mathbb{E}\|\bar{\mathbf{x}}_{t,s}^k - \mathbf{x}_t\|^2 + \eta_l^2\mathbb{E}\|\bar{\mathbf{g}}_{t,s}^k - \nabla\bar{f}_k(\bar{\mathbf{x}}_{t,s}^k)\|^2 \\
&\quad + 2(1+\gamma^{-1})\eta_l^2\mathbb{E}[\|\nabla\bar{f}_k(\bar{\mathbf{x}}_{t,s}^k) - \nabla\bar{f}_k(\mathbf{x}_t)\|^2] + 2(1+\gamma^{-1})\eta_l^2\mathbb{E}[\|\nabla\bar{f}_k(\mathbf{x}_t)\|^2] \\
&\leq (1+\gamma)\mathbb{E}\|\bar{\mathbf{x}}_{t,s}^k - \mathbf{x}_t\|^2 + \eta_l^2\frac{\sigma^2}{n} + 2(1+\gamma^{-1})\eta_l^2 L^2\mathbb{E}\|\bar{\mathbf{x}}_{t,s}^k - \mathbf{x}_t\|^2 + 2(1+\gamma^{-1})\eta_l^2\mathbb{E}[\|\nabla\bar{f}_k(\mathbf{x}_t)\|^2] \\
&\leq [(1+\gamma) + 2(1+\gamma^{-1})\eta_l^2 L^2] \cdot \mathbb{E}\|\bar{\mathbf{x}}_{t,s}^k - \mathbf{x}_t\|^2 + \eta_l^2\frac{\sigma^2}{n} + 2(1+\gamma^{-1})\eta_l^2\mathbb{E}[\|\nabla\bar{f}_k(\mathbf{x}_t)\|^2], \quad \text{(E.2)}
\end{aligned}
$$

where the first equality holds by Eq. B.2. The first inequality holds due to $\mathbf{g}_{t,s}^i$ is an unbiased estimator of $\nabla f_i(\mathbf{x}_{t,s}^i)$ and Young's inequality. The second inequality holds by Assumption 5.1 and 5.3, also the independency with $\mathbf{g}_{t,s}^i$ and $\mathbf{g}_{t,s}^j$ for $i \neq j$.

Averaging Eq. E.2 over $k = 1, ..., K$ clusters, we have

$$
\frac{1}{K}\sum_{k=1}^{K}\mathbb{E}\|\bar{\mathbf{x}}_{t,s+1}^k - \mathbf{x}_t\|^2
$$

$$
\leq [(1+\gamma) + 2(1+\gamma^{-1})\eta_l^2 L^2]\frac{1}{K}\sum_{k=1}^{K}\mathbb{E}\|\bar{\mathbf{x}}_{t,s}^k - \mathbf{x}_t\|^2
$$

$$
+ 2(1+\gamma^{-1})\eta_l^2\frac{1}{K}\sum_{k=1}^{K}\mathbb{E}[\|\nabla\bar{f}_k(\mathbf{x}_t)\|^2] + \eta_l^2\frac{\sigma^2}{n}
$$

$$
\leq [(1+\gamma) + 2(1+\gamma^{-1})\eta_l^2 L^2]\frac{1}{K}\sum_{k=1}^{K}\mathbb{E}\|\bar{\mathbf{x}}_{t,s}^k - \mathbf{x}_t\|^2
$$

$$
+ 2(1+\gamma^{-1})\eta_l^2(\alpha^2\mathbb{E}[\|\nabla f(\mathbf{x}_t)\|^2] + \sigma_g^2) + \eta_l^2\frac{\sigma^2}{n}, \tag{E.3}
$$

where the second inequality holds by Assumption 5.4. Choosing $\gamma = \frac{1}{4\tau-1}$ with the condition of $\eta_l \le \frac{1}{8\tau L}$, we have

$$\frac{1}{K}\sum_{k=1}^{K}\mathbb{E}\|\bar{\mathbf{x}}_{t,s+1}^k - \mathbf{x}_t\|^2$$

$$\le \left(1 + \frac{1}{4\tau-1} + \frac{1}{2(4\tau-1)}\right)\frac{1}{K}\sum_{k=1}^{K}\mathbb{E}\|\bar{\mathbf{x}}_{t,s}^k - \mathbf{x}_t\|^2 + 8\tau\eta_l^2(\alpha^2\mathbb{E}[\|\nabla f(\mathbf{x}_t)\|^2] + \sigma_g^2) + \eta_l^2\frac{\sigma^2}{n}$$

$$= C_\tau\frac{1}{K}\sum_{k=1}^{K}\mathbb{E}\|\bar{\mathbf{x}}_{t,s}^k - \mathbf{x}_t\|^2 + 8\tau\eta_l^2(\alpha^2\mathbb{E}[\|\nabla f(\mathbf{x}_t)\|^2] + \sigma_g^2) + \eta_l^2\frac{\sigma^2}{n}, \tag{E.4}$$

where $C_\tau = 1 + \frac{3}{2} \cdot \frac{1}{4\tau-1}$. This concludes the proof. $\qquad\square$

### E.2 Lemma for intra-cluster consensus error

**Lemma E.2.** The intra-cluster consensus error $\sum_{i=1}^{n}\|\bar{\mathbf{x}}_{t,s}^k - \mathbf{x}_{t,s}^i\|^2$, also known as $\|X_{t,s}^{k,\perp}\|_F^2$, has the following upper bound,

$$\frac{1}{N}\mathbb{E}\sum_{k=1}^{K}\|X_{t,s+1}^{k,\perp}\|^2$$

$$\le \left(\max_{k\in[K]}\rho_k^2(1+\zeta_k^{-1}) + \eta_l^2 \cdot 4L^2\max_{k\in[K]}\{\rho_k^2(1+\zeta_k)\}\right)\frac{1}{N}\sum_{k=1}^{K}\mathbb{E}\|X_{t,s}^{k,\perp}\|^2$$

$$+ \eta_l^2\max_{k\in[K]}\{\rho_k^2(1+\zeta_k)\} \cdot 4L^2\mathbb{E}\|\bar{\mathbf{x}}_{t,s}^k - \mathbf{x}_t\|^2 + \eta_l^2\max_{k\in[K]}\{\rho_k^2(1+\zeta_k)\} \cdot 4(\alpha^2\mathbb{E}\|\nabla f(\mathbf{x}_t)\|^2 + \sigma_g^2)$$

$$+ \eta_l^2\frac{1}{K}\sum_{k=1}^{K}\rho_k^2(1+\zeta_k)4\sigma_k^2 + \eta_l^2\sigma^2\rho_{\max}^2, \tag{E.5}$$

where $\zeta_k$ is some constant related to the Young's inequality, and it could be uniformly chosen for all $k = 1, ..., K$.

*Proof.* By definition we have $X_k = (\mathbf{x}_{t,s}^1, ..., \mathbf{x}_{t,s}^n)'$ and $X_{t,s}^{k,\perp} = X_{t,s}^k(I_n - J)$, where $J = \frac{1}{n}\mathbf{1}_n \cdot \mathbf{1}_n'$. Thus we have

$$\sum_{i=1}^{n}\|\bar{\mathbf{x}}_{t,s}^k - \mathbf{x}_{t,s}^i\|^2 = \|(\mathbf{x}_{t,s}^1, ..., \mathbf{x}_{t,s}^n)(I_n - J) \cdot I_n \cdot (I_n - J)(\mathbf{x}_{t,s}^1, ..., \mathbf{x}_{t,s}^n)'\|_F$$

$$= \|X_{t,s}^k{}'(I_n - J) \cdot (I_n - J)X_{t,s}^k\|_F$$

$$= \|X_{t,s}^{k,\perp}{}' \cdot X_{t,s}^{k,\perp}\|_F$$

$$= \|X_{t,s}^{k,\perp}\|_F^2, \tag{E.6}$$

Recall the update rule of HA-Fed, there is $X_{t,s+1}^{k,\perp} = (W_k - J)(X_{t,s}^{k,\perp} - \eta_l G_{t,s}^k)$, then we have

$$\mathbb{E}\|X_{t,s+1}^{k,\perp}\|^2 = \mathbb{E}(\mathbb{E}(\|(W_k - J)(X_{t,s}^{k,\perp} - \eta_l G_{t,s}^k)\|^2|\mathcal{F}_{t,s-1}))$$

$$= \mathbb{E}(\mathbb{E}(\|(W_k - J)(X_{t,s}^{k,\perp} - \eta_l\nabla F(X_{t,s}^k) + \eta_l\nabla F(X_{t,s}^k) - \eta_l G_{t,s}^k)\|^2|\mathcal{F}_{t,s-1}))$$

$$= \mathbb{E}(\mathbb{E}(\|(W_k - J)(X_{t,s}^{k,\perp} - \eta_l\nabla F(X_{t,s}^k))\|^2|\mathcal{F}_{t,s-1}))$$

$$\quad + \eta_l^2\mathbb{E}(\mathbb{E}(\|(W_k - J)(\nabla F(X_{t,s}^k) - G_{t,s}^k)\|^2|\mathcal{F}_{t,s-1}))$$

$$\le \mathbb{E}(\|(W_k - J)(X_{t,s}^{k,\perp} - \eta_l\nabla F(X_{t,s}^k))\|^2) + \eta_l^2\rho_k^2n\sigma^2$$

$$\le \rho_k^2(1+\zeta_k^{-1}) \cdot \mathbb{E}\|X_{t,s}^{k,\perp}\|^2 + \rho_k^2(1+\zeta_k)\eta_l^2\mathbb{E}\|\nabla F(X_{t,s}^k)\|^2 + \eta_l^2\rho_k^2n\sigma^2, \tag{E.7}$$

where the $\nabla F_k(X^k) \in \mathbb{R}^{n\times d}$ is associated to cluster $k$ by stacking $\nabla f_i(\mathbf{x}^i)$ for $i \in \mathcal{V}_k$ row-wise. The third equality is due to the unbiasedness of stochastic gradient. The first inequality holds by

Assumption 5.3 and $\|\nabla F(X_{t,s}^k) - G_{t,s}^k\|_F = \sum_{i=1}^n \|\nabla f_i(\mathbf{x}_{t,s}^i) - \mathbf{g}_{t,s}^i\|^2$. For the Frobenius norm, there is $\|AB\|_F \leq \|A\|_2 \|B\|_F$. The second inequality holds by Young's inequality with some parameter $\zeta_k > 0$ and $\|AB\|_F \leq \|A\|_2 \|B\|_F$ as well. For $\nabla F_k(X_{t,s}^k)$, by definition, we have

$$\|\nabla F_k(X_{t,s}^k)\|_F^2 = \sum_{i \in \mathcal{V}_k} \|\nabla f_i(\mathbf{x}_{t,s}^i)\|^2$$

$$= \sum_{i \in \mathcal{V}_k} \|\nabla f_i(\mathbf{x}_{t,s}^i) - \nabla f_i(\bar{\mathbf{x}}_{t,s}^k) + \nabla f_i(\bar{\mathbf{x}}_{t,s}^k) - \nabla \bar{f}_k(\bar{\mathbf{x}}_{t,s}^k) + \nabla \bar{f}_k(\bar{\mathbf{x}}_{t,s}^k) - \nabla \bar{f}_k(\mathbf{x}_t) + \nabla \bar{f}_k(\mathbf{x}_t)\|^2$$

$$\leq \sum_{i \in \mathcal{V}_k} \Big[ 4\|\nabla f_i(\mathbf{x}_{t,s}^i) - \nabla f_i(\bar{\mathbf{x}}_{t,s}^k)\|^2 + 4\|\nabla f_i(\bar{\mathbf{x}}_{t,s}^k) - \nabla \bar{f}_k(\bar{\mathbf{x}}_{t,s}^k)\|^2 + 4\|\nabla \bar{f}_k(\bar{\mathbf{x}}_{t,s}^k) - \nabla \bar{f}_k(\mathbf{x}_t)\|^2$$

$$+ 4\|\nabla \bar{f}_k(\mathbf{x}_t)\|^2 \Big]$$

$$\leq \sum_{i \in \mathcal{V}_k} \Big[ 4\|\nabla f_i(\bar{\mathbf{x}}_{t,s}^k) - \nabla \bar{f}_k(\bar{\mathbf{x}}_{t,s}^k)\|^2 + 4L^2 \|\mathbf{x}_{t,s}^i - \bar{\mathbf{x}}_{t,s}^k\|^2 + 4L^2 \|\bar{\mathbf{x}}_{t,s}^k - \mathbf{x}_t\|^2 + 4\|\nabla \bar{f}_k(\mathbf{x}_t)\|^2 \Big]$$

$$\leq 4L^2 \|X_{t,s}^{k,\perp}\|^2 + 4L^2 n \|\bar{\mathbf{x}}_{t,s}^k - \mathbf{x}_t\|^2 + 4n \|\nabla \bar{f}_k(\mathbf{x}_t)\|^2 + 4n\sigma_k^2, \tag{E.8}$$

where the first inequality holds by Cauchy inequality, and the last inequality holds by Assumption 5.4. Averaging Eq. E.8 over $k = 1, ..., K$, we have the following iteration

$$\frac{1}{N} \sum_{k=1}^K \mathbb{E}\|X_{t,s+1}^{k,\perp}\|^2$$

$$\leq \frac{1}{N} \sum_{k=1}^K \rho_k^2(1 + \zeta_k^{-1}) \cdot \mathbb{E}\|X_{t,s}^{k,\perp}\|^2 + \frac{1}{N} \sum_{k=1}^K \rho_k^2(1 + \zeta_k)\eta_l^2 \mathbb{E}\|\nabla F(X_{t,s}^k)\|^2 + \eta_l^2 \sigma^2 \frac{1}{K} \sum_{k=1}^K \rho_k^2$$

$$\leq \frac{1}{N} \sum_{k=1}^K \rho_k^2(1 + \zeta_k^{-1}) \cdot \mathbb{E}\|X_{t,s}^{k,\perp}\|^2 + \eta_l^2 \frac{1}{N} \sum_{k=1}^K \rho_k^2(1 + \zeta_k) \cdot 4L^2 \mathbb{E}\|X_{t,s}^{k,\perp}\|^2$$

$$+ \eta_l^2 \frac{1}{K} \sum_{k=1}^K \rho_k^2(1 + \zeta_k) \cdot 4L^2 \mathbb{E}\|\bar{\mathbf{x}}_{t,s}^k - \mathbf{x}_t\|^2 + \eta_l^2 \frac{1}{K} \sum_{k=1}^K \rho_k^2(1 + \zeta_k) \cdot 4\mathbb{E}\|\nabla \bar{f}_k(\mathbf{x}_t)\|^2$$

$$+ \eta_l^2 \frac{1}{K} \sum_{k=1}^K \rho_k^2(1 + \zeta_k) \cdot 4\sigma_k^2 + \eta_l^2 \sigma^2 \rho_{\max}^2$$

$$\leq \left( \max_{k \in [K]} \rho_k^2(1 + \zeta_k^{-1}) + \eta_l^2 \cdot 4L^2 \max_{k \in [K]} \{\rho_k^2(1 + \zeta_k)\} \right) \frac{1}{N} \sum_{k=1}^K \mathbb{E}\|X_{t,s}^{k,\perp}\|^2$$

$$+ \eta_l^2 \max_{k \in [K]} \{\rho_k^2(1 + \zeta_k)\} \cdot 4L^2 \mathbb{E}\|\bar{\mathbf{x}}_{t,s}^k - \mathbf{x}_t\|^2 + \eta_l^2 \max_{k \in [K]} \{\rho_k^2(1 + \zeta_k)\} \cdot 4(\alpha^2 \mathbb{E}\|\nabla f(\mathbf{x}_t)\|^2 + \sigma_g^2)$$

$$+ \eta_l^2 \frac{1}{K} \sum_{k=1}^K \rho_k^2(1 + \zeta_k)4\sigma_k^2 + \eta_l^2 \sigma^2 \rho_{\max}^2. \tag{E.9}$$

This concludes the proof. □

## E.3 Lemma for summation of intra-cluster and inter-cluster consensus errors

**Lemma E.3.** If the local learning rate satisfies the condition: $\eta_l \leq \frac{1}{8\tau L}$, the for all local round $s = 0, ..., \tau - 1$, there is

$$\frac{1}{N} \sum_{k=1}^K \mathbb{E}\|X_{t,s}^{k,\perp}\|^2 + \frac{1}{K} \sum_{k=1}^K \mathbb{E}\|\bar{\mathbf{x}}_{t,s}^k - \mathbf{x}_t\|^2$$

$$\leq (s+1)C_1 \eta_l^2(\tau + \rho_{\max}^2 D_{\tau,\rho})(\alpha^2 \mathbb{E}\|\nabla f(\mathbf{x}_t)\|^2 + \sigma_g^2) + (s+1)C_1 \rho_{\max}^2 D_{\tau,\rho} \eta_l^2 \bar{\sigma}_L^2$$

$$+ (s+1)C_1 \eta_l^2 \sigma^2 \rho_{\max}^2 + (s+1)C_1 \left( 1 + \frac{D_{\tau,\rho}^2}{\tau^2} \cdot \rho_{\max}^2 \right) \eta_l^2 \frac{\sigma^2}{n}, \tag{E.10}$$

where $C_1$ is a constant independent to parameters.

*Proof.* Denote an auxiliary vector

$$M_{t,s} = \left( \frac{1}{N} \sum_{k=1}^{K} \mathbb{E}\|X_{t,s}^{k,\perp}\|^2, \quad \frac{1}{K} \sum_{k=1}^{K} \mathbb{E}\|\bar{\mathbf{x}}_{t,s}^k - \mathbf{x}_t\|^2 \right)^T. \tag{E.11}$$

From Lemma E.1 and E.2, we have the following inequality which is defined element-wise for $s = 0, ..., \tau - 1$

$$M_{t,s+1} \leq G \cdot M_{t,s} + B_{t,s}, \tag{E.12}$$

where

$$G = \begin{pmatrix} \max_{k \in [K]} \rho_k^2 (1 + \zeta_k^{-1}) + \eta_l^2 \rho_L \cdot 4L^2 & \eta_l^2 \rho_L \cdot 4L^2 \\ 0 & C_\tau \end{pmatrix} \tag{E.13}$$

$$B_{t,s} = \begin{pmatrix} 4\rho_L \eta_l^2 (\alpha^2 \mathbb{E}\|\nabla f(\mathbf{x}_t)\|^2 + \sigma_g^2) + 4\rho_L \eta_l^2 \bar{\sigma}_L^2 + \eta_l^2 \sigma^2 \rho_{\max}^2 \\ 8\tau \eta_l^2 (\alpha^2 \mathbb{E}\|\nabla f(\mathbf{x}_t)\|^2 + \sigma_g^2) + \eta_l^2 \frac{\sigma^2}{n} \end{pmatrix} = \begin{pmatrix} b^{(1)} \\ b^{(2)} \end{pmatrix}. \tag{E.14}$$

Consider the eigen-decomposition of matrix $G$,

$$G = \begin{pmatrix} 1 & -\frac{4\eta_l^2 \rho_L L^2}{\lambda_1 - \lambda_2} \\ 0 & 1 \end{pmatrix} \cdot \begin{pmatrix} \lambda_1 & 0 \\ 0 & \lambda_2 \end{pmatrix} \cdot \begin{pmatrix} 1 & \frac{4\eta_l^2 \rho_L L^2}{\lambda_1 - \lambda_2} \\ 0 & 1 \end{pmatrix}, \tag{E.15}$$

where we assume $\lambda_1 \leq \lambda_2$, thus we have

$$G^j B = \begin{pmatrix} 1 & -\frac{4\eta_l^2 \rho_L L^2}{\lambda_1 - \lambda_2} \\ 0 & 1 \end{pmatrix} \begin{pmatrix} \lambda_1^j & 0 \\ 0 & \lambda_2^j \end{pmatrix} \begin{pmatrix} 1 & \frac{4\eta_l^2 \rho_L L^2}{\lambda_1 - \lambda_2} \\ 0 & 1 \end{pmatrix} \begin{pmatrix} b^{(1)} \\ b^{(2)} \end{pmatrix}$$

$$= \begin{pmatrix} \lambda_1^j (b^{(1)} + \frac{4\eta_l^2 \rho_L L^2}{\lambda_1 - \lambda_2} b^{(2)}) - \frac{4\eta_l^2 \rho_L L^2}{\lambda_1 - \lambda_2} \lambda_2^j b^{(2)} \\ \lambda_2^j b^{(2)} \end{pmatrix}. \tag{E.16}$$

Therefore the sum of two elements has the following result

$$(1,1)G^j B_{t,s-j} = \lambda_1^j b^{(1)} + \lambda_2^j b^{(2)} + \frac{\lambda_2^j - \lambda_1^j}{\lambda_2 - \lambda_1} 4\eta_l^2 \rho_L L^2 b^{(2)}$$

$$\leq \lambda_2^j (b^{(1)} + b^{(2)}) + \frac{\lambda_2^j - \lambda_1^j}{\lambda_2 - \lambda_1} 4\eta_l^2 \rho_L L^2 b^{(2)}. \tag{E.17}$$

Therefore, we have the following result

$$\sum_{j=0}^{s} (1,1)G^j B_{t,s-j} \leq \sum_{j=0}^{s} \left( \lambda_2^j (b_{t,s-j}^{(1)} + b_{t,s-j}^{(2)}) + \frac{\lambda_2^j - \lambda_1^j}{\lambda_2 - \lambda_1} \eta_l^2 \cdot 4\rho_L L^2 b_{t,s-j}^{(2)} \right). \tag{E.18}$$

Since $\lambda_2 \geq C_\tau > 1$, we have

$$\frac{\lambda_2^j - \lambda_1^j}{\lambda_2 - \lambda_1} = \lambda_2^{l-1} \sum_{s=0}^{l-1} \left( \frac{\lambda_1}{\lambda_2} \right)^s \leq \lambda_2^{j-1} \min \left\{ \frac{\lambda_2}{\lambda_2 - \lambda_1}, l \right\} \leq \lambda_2^j \min \left\{ \frac{1}{\lambda_2 - \lambda_1}, l \right\}, \tag{E.19}$$

thus we have

$$\sum_{j=0}^{s} (1,1)G^j B_{t,s-j} \leq \sum_{j=0}^{s} \lambda_2^j (b_{t,s-j}^{(1)} + b_{t,s-j}^{(2)}) + \sum_{l=0}^{s} \left( \lambda_2^j \min \left\{ \frac{1}{\lambda_2 - \lambda_1}, l \right\} \eta_l^2 \cdot 4\rho_L L^2 b_{t,s-j}^{(2)} \right). \tag{E.20}$$

By the definition of $\rho_L = \max_{k \in [k]} \rho_k^2 (1 + \zeta_k)$ and by the Gershgorin's theorem, since $\eta_l > 0$, we have the upper bound for $\lambda_2$,

$$\lambda_2 \leq \max \left\{ \max_{k \in [K]} \rho_k^2 (1 + \zeta_k^{-1}) + \eta_l^2 \rho_L \cdot 8L^2, C_\tau \right\}$$

$$< \max \left\{ \max_{k \in [K]} \rho_k^2 (1 + \zeta_k^{-1}) + \frac{\rho_L}{(4\tau - 1)2\tau}, 1 + \frac{3}{2(4\tau - 1)} \right\}, \tag{E.21}$$

where the last inequality holds by the bound of $\eta_l^2 \leq \frac{1}{64\tau^2 L^2} < \frac{1}{16\tau(4\tau-1)L^2}$. Define a distance constant $D_{\tau,\rho} = \min\left\{\tau, \frac{1}{1-\rho_{\max}}\right\}$. Next we consider two cases: *small or dense* communication network with $\rho_{\max} \leq 1 - \frac{1}{\tau}$ and *large and sparse* communication network with $\rho_{\max} > 1 - \frac{1}{\tau}$.

**Case 1:** For $\rho_{\max} \leq 1 - \frac{1}{\tau}$, i.e., $\frac{1}{1-\rho_{\max}} \leq \tau$, thus we have $D_{\tau,\rho} = \frac{1}{1-\rho_{\max}}$. Let $\zeta_k = \frac{\rho_k}{1-\rho_k}$, then we have

$$\max_{k \in [k]} \rho_k^2(1 + \zeta_k^{-1}) = \rho_{\max}, \quad \rho_L = \max_{k \in [k]}\left\{\frac{\rho_k^2}{1-\rho_k}\right\} = \frac{\rho_{\max}^2}{1-\rho_{\max}} = \rho_{\max}^2 D_{\tau,\rho}, \tag{E.22}$$

where the middle part of the second equality holds by the monotonically increasing of $\frac{x^2}{1-x}$. Then the bound for $\lambda_2$ is formalized as

$$\lambda_2 \leq \max\left\{\rho_{\max} + \frac{\rho_{\max}^2}{(1-\rho_{\max})2\tau(4\tau-1)}, 1 + \frac{3}{2(4\tau-1)}\right\}$$

$$\leq \max\left\{1 - \frac{1}{\tau} + \frac{(1-\frac{1}{\tau})^2}{2(\tau-1)}, 1 + \frac{3}{2(4\tau-1)}\right\}$$

$$< 1 + \frac{2}{4\tau-1}, \tag{E.23}$$

where the second inequality holds by $\rho_{\max} \leq 1 - \frac{1}{\tau}$. Then by $s \leq \tau$ and $\lambda_2 \geq 1$ (just by the definition of matrix $G$ can get this result), we can obtain the following bound

$$\sum_{j=0}^{s} \lambda_2^j b_{s-j}^1 \leq \left(\left(1 + \frac{2}{4\tau-1}\right)^\tau\right) \cdot \sum_{j=0}^{s} b_j^{(1)} \leq 3 \cdot \sum_{j=0}^{s} b_j^{(1)}. \tag{E.24}$$

We also have

$$\rho_{\max} + \eta_l^2 \rho_L 4L^2 \leq \rho_{\max} + \frac{\rho_{\max}}{(1-\rho_{\max})(4\tau-1)4\tau} \leq 1 - \frac{1}{\tau} + \frac{(1-\frac{1}{\tau})^2}{4(4\tau-1)} \leq C_\tau, \tag{E.25}$$

where the second inequality holds by the upper bound for $\rho_{\max}$. By the definition of matrix $G$, we bound the difference of $\lambda_2 - \lambda_1$,

$$\lambda_2 - \lambda_1 = C_\tau - \rho_{\max} - \eta_l^2 \rho_L 4L^2$$

$$\geq C_\tau - \left(\rho_{\max} + \frac{\rho_{\max}}{(1-\rho_{\max})(4\tau-1)4\tau}\right)$$

$$\geq C_\tau - \left(\rho_{\max} + \rho_{\max} \cdot \frac{1-\frac{1}{\tau}}{4(4\tau-1)}\right)$$

$$\geq 1 + \frac{1}{4\tau-1} - \left(\rho_{\max} + \rho_{\max} \cdot \frac{1}{4\tau-1}\right)$$

$$= (1-\rho_{\max})\left(1 + \frac{1}{4\tau-1}\right)$$

$$\geq 1 - \rho_{\max}. \tag{E.26}$$

where the first and second inequality hold by the defined notations. Then we have

$$\sum_{j=0}^{s}(1,1)G^j B_{t,s-j} \leq \sum_{j=0}^{s} \lambda_2^j(b_{t,s-j}^{(1)} + b_{t,s-j}^{(2)}) + \sum_{j=0}^{s}\left(\lambda_2^j \min\left\{\frac{1}{\lambda_2-\lambda_1}, j\right\}\eta_l^2 \cdot 4\rho_L L^2 b_{t,s-j}^2\right)$$

$$\leq \sum_{j=0}^{s} 3(b_{t,j}^{(1)} + b_{t,j}^{(2)}) + \sum_{j=0}^{s} 3\eta_l^2 \cdot 4\rho_L L^2 b_{t,j}^{(2)}\left(\min\left\{\frac{1}{\lambda_2-\lambda_1}, \tau\right\}\right)$$

$$\leq \sum_{j=0}^{s} 3(b_{t,j}^{(1)} + b_{t,j}^{(2)}) + \sum_{j=0}^{s} 3\eta_l^2 \cdot D_{\tau,\rho}^2 \rho_{\max}^2 4L^2 b_{t,j}^{(2)}$$

$$\leq \sum_{j=0}^{s} 3(b_{t,j}^{(1)} + b_{t,j}^{(2)}) + \sum_{j=0}^{s} \frac{12}{16\tau(4\tau-1)} \cdot D_{\tau,\rho}^2 \rho_{\max}^2 b_{t,j}^{(2)}$$

$$\leq \sum_{j=0}^{s} 3(b_{t,j}^{(1)} + b_{t,j}^{(2)}) + \sum_{j=0}^{s} \frac{1}{\tau^2} \cdot D_{\tau,\rho}^2 \cdot \rho_{\max}^2 b_{t,j}^{(2)}, \tag{E.27}$$

then by the definition of $b^{(1)}$ and $b^{(2)}$, we have

$$\sum_{j=0}^{s} (1,1) G^j B_{t,s-j}$$

$$\leq \sum_{j=0}^{s} 3\left( (4\rho_L \eta_l^2 + 8\tau \eta_l^2)(\alpha^2 \mathbb{E}\|\nabla f(\mathbf{x}_t)\|^2 + \sigma_g^2) + 4\rho_L \eta_l^2 \bar{\sigma}_L^2 + \eta_l^2 \sigma^2 \rho_{\max}^2 + \eta_l^2 \frac{\sigma^2}{n} \right)$$

$$+ \sum_{j=0}^{s} \frac{D_{\tau,\rho}^2}{\tau^2} \cdot \rho_{\max}^2 \left( 8\tau \eta_l^2 (\alpha^2 \mathbb{E}\|\nabla f(\mathbf{x}_t)\|^2 + \sigma_g^2) + \eta_l^2 \frac{\sigma^2}{n} \right)$$

$$= (s+1) \left[ \left( 3(4\rho_L \eta_l^2 + 8\tau \eta_l^2) + \frac{D_{\tau,\rho}^2}{\tau^2} \cdot \rho_{\max}^2 8\tau \eta_l^2 \right)(\alpha^2 \mathbb{E}\|\nabla f(\mathbf{x}_t)\|^2 + \sigma_g^2) + 12\rho_L \eta_l^2 \bar{\sigma}_L^2 \right.$$

$$\left. + 3\eta_l^2 \sigma^2 \rho_{\max}^2 + \left( 3\eta_l^2 + \frac{D_{\tau,\rho}^2}{\tau^2} \cdot \rho_{\max}^2 \eta_l^2 \right) \frac{\sigma^2}{n} \right]$$

$$\leq (s+1) \left[ \left( 12\rho_{\max}^2 D_{\tau,\rho} \eta_l^2 + 8\tau \eta_l^2 + 8\eta_l^2 D_{\tau,\rho} \cdot \rho_{\max}^2 \right)(\alpha^2 \mathbb{E}\|\nabla f(\mathbf{x}_t)\|^2 + \sigma_g^2) + 12\rho_{\max}^2 D_{\tau,\rho} \eta_l^2 \bar{\sigma}_L^2 \right.$$

$$\left. + 3\eta_l^2 \sigma^2 \rho_{\max}^2 + \left( 3\eta_l^2 + \frac{D_{\tau,\rho}^2}{\tau^2} \cdot \rho_{\max}^2 \eta_l^2 \right) \frac{\sigma^2}{n} \right]$$

$$\leq (s+1) C_1 \eta_l^2 (\tau + \rho_{\max}^2 D_{\tau,\rho})(\alpha^2 \mathbb{E}\|\nabla f(\mathbf{x}_t)\|^2 + \sigma_g^2) + (s+1) C_1 \rho_{\max}^2 D_{\tau,\rho} \eta_l^2 \bar{\sigma}_L^2$$

$$+ (s+1) C_1 \eta_l^2 \sigma^2 \rho_{\max}^2 + (s+1) C_1 \left( 1 + \frac{D_{\tau,\rho}^2}{\tau^2} \cdot \rho_{\max}^2 \right) \eta_l^2 \frac{\sigma^2}{n}, \tag{E.28}$$

where $C_1$ is some universal constant. The inequality holds by $\rho_L = \rho_{\max}^2 D_{\tau,\rho}$ and $D_{\tau,\rho} \leq \tau$.

**Case 2:** In this case we have $\rho_{\max} > 1 - \frac{1}{\tau}$, which means $D_{\tau,\rho} = \tau$. Let $\zeta_k = (4\tau - 1)$, thus we have

$$\max_{k \in [K]} \rho_k^2 (1 + \zeta_k^{-1}) = \rho_{\max}^2 (1 + (4\tau - 1)^{-1}), \quad \rho_L = 4\tau \rho_{\max}^2 D_{\tau,\rho}. \tag{E.29}$$

The upper bound for $\lambda_2$ has the form of

$$\lambda_2 \leq \max \left\{ \max_{k \in [K]} \rho_k^2 (1 + \zeta_k^{-1}) + \eta_l^2 \rho_L \cdot 8L^2, C_\tau \right\}$$

$$\leq \max \left\{ \rho_{\max}^2 (1 + (4\tau - 1)^{-1}) + \frac{2\rho_{\max}^2}{4\tau - 1}, 1 + \frac{3}{2(4\tau - 1)} \right\}$$

$$\leq 1 + \frac{3}{4\tau - 1}. \tag{E.30}$$

By the fact of $\min\left\{ \frac{1}{\lambda_2 - \lambda_1}, l \right\} \leq \tau = D_{\tau,\rho}$, we have

$$\sum_{j=0}^{s} (1,1) G^j B_{t,s-j} \leq \sum_{j=0}^{s} \lambda_2^j (b_{t,s-j}^{(1)} + b_{t,s-j}^{(2)}) + \sum_{j=0}^{s} \left( \lambda_2^j \min\left\{ \frac{1}{\lambda_2 - \lambda_1}, l \right\} \eta_l^2 \cdot 4\rho_L L^2 b_{t,s-j}^{(2)} \right)$$

$$\leq \sum_{j=0}^{s} 3(b_{t,j}^{(1)} + b_{t,j}^{(2)}) + \sum_{j=0}^{s} \eta_l^2 \cdot 16\rho_{\max} D_{\tau,\rho} L^2 b_l^{(2)} \cdot 3D_{\tau,\rho}$$

$$\leq \sum_{j=0}^{s} 3(b_{t,j}^{(1)} + b_{t,j}^{(2)}) + \sum_{j=0}^{s} 16\rho_{\max} D_{\tau,\rho} b_l^{(2)} \cdot \frac{3}{16} \frac{D_{\tau,\rho}}{\tau^2}$$

$$= \sum_{j=0}^{s} 3(b_{t,j}^{(1)} + b_{t,j}^{(2)}) + \sum_{j=0}^{s} 3\rho_{\max} b_l^{(2)} \cdot \frac{D_{\tau,\rho}^2}{\tau^2}, \tag{E.31}$$

where the above inequalities hold by the fact that $\rho_L = 4\tau \rho_{\max}^2 = 4D_{\tau,\rho} \rho_{\max}^2$ and the constraint on step size $\eta_l$. Thus we can get a similar upper bound as Eq. E.28 in Case 1. This concludes the proof. $\square$

**Lemma E.4.** With the similar condition in Lemma E.3, we have the corresponding bound for the intra-cluster consensus error $\|X_{t,s}^{k,\perp}\|_F^2$,

$$\frac{1}{N}\sum_{k=1}^{K}\mathbb{E}[\|X_{t,s}^{k,\perp}\|^2] \leq (s+1)C_1\eta_l^2 D_{\tau,\rho}\rho_{\max}^2(\alpha^2\mathbb{E}[\|\nabla f(\mathbf{x}_t)\|]^2 + \sigma_g^2) + (s+1)C_1\eta_l^2 D_{\tau,\rho}\rho_{\max}^2\bar{\sigma}_L^2$$

$$+ (s+1)C_1\eta_l^2\rho_{\max}^2\sigma^2 + (s+1)C_1\eta_l^2\frac{D_{\tau,\rho}^2}{\tau^2}\cdot\rho_{\max}^2\frac{\sigma^2}{n}. \tag{E.32}$$

*Proof.* With the same definition of the auxiliary vector $M_{t,s}$ and the matrix $G$ and $B_{t,s}$ in the proof of Lemma E.3, there is

$$M_{t,s} = \left(\frac{1}{N}\sum_{k=1}^{K}\mathbb{E}[\|X_{t,s}^{k,\perp}\|^2], \quad \frac{1}{K}\sum_{k=1}^{K}\mathbb{E}[\|\bar{\mathbf{x}}_{t,s}^k - \mathbf{x}_t\|^2]\right)^{\top},$$

$$M_{t,s} = G^{s+1}M_{t,0} + \sum_{j=0}^{s}G^j B_{t,s-j} = \sum_{j=0}^{s}G^j B_{t,s-j}, \tag{E.33}$$

hence we have

$$\frac{1}{N}\sum_{k=1}^{K}\mathbb{E}[\|X_{t,s}^{k,\perp}\|^2] = (1,0)\cdot M_{t,s} = (1,0)\cdot\sum_{j=0}^{s}G^j B_{t,s-j}$$

$$= \sum_{j=0}^{s}\left[\lambda_1^j b_{t,j}^{(1)} + \frac{\lambda_2^j - \lambda_1^j}{\lambda_2 - \lambda_1}4\eta_l^2\rho_L L^2 b_{t,j}^{(2)}\right]$$

$$\leq \sum_{j=0}^{s}\left[\lambda_2^j b_{t,j}^{(1)} + \frac{\lambda_2^j - \lambda_1^j}{\lambda_2 - \lambda_1}4\eta_l^2\rho_L L^2 b_{t,j}^{(2)}\right]$$

$$\leq \sum_{j=0}^{s}\left[\lambda_2^j b_{t,j}^{(1)} + \frac{\lambda_2^j - \lambda_1^j}{\lambda_2 - \lambda_1}4\eta_l^2\rho_L L^2 b_{t,j}^{(2)}\right], \tag{E.34}$$

with the similar proof techniques as in Lemma E.3, there is

$$\frac{1}{N}\sum_{k=1}^{K}\mathbb{E}[\|X_{t,s}^{k,\perp}\|^2] \leq \sum_{j=0}^{s}\left[3b_{t,j}^{(1)} + \frac{1}{\tau^2}D_{\tau,\rho}^2\rho_{\max}^2 b_{t,j}^{(2)}\right]$$

$$\leq \sum_{j=0}^{s}\left[12\rho_L\eta_l^2(\alpha^2\mathbb{E}[\|\nabla f(\mathbf{x}_t)\|]^2 + \sigma_g^2) + 12\rho_L\eta_l^2\bar{\sigma}_L^2 + 3\eta_l^2\sigma^2\rho_{\max}^2\right.$$

$$\left. + \frac{D_{\tau,\rho}^2}{\tau^2}\rho_{\max}^2\left(8\tau\eta_l^2(\alpha^2\mathbb{E}[\|\nabla f(\mathbf{x}_t)\|]^2 + \sigma_g^2) + \eta_l^2\frac{\sigma^2}{n}\right)\right]$$

$$\leq (s+1)C_1\eta_l^2 D_{\tau,\rho}\rho_{\max}^2(\alpha^2\mathbb{E}[\|\nabla f(\mathbf{x}_t)\|]^2 + \sigma_g^2) + (s+1)C_1\eta_l^2 D_{\tau,\rho}\rho_{\max}^2\bar{\sigma}_L^2$$

$$+ (s+1)C_1\eta_l^2\rho_{\max}^2\sigma^2 + (s+1)C_1\eta_l^2\frac{D_{\tau,\rho}^2}{\tau^2}\cdot\rho_{\max}^2\frac{\sigma^2}{n}. \tag{E.35}$$

$\square$

### E.4 Lemmas for model difference $\Delta_t$

### E.4.1 Full participation

**Lemma E.5.** The global model difference $\Delta_t = \sum_{k=1}^{K}\sum_{i\in\mathcal{V}_k}\Delta_t^i$ in full participation cases satisfy

$$\mathbb{E}[\|\Delta_t\|^2] \leq \frac{\eta_l^2\tau}{N}\sigma^2 + \eta_l^2\tau\sum_{s=0}^{\tau-1}\mathbb{E}\left[\left\|\frac{1}{N}\sum_{i=1}^{N}\nabla f_i(\mathbf{x}_{t,s}^i)\right\|^2\right]. \tag{E.36}$$

*Proof.* Under full participation case, we have $\Delta_t = -\eta_l \sum_{s=0}^{\tau-1} \frac{1}{K} \sum_{k \in [K]} \frac{1}{n} \sum_{i \in \mathcal{V}_k} \mathbf{g}_{t,s}^i = -\eta_l \sum_{s=0}^{\tau-1} \frac{1}{N} \sum_{i=1}^{N} \mathbf{g}_{t,s}^i$

$$
\begin{aligned}
\mathbb{E}[\|\Delta_t\|^2] &= \mathbb{E}\left[\left\|\eta_l \sum_{s=0}^{\tau-1} \frac{1}{N} \sum_{i=1}^{N} \mathbf{g}_{t,s}^i\right\|^2\right] \\
&= \eta_l^2 \mathbb{E}\left[\left\|\frac{1}{N} \sum_{s=0}^{\tau-1} \sum_{i=1}^{N} \mathbf{g}_{t,s}^i\right\|^2\right] \\
&= \eta_l^2 \mathbb{E}\left[\left\|\sum_{s=0}^{\tau-1} \left(\frac{1}{N} \sum_{i=1}^{N} \mathbf{g}_{t,s}^i - \frac{1}{N} \sum_{i=1}^{N} \nabla f_i(\mathbf{x}_{t,s}^i)\right) + \sum_{s=0}^{\tau-1} \frac{1}{N} \sum_{i=1}^{N} \nabla f_i(\mathbf{x}_{t,s}^i)\right\|^2\right] \\
&\leq \frac{\eta_l^2 \tau}{N} \sigma^2 + \eta_l^2 \tau \sum_{s=0}^{\tau-1} \mathbb{E}\left[\left\|\frac{1}{N} \sum_{i=1}^{N} \nabla f_i(\mathbf{x}_{t,s}^i)\right\|^2\right].
\end{aligned}
\tag{E.37}
$$

where the inequalities holds by the fact of $\mathbf{g}_{t,s}^i$ is the unbiased estimator of $\nabla f_i(\mathbf{x}_{t,s}^i)$ and by Assumption 5.3. This concludes the proof. $\qquad\square$

### E.4.2  Partial participation

There is a corresponding Lemma about model difference $\Delta_t$ for the partial participation settings.

**Lemma E.6.** The global model difference $\Delta_t = \sum_{k=1}^{K} \sum_{i \in cS_t} \Delta_t^i$ in partial participation settings satisfies

$$
\begin{aligned}
&\mathbb{E}[\|\Delta_t\|^2] \\
&\leq \frac{2\eta_l^2 \tau}{N} \sigma^2 + 2\eta_l^2 (\tau-1) \sum_{s=0}^{\tau-2} \mathbb{E}\left[\left\|\frac{1}{N} \sum_{i=1}^{N} \nabla f_i(\mathbf{x}_{t,s}^i)\right\|^2\right] + 4\eta_l^2 \mathbb{E}\left[\left\|\frac{1}{K} \sum_{k=1}^{K} \nabla \bar{f}_k(\bar{\mathbf{x}}_{t,\tau-1}^k)\right\|^2\right] \\
&\quad + 8\left(\frac{n-m}{m(n-1)} + \eta_l^2 L^2\right)\left(\frac{1}{N} \sum_{k=1}^{K} \mathbb{E}[\|X_{t,\tau-1}^{k,\perp}\|^2]\right) + \frac{2\eta_l^2 \sigma^2}{N}\left(\frac{n-m}{m} \cdot \rho_{\max}^2\right).
\end{aligned}
\tag{E.38}
$$

*Proof.* Recall the definition in B, there are $\bar{\mathbf{x}}_{t,s} = \frac{1}{N} \sum_{i=1}^{N} \mathbf{x}_{t,s}^i$ (without consideration of client sampling) and the intra-cluster average $\bar{\mathbf{x}}_{t,s+1}^k = \bar{\mathbf{x}}_{t,s}^k - \eta_l \bar{\mathbf{x}}_{t,s+1}^k$, where $\bar{\mathbf{x}}_{t,s+1}^k = \frac{1}{n} \sum_{i \in \mathcal{V}_k} \mathbf{g}_{t,s}^i$.

Consider the partial participation in the last step bfore the communication round, there is $\bar{\mathbf{x}}_{t+1} = \frac{1}{K} \sum_{k=1}^{K} \frac{1}{m} \sum_{i \in \mathcal{S}_t^k} \mathbf{x}_{t,\tau}^i = \frac{1}{Np} \sum_{k=1}^{K} \sum_{i \in \mathcal{S}_t^k} \mathbf{x}_{t,\tau}^i$ and by Algorithm 2, there is $\bar{\mathbf{x}}_{t+1} - \mathbf{x}_t = \Delta_t$.

For the model difference $\Delta_t$, we have

$$
\begin{aligned}
\mathbb{E}[\|\Delta_t\|^2] &= \mathbb{E}\left[\left\|\frac{1}{K} \sum_{k=1}^{K} \frac{1}{m} \sum_{i \in \mathcal{S}_t^k} \mathbf{x}_{t,\tau}^i - \mathbf{x}_t\right\|^2\right] \\
&= \mathbb{E}\left[\left\|\frac{1}{K} \sum_{k=1}^{K} \frac{1}{m} \sum_{i \in \mathcal{S}_t^k} \mathbf{x}_{t,\tau}^i \mp \bar{\mathbf{x}}_{t,\tau-1} \mp \cdots \mp \bar{\mathbf{x}}_{t,1} - \mathbf{x}_t\right\|^2\right] \\
&\leq 2\mathbb{E}\left[\left\|\frac{1}{Np} \sum_{k=1}^{K} \sum_{i \in \mathcal{S}_t^k} \mathbf{x}_{t,\tau}^i - \bar{\mathbf{x}}_{t,\tau-1}\right\|^2\right] + 2\mathbb{E}\left[\left\|\bar{\mathbf{x}}_{t,\tau-1} \mp \cdots \mp \bar{\mathbf{x}}_{t,1} - \mathbf{x}_t\right\|^2\right], \quad \text{(E.39)}
\end{aligned}
$$

where the inequality holds by Cauchy-Schwarz inequality. For the first term in Eq. E.39, we have

$$\mathbb{E}\left[\left\|\frac{1}{Np}\sum_{k=1}^{K}\sum_{i\in\mathcal{S}_t^k}\mathbf{x}_{t,\tau}^i - \bar{\mathbf{x}}_{t,\tau-1}\right\|^2\right]$$

$$= \mathbb{E}\left[\left\|\frac{1}{Np}\sum_{k=1}^{K}\sum_{i\in\mathcal{S}_t^k}\left(\sum_{j\in\mathcal{V}_k}(W_k)_{i,j}(\mathbf{x}_{t,\tau-1}^j - \eta_l\mathbf{g}_{t,\tau-1}^j)\right) - \bar{\mathbf{x}}_{t,\tau-1}\right\|^2\right]$$

$$= \mathbb{E}\left[\left\|\frac{1}{Np}\sum_{k=1}^{K}\sum_{i\in\mathcal{V}_k}\mathbb{I}(i\in\mathcal{S}_t^k)\left(\sum_{j\in\mathcal{V}_k}(W_k)_{i,j}(\mathbf{x}_{t,\tau-1}^j - \eta_l\mathbf{g}_{t,\tau-1}^j)\right) - \bar{\mathbf{x}}_{t,\tau-1}\right\|^2\right]$$

$$= \mathbb{E}\left[\left\|\frac{1}{Np}\sum_{k=1}^{K}\sum_{i\in\mathcal{V}_k}\mathbb{I}(i\in\mathcal{S}_t^k)\left(\sum_{j\in\mathcal{V}_k}(W_k)_{i,j}(\mathbf{x}_{t,\tau-1}^j - \eta_l\nabla f_i(\mathbf{x}_{t,\tau-1}^j))\right) - \bar{\mathbf{x}}_{t,\tau-1}\right\|^2\right]$$

$$+ \eta_l^2\mathbb{E}\left[\left\|\frac{1}{Np}\sum_{k=1}^{K}\sum_{i\in\mathcal{V}_k}\mathbb{I}(i\in\mathcal{S}_t^k)\left(\sum_{j\in\mathcal{V}_k}(W_k)_{i,j}(\nabla f_i(\mathbf{x}_{t,\tau-1}^j) - \mathbf{g}_{t,\tau-1}^j)\right)\right\|^2\right]. \qquad \text{(E.40)}$$

For the first term in Eq. E.40, we have

$$\mathbb{E}\left[\left\|\frac{1}{Np}\sum_{k=1}^{K}\sum_{i\in\mathcal{V}_k}\mathbb{I}(i\in\mathcal{S}_t^k)\left(\sum_{j\in\mathcal{V}_k}(W_k)_{i,j}(\mathbf{x}_{t,\tau-1}^j - \eta_l\nabla f_j(\mathbf{x}_{t,\tau-1}^j))\right) - \bar{\mathbf{x}}_{t,\tau-1}\right\|^2\right]$$

$$= \mathbb{E}\left[\left\|\frac{1}{Np}\sum_{k=1}^{K}\sum_{i\in\mathcal{V}_k}\mathbb{I}(i\in\mathcal{S}_t^k)\left(\sum_{j\in\mathcal{V}_k}(W_k)_{i,j}(\mathbf{x}_{t,\tau-1}^j - \eta_l\nabla f_j(\mathbf{x}_{t,\tau-1}^j) - \bar{\mathbf{x}}_{t,\tau-1}^k)\right)\right\|^2\right]$$

$$\leq 2\mathbb{E}\left[\left\|\frac{1}{Np}\sum_{k=1}^{K}\sum_{i\in\mathcal{V}_k}\mathbb{I}(i\in\mathcal{S}_t^k)\left(\sum_{j\in\mathcal{V}_k}(W_k)_{i,j}\left(\mathbf{x}_{t,\tau-1}^j - \bar{\mathbf{x}}_{t,\tau-1}^k - \eta_l\nabla f_j(\mathbf{x}_{t,\tau-1}^j) + \eta_l\nabla\bar{f}_k(\bar{\mathbf{x}}_{t,\tau-1}^k)\right)\right)\right\|^2\right]$$

$$+ 2\mathbb{E}\left[\left\|\frac{1}{Np}\sum_{k=1}^{K}\sum_{i\in\mathcal{V}_k}\mathbb{I}(i\in\mathcal{S}_t^k)\left(\sum_{j\in\mathcal{V}_k}(W_k)_{i,j}\eta_l\nabla\bar{f}_k(\bar{\mathbf{x}}_{t,\tau-1}^k)\right)\right\|^2\right]$$

$$\leq 2\mathbb{E}\left[\left\|\frac{1}{Np}\sum_{k=1}^{K}\sum_{i\in\mathcal{V}_k}\mathbb{I}(i\in\mathcal{S}_t^k)\left(\sum_{j\in\mathcal{V}_k}(W_k)_{i,j}\left(\mathbf{x}_{t,\tau-1}^j - \bar{\mathbf{x}}_{t,\tau-1}^k - \eta_l\nabla f_j(\mathbf{x}_{t,\tau-1}^j) + \eta_l\nabla\bar{f}_k(\bar{\mathbf{x}}_{t,\tau-1}^k)\right)\right)\right\|^2\right]$$

$$+ 2\eta_l^2\mathbb{E}\left[\left\|\frac{1}{K}\sum_{k=1}^{K}\nabla\bar{f}_k(\bar{\mathbf{x}}_{t,\tau-1}^k)\right\|^2\right], \qquad \text{(E.41)}$$

where the first equation holds because

$$\bar{\mathbf{x}}_{t,\tau-1} = \frac{1}{N}\sum_{k=1}^{K}\sum_{i\in\mathcal{V}_k}\mathbf{x}_{t,\tau-1}^i = \frac{1}{K}\sum_{k=1}^{K}\bar{\mathbf{x}}_{t,\tau-1}^k = \frac{1}{K}\sum_{k=1}^{K}\frac{1}{m}\sum_{i\in\mathcal{V}_k}\mathbb{I}(i\in\mathcal{S}_t^k)\bar{\mathbf{x}}_{t,\tau-1}^k$$

$$= \frac{1}{Np}\sum_{k=1}^{K}\sum_{i\in\mathcal{V}_k}\mathbb{I}(i\in\mathcal{S}_t^k)\bar{\mathbf{x}}_{t,\tau-1}^k = \frac{1}{Np}\sum_{k=1}^{K}\sum_{i\in\mathcal{V}_k}\mathbb{I}(i\in\mathcal{S}_t^k)\sum_{j\in\mathcal{V}_k}(W_k)_{i,j}\bar{\mathbf{x}}_{t,\tau-1}^k, \qquad \text{(E.42)}$$

and the second inequality holds by the similar relationship

$$\frac{1}{Np}\sum_{k=1}^{K}\sum_{i\in\mathcal{V}_k}\mathbb{I}(i\in\mathcal{S}_t^k)\left(\sum_{j\in\mathcal{V}_k}(W_k)_{i,j}\eta_l\nabla\bar{f}_k(\bar{\mathbf{x}}_{t,\tau-1}^k)\right)$$

$$= \frac{1}{Np}\sum_{k=1}^{K}np\eta_l\nabla\bar{f}_k(\bar{\mathbf{x}}_{t,\tau-1}^k) = \frac{1}{K}\sum_{k=1}^{K}\eta_l\nabla\bar{f}_k(\bar{\mathbf{x}}_{t,\tau-1}^k). \qquad \text{(E.43)}$$

Denotes $\mathbf{y}_t^{k,i} = \sum_{j \in \mathcal{V}_k}(W_k)_{i,j}\big(\mathbf{x}_{t,\tau-1}^j - \bar{\mathbf{x}}_{t,\tau-1}^k - \eta_l \nabla f_j(\mathbf{x}_{t,\tau-1}^j) + \eta_l \nabla \bar{f}_k(\bar{\mathbf{x}}_{t,\tau-1}^k)\big)$

$$
\mathbb{E}\bigg[\Big\|\sum_{i=1}^n\sum_{i\in\mathcal{V}_k}\mathbb{I}\{i\in\mathcal{S}_k^t\}\mathbf{y}_t^{k,i}\Big\|^2\bigg] = \mathbb{E}\bigg[\Big\|\sum_{k=1}^K\sum_{i\in\mathcal{V}_k}\mathbb{P}\{i\in\mathcal{S}_k^t\}\mathbf{y}_t^{k,i}\Big\|^2\bigg]
$$

$$
= \mathbb{E}\bigg[\mathbb{P}\{i\in\mathcal{S}_k^t\}\sum_{k=1}^K\sum_{i\in\mathcal{V}_k}\|\mathbf{y}_t^{k,i}\|^2 + \mathbb{P}\{i\neq j\in\mathcal{S}_k^t\}\sum_{k=1}^K\sum_{i\neq j\in\mathcal{V}_k}(\mathbf{y}_t^{k,i})'(\mathbf{y}_t^j)
$$

$$
+ \mathbb{P}\{i\in\mathcal{S}_k^t, j\in\mathcal{S}_l^t|k\neq l\in[K]\}\sum_{k\neq l}\sum_{i\in\mathcal{V}_k}\sum_{j\in\mathcal{V}_l}(\mathbf{y}_t^{i,k})'(\mathbf{y}_t^{j,l})\bigg]
$$

$$
= \mathbb{E}\bigg[\frac{m}{n}\sum_{k=1}^K\sum_{i\in\mathcal{V}_k}\|\mathbf{y}_t^{k,i}\|^2 + \frac{m(m-1)}{n(n-1)}\sum_{k=1}^K\sum_{i\neq j\in\mathcal{V}_k}(\mathbf{y}_t^{k,i})'(\mathbf{y}_t^{k,j})
$$

$$
+ \frac{m^2}{n^2}\sum_{k\neq l}\sum_{i\in\mathcal{V}_k}\sum_{j\in\mathcal{V}_l}(\mathbf{y}_t^{i,k})'(\mathbf{y}_t^{j,l})\bigg]
$$

$$
= \mathbb{E}\bigg[\frac{m(m-1)}{n(n-1)}\Big\|\sum_{k=1}^K\sum_{i\in\mathcal{V}_k}\mathbf{y}_t^{k,i}\Big\|^2 + \frac{m(n-m)}{n(n-1)}\sum_{k=1}^K\sum_{i\in\mathcal{V}_k}\|\mathbf{y}_t^{k,i}\|^2
$$

$$
+ \frac{m(n-m)}{n^2(n-1)}\sum_{k\neq l}\sum_{i\in\mathcal{V}_k}\sum_{j\in\mathcal{V}_l}(\mathbf{y}_t^{i,k})'(\mathbf{y}_t^{j,l})\bigg]
$$

$$
\leq \mathbb{E}\bigg[\frac{m(m-1)}{n(n-1)}\Big\|\sum_{k=1}^K\sum_{i\in\mathcal{V}_k}\mathbf{y}_t^{k,i}\Big\|^2 + \frac{m(n-m)}{n(n-1)}\sum_{k=1}^K\sum_{i\in\mathcal{V}_k}\|\mathbf{y}_t^{k,i}\|^2
$$

$$
+ \frac{m(n-m)}{n^2(n-1)}\sum_{k\neq l}\sum_{i\in\mathcal{V}_k}\sum_{j\in\mathcal{V}_l}\Big(\frac{1}{2}\|\mathbf{y}_t^{i,k}\|^2 + \frac{1}{2}\|\mathbf{y}_t^{j,l}\|^2\Big)\bigg], \tag{E.44}
$$

where the third equation holds by the probability of random sampling with replacement, i.e., $\mathbb{P}\{i\in\mathcal{S}_k^t\} = \frac{m}{n}, \mathbb{P}\{i\neq j\in\mathcal{S}_k^t\} = \frac{m(m-1)}{n(n-1)}, \mathbb{P}\{i\in\mathcal{S}_k^t, j\in\mathcal{S}_l^t|k\neq l\in[K]\} = \frac{m^2}{n^2}$. The forth equation holds by $\langle\mathbf{a},\mathbf{b}\rangle = \frac{1}{2}[\|\mathbf{a}\|^2 + \|\mathbf{b}\|^2 - \|\mathbf{a}-\mathbf{b}\|^2]$, $\frac{1}{2}\sum_{i\neq j}\|\mathbf{a}_i - \mathbf{a}_j\|^2 = \sum_{i=1}^n n\|\mathbf{a}_i\|^2 - \|\sum_{i=1}^n \mathbf{a}_i\|^2$, and $\|\sum_{k=1}^K\sum_{i\in\mathcal{V}_k}\mathbf{y}_t^{k,i}\|^2 = \sum_{k=1}^K\|\sum_{i\in\mathcal{V}_k}\mathbf{y}_t^{k,i}\|^2 + \sum_{k\neq l}\sum_{i\in\mathcal{V}_k}\sum_{j\in\mathcal{V}_l}\langle\mathbf{y}_t^{k,i}\mathbf{y}_t^{l,j}\rangle$. The last inequality holds by $\mathbf{a}'\mathbf{b} \leq \frac{1}{2}\|\mathbf{a}\|^2 + \frac{1}{2}\|\mathbf{b}\|^2$. Re-organize the last item,

$$
\frac{m(n-m)}{n^2(n-1)}\sum_{k\neq l}\sum_{i\in\mathcal{V}_k}\sum_{j\in\mathcal{V}_l}\Big(\frac{1}{2}\|\mathbf{y}_t^{i,k}\|^2 + \frac{1}{2}\|\mathbf{y}_t^{j,l}\|^2\Big) = \frac{m(n-m)}{n^2(n-1)}(K-1)n\sum_{k=1}^K\sum_{i\in\mathcal{V}_k}\|\mathbf{y}_t^{k,i}\|^2, \tag{E.45}
$$

then we have

$$
\mathbb{E}\bigg[\Big\|\sum_{k\in[K]}\sum_{i=1}^n\mathbb{I}\{i\in\mathcal{S}_k^t\}\mathbf{y}_t^{k,i}\Big\|^2\bigg]
$$

$$
\leq \mathbb{E}\bigg[\frac{m(m-1)}{n(n-1)}\Big\|\sum_{k=1}^K\sum_{i\in\mathcal{V}_k}\mathbf{y}_t^{k,i}\Big\|^2 + \frac{m(n-m)}{n(n-1)}\sum_{k=1}^K\sum_{i\in\mathcal{V}_k}\|\mathbf{y}_t^{k,i}\|^2
$$

$$
+ \frac{m(n-m)}{n^2(n-1)}(K-1)n\sum_{k=1}^K\sum_{i\in\mathcal{V}_k}\|\mathbf{y}_t^i\|^2\bigg]
$$

$$
= \mathbb{E}\bigg[\frac{m(m-1)}{n(n-1)}\Big\|\sum_{k=1}^K\sum_{i\in\mathcal{V}_k}\mathbf{y}_t^{k,i}\Big\|^2 + \frac{Km(n-m)}{n(n-1)}\sum_{k=1}^K\sum_{i\in\mathcal{V}_k}\|\mathbf{y}_t^{k,i}\|^2\bigg]. \tag{E.46}
$$

By the definition of $\mathbf{y}_t^{k,i}$, we have

$$
\left\| \sum_{k=1}^{K} \sum_{i \in \mathcal{V}_k} \mathbf{y}_t^{k,i} \right\|^2 = \left\| \sum_{k=1}^{K} \sum_{i \in \mathcal{V}_k} \sum_{j \in \mathcal{V}_k} (W_k)_{i,j} \big( \mathbf{x}_{t,\tau-1}^{j} - \bar{\mathbf{x}}_{t,\tau-1}^{k} - \eta_l \nabla f_j(\mathbf{x}_{t,\tau-1}^{j}) + \eta_l \nabla \bar{f}_k(\bar{\mathbf{x}}_{t,\tau-1}^{k}) \big) \right\|^2
$$

$$
= \eta_l^2 \left\| \sum_{k=1}^{K} \sum_{i \in \mathcal{V}_k} \sum_{j \in \mathcal{V}_k} (W_k)_{i,j} \big( \nabla \bar{f}_k(\bar{\mathbf{x}}_{t,\tau-1}^{k}) - \nabla f_j(\mathbf{x}_{t,\tau-1}^{j}) \big) \right\|^2
$$

$$
= \eta_l^2 \left\| \sum_{k=1}^{K} \sum_{i \in \mathcal{V}_k} \sum_{j \in \mathcal{V}_k} (W_k)_{i,j} \big( \nabla f_j(\bar{\mathbf{x}}_{t,\tau-1}^{k}) - \nabla f_j(\mathbf{x}_{t,\tau-1}^{j}) \big) \right\|^2
$$

$$
\leq \eta_l^2 N L^2 \sum_{k=1}^{K} \sum_{i \in \mathcal{V}_k} \sum_{j \in \mathcal{V}_k} (W_k)_{i,j} \| \bar{\mathbf{x}}_{t,\tau-1}^{k} - \mathbf{x}_{t,\tau-1}^{j} \|^2
$$

$$
\leq \eta_l^2 N L^2 \sum_{k=1}^{K} \sum_{i \in \mathcal{V}_k} \| \bar{\mathbf{x}}_{t,\tau-1}^{k} - \mathbf{x}_{t,\tau-1}^{i} \|^2
$$

$$
= \eta_l^2 N L^2 \sum_{k=1}^{K} \| X_{t,\tau-1}^{k,\perp} \|^2, \tag{E.47}
$$

where the second equation holds by $\sum_{i \in \mathcal{V}_k} \sum_{j \in \mathcal{V}_k} (W_k)_{i,j} \bar{\mathbf{x}}_{t,\tau-1}^{k} = \sum_{i \in \mathcal{V}_k} \sum_{j \in \mathcal{V}_k} (W_k)_{i,j} \bar{\mathbf{x}}_{t,\tau-1}^{k}$, the third equation holds by $\sum_{i \in \mathcal{V}_k} \sum_{j \in \mathcal{V}_k} (W_k)_{i,j} \nabla \bar{f}_k(\bar{\mathbf{x}}_{t,\tau-1}^{k}) = \sum_{i \in \mathcal{V}_k} \sum_{j \in \mathcal{V}_k} (W_k)_{i,j} \nabla f_j(\bar{\mathbf{x}}_{t,\tau-1}^{k})$, the first inequality holds by $L$-smoothness and Cauchy-Schwarz inequality, the second inequality holds by the assumption of doubly stochastic weighting matrix, and the last inequality is due to the previous definition of $X_{t,\tau-1}^{k,\perp}$. By the definition of $\mathbf{y}_t^{k,i}$, we also have

$$
\sum_{k=1}^{K} \sum_{i \in \mathcal{V}_k} \| \mathbf{y}_t^{k,i} \|^2 = \sum_{k=1}^{K} \sum_{i \in \mathcal{V}_k} \left\| \sum_{j \in \mathcal{V}_k} (W_k)_{i,j} \big( \mathbf{x}_{t,\tau-1}^{j} - \bar{\mathbf{x}}_{t,\tau-1}^{k} - \eta_l \nabla f_j(\mathbf{x}_{t,\tau-1}^{j}) + \eta_l \nabla \bar{f}_k(\bar{\mathbf{x}}_{t,\tau-1}^{k}) \big) \right\|^2
$$

$$
\leq \sum_{k=1}^{K} \sum_{i \in \mathcal{V}_k} \sum_{j \in \mathcal{V}_k} (W_k)_{i,j} \left\| \mathbf{x}_{t,\tau-1}^{j} - \bar{\mathbf{x}}_{t,\tau-1}^{k} - \eta_l \nabla f_j(\mathbf{x}_{t,\tau-1}^{j}) + \eta_l \nabla \bar{f}_k(\bar{\mathbf{x}}_{t,\tau-1}^{k}) \right\|^2
$$

$$
\leq \sum_{k=1}^{K} \sum_{j \in \mathcal{V}_k} \left( 2 \| \mathbf{x}_{t,\tau-1}^{j} - \bar{\mathbf{x}}_{t,\tau-1}^{k} \|^2 + 2 \eta_l^2 \| \nabla f_j(\mathbf{x}_{t,\tau-1}^{j}) - \nabla \bar{f}_k(\bar{\mathbf{x}}_{t,\tau-1}^{k}) \|^2 \right)
$$

$$
\leq \sum_{k=1}^{K} \sum_{i \in \mathcal{V}_k} 2 \| \mathbf{x}_{t,\tau-1}^{i} - \bar{\mathbf{x}}_{t,\tau-1}^{k} \|^2 + \sum_{k=1}^{K} \sum_{i \in \mathcal{V}_k} 2 \eta_l^2 L^2 \| \mathbf{x}_{t,\tau-1}^{i} - \bar{\mathbf{x}}_{t,\tau-1}^{k} \|^2
$$

$$
= \sum_{k=1}^{K} 2(1 + \eta_l^2 L^2) \| X_{t,\tau-1}^{k,\perp} \|^2, \tag{E.48}
$$

where the first inequality holds by the fact of $0 \leq (W_k))_{i,j} \leq 1$ and $\| w_{11} \mathbf{a}_1 + w_{12} \mathbf{a}_2 + \cdots + w_{1n} \mathbf{a}_n \|^2 = w_{11}^2 \| \mathbf{a}_1 \|^2 + w_{12}^2 \| \mathbf{a}_2 \|^2 + \cdots + w_{1n}^2 \| \mathbf{a}_n \|^2 \leq w_{11} \| \mathbf{a}_1 \|^2 + w_{12} \| \mathbf{a}_2 \|^2 + \cdots + w_{1n} \| \mathbf{a}_n \|^2$. The second inequality holds by Cauchy-Schwarz inequality, the third one holds by $L$-smoothness (Assumption 5.5) and the double-stochasticity of matrix $W_k$. Hence for (E.41), we have the following

result

$$\mathbb{E}\left[\left\|\frac{1}{Np}\sum_{k=1}^{K}\sum_{i\in\mathcal{V}_k}\mathbb{I}(i\in\mathcal{S}_t^k)\left(\sum_{j\in\mathcal{V}_k}(W_k)_{i,j}(\mathbf{x}_{t,\tau-1}^j-\eta_l\nabla f_j(\mathbf{x}_{t,\tau-1}^j))\right)-\bar{\mathbf{x}}_{t,\tau-1}\right\|^2\right]$$

$$\leq 2\eta_l^2\mathbb{E}\left[\left\|\frac{1}{K}\sum_{k=1}^{K}\nabla\bar{f}_k(\bar{\mathbf{x}}_{t,\tau-1}^k)\right\|^2\right]$$

$$+\frac{2}{(Np)^2}\left(\frac{m(m-1)}{n(n-1)}\eta_l^2NL^2+\frac{Km(n-m)}{n(n-1)}2(1+\eta_l^2L^2)\right)\sum_{k=1}^{K}\mathbb{E}[\|X_{t,\tau-1}^{k,\perp}\|^2]$$

$$\leq 2\eta_l^2\mathbb{E}\left[\left\|\frac{1}{K}\sum_{k=1}^{K}\nabla\bar{f}_k(\bar{\mathbf{x}}_{t,\tau-1}^k)\right\|^2\right]$$

$$+\left(\frac{m-1}{Km(n-1)}2\eta_l^2L^2+\frac{n-m}{Kmn(n-1)}4(1+\eta_l^2L^2)\right)\sum_{k=1}^{K}\mathbb{E}[\|X_{t,\tau-1}^{k,\perp}\|^2]$$

$$\leq 2\eta_l^2\mathbb{E}\left[\left\|\frac{1}{K}\sum_{k=1}^{K}\nabla\bar{f}_k(\bar{\mathbf{x}}_{t,\tau-1}^k)\right\|^2\right]+\frac{4}{N}\left(\frac{n-m}{m(n-1)}+\eta_l^2L^2\right)\sum_{k=1}^{K}\mathbb{E}[\|X_{t,\tau-1}^{k,\perp}\|^2],\qquad(\text{E.49})$$

where the last inequality holds with the constraint on the learning rate $\eta_l\leq\frac{1}{8\tau L}$.

For the second term in Eq. E.40, we perform the similar strategies related to the client sampling and weighted gossip mixing (corresponding to Eq. E.46)

$$\eta_l^2\mathbb{E}\left[\left\|\frac{1}{Np}\sum_{k=1}^{K}\sum_{i\in\mathcal{V}_k}\mathbb{I}(i\in\mathcal{S}_t^k)\underbrace{\left(\sum_{j\in\mathcal{V}_k}(W_k)_{i,j}(\nabla f_i(\mathbf{x}_{t,\tau-1}^j)-\mathbf{g}_{t,\tau-1}^j)\right)}_{\mathbf{e}_t^{k,i}}\right\|^2\right]$$

$$=\frac{\eta_l^2}{(Np)^2}\mathbb{E}\left[\frac{m(m-1)}{n(n-1)}\left\|\sum_{k=1}^{K}\sum_{i\in\mathcal{V}_k}\mathbf{e}_t^{k,i}\right\|^2+\frac{m(n-m)}{n(n-1)}\sum_{k=1}^{K}\sum_{i\in\mathcal{V}_k}\|\mathbf{e}_t^{k,i}\|^2\right.$$

$$\left.+\frac{2m(n-m)}{n^2(n-1)}\sum_{k\neq l}\sum_{i\in\mathcal{V}_k}\sum_{j\in\mathcal{V}_l}(\mathbf{e}_t^{i,k})'(\mathbf{e}_t^{j,l})\right]$$

$$=\frac{\eta_l^2}{(Np)^2}\mathbb{E}\left[\frac{m(m-1)}{n(n-1)}\left\|\sum_{k=1}^{K}\sum_{i\in\mathcal{V}_k}\mathbf{e}_t^{k,i}\right\|^2+\frac{m(n-m)}{n(n-1)}\sum_{k=1}^{K}\sum_{i\in\mathcal{V}_k}\|\mathbf{e}_t^{k,i}\|^2\right],\qquad(\text{E.50})$$

where the second equation holds by the stochastic gradient noise is zero mean and independent distributed. By the independency of $(\nabla f_i(\mathbf{x}_{t,\tau-1}^j)-\mathbf{g}_{t,\tau-1}^j)$ and the double-stochasticity of $W_k$, i.e., $W_k\mathbf{1}=\mathbf{1}$ and $\mathbf{1}^\top W_k=\mathbf{1}^\top$, we have

$$\mathbb{E}\left[\left\|\sum_{k=1}^{K}\sum_{i\in\mathcal{V}_k}\mathbf{e}_t^{k,i}\right\|^2\right]=\mathbb{E}\left[\left\|\sum_{k=1}^{K}\sum_{i\in\mathcal{V}_k}\left(\sum_{j\in\mathcal{V}_k}(W_k)_{i,j}(\nabla f_i(\mathbf{x}_{t,\tau-1}^j)-\mathbf{g}_{t,\tau-1}^j)\right)\right\|^2\right]$$

$$=\mathbb{E}\left[\left\|\sum_{k=1}^{K}\sum_{i\in\mathcal{V}_k}(\nabla f_i(\mathbf{x}_{t,\tau-1}^j)-\mathbf{g}_{t,\tau-1}^j)\right\|^2\right]$$

$$=N\sigma^2,\qquad(\text{E.51})$$

and

$$\sum_{k=1}^{K}\sum_{i\in\mathcal{V}_k}\mathbb{E}[\|\mathbf{e}_t^{k,i}\|^2] = \sum_{k=1}^{K}\sum_{i\in\mathcal{V}_k}\mathbb{E}\left[\left\|\left(\sum_{j\in\mathcal{V}_k}(W_k)_{i,j}(\nabla f_i(\mathbf{x}_{t,\tau-1}^j) - \mathbf{g}_{t,\tau-1}^j)\right)\right\|^2\right]$$

$$= \sum_{k=1}^{K}\sum_{i\in\mathcal{V}_k}\sum_{j\in\mathcal{V}_k}(W_k)_{i,j}^2\sigma^2$$

$$= \sum_{k=1}^{K}\|W_k\|^2\sigma^2. \tag{E.52}$$

For the Frobenius norm $\|W_k\|^2$, by the fact of the Frobenius norm of a matrix is equal to $L-2$ norm its singular values, denote the singular values of $W_k$ as $d_{k,1} \le d_{k,2} \le \cdots \le d_{k,n} = 1$, we have

$$\sum_{k=1}^{K}\|W_k\|^2\sigma^2 = \sum_{k=1}^{K}\sigma^2\sum_{i=1}^{n}(d_k^i)^2 \le \sum_{k=1}^{K}\sigma^2(1+(n-1)\rho_k^2) \le K\sigma^2(1+(n-1)\rho_{\max}^2). \tag{E.53}$$

Summarize the items, we have

$$\eta_l^2\mathbb{E}\left[\left\|\frac{1}{Np}\sum_{k=1}^{K}\sum_{i\in\mathcal{V}_k}\mathbb{I}(i\in\mathcal{S}_t^k)\underbrace{\left(\sum_{j\in\mathcal{V}_k}(W_k)_{i,j}(\nabla f_i(\mathbf{x}_{t,\tau-1}^j) - \mathbf{g}_{t,\tau-1}^j)\right)}_{\mathbf{e}_t^{k,i}}\right\|^2\right]$$

$$\le \frac{\eta_l^2}{(Np)^2}\frac{m(m-1)}{n(n-1)}N\sigma^2 + \frac{\eta_l^2}{(Np)^2}\frac{m(n-m)}{n(n-1)}K\sigma^2(1+(n-1)\rho_{\max}^2)$$

$$= \frac{\eta_l^2}{Np}\frac{m-1}{n-1}\sigma^2 + \frac{\eta_l^2}{Np}\frac{n-m}{n(n-1)}\sigma^2 + \frac{\eta_l^2}{Np}\frac{n-m}{n}\sigma^2\rho_{\max}^2$$

$$= \frac{\eta_l^2}{N}\sigma^2 + \frac{\eta_l^2}{N}\frac{n-m}{m}\sigma^2\rho_{\max}^2. \tag{E.54}$$

Therefore, we have the following result,

$$\mathbb{E}[\|\bar{\mathbf{x}}_{t+1} - \bar{\mathbf{x}}_{t,\tau-1}\|^2]$$

$$\le 2\eta_l^2\mathbb{E}\left[\left\|\frac{1}{K}\sum_{k=1}^{K}\nabla\bar{f}_k(\bar{\mathbf{x}}_{t,\tau-1}^k)\right\|^2\right] + 4\left(\frac{n-m}{m(n-1)} + \eta_l^2L^2\right)\left(\frac{1}{N}\sum_{k=1}^{K}\mathbb{E}[\|X_{t,\tau-1}^{k,\perp}\|^2]\right)$$

$$+ \frac{\eta_l^2\sigma^2}{N}\left(1 + \frac{n-m}{m}\cdot\rho_{\max}^2\right). \tag{E.55}$$

For the remaining part in Eq. E.39, we have

$$\mathbb{E}\left[\left\|\bar{\mathbf{x}}_{t,\tau-1}\pm\cdots\pm\bar{\mathbf{x}}_{t,1} - \mathbf{x}_t\right\|^2\right]$$

$$= \mathbb{E}\left[\left\|\eta_l\sum_{s=0}^{\tau-2}\bar{\mathbf{g}}_{t,s}\right\|^2\right]$$

$$= \mathbb{E}\left[\left\|\frac{\eta_l}{N}\sum_{s=0}^{\tau-2}\sum_{i=1}^{N}\mathbf{g}_{t,s}^i\right\|^2\right]$$

$$= \mathbb{E}\left[\left\|\frac{\eta_l}{N}\sum_{s=0}^{\tau-2}\sum_{i=1}^{N}(\mathbf{g}_{t,s}^i \pm \nabla f_i(\mathbf{x}_{t,s}^i))\right\|^2\right]$$

$$= \mathbb{E}\left[\left\|\frac{\eta_l}{N}\sum_{s=0}^{\tau-2}\sum_{i=1}^{N}(\mathbf{g}_{t,s}^i - \nabla f_i(\mathbf{x}_{t,s}^i))\right\|^2\right] + \mathbb{E}\left[\left\|\frac{\eta_l}{N}\sum_{s=0}^{\tau-2}\sum_{i=1}^{N}\nabla f_i(\mathbf{x}_{t,s}^i)\right\|^2\right]$$

$$\le \frac{\eta_l^2(\tau-1)}{N}\sigma^2 + \eta_l^2(\tau-1)\sum_{s=0}^{\tau-2}\mathbb{E}\left[\left\|\frac{1}{N}\sum_{i=1}^{N}\nabla f_i(\mathbf{x}_{t,s}^i)\right\|^2\right], \tag{E.56}$$

where the forth equation holds since the stochastic noise $(\mathbf{g}_{t,s}^i - \nabla f_i(\mathbf{x}_{t,s}^i))$ is zero mean Hence for partial participation, the model difference $\Delta$ satisfies

$$\mathbb{E}[\|\Delta_t\|^2]$$

$$\leq \frac{2\eta_l^2 \tau}{N}\sigma^2 + 2\eta_l^2(\tau - 1)\sum_{s=0}^{\tau-2}\mathbb{E}\left[\left\|\frac{1}{N}\sum_{i=1}^{N}\nabla f_i(\mathbf{x}_{t,s}^i)\right\|^2\right] + 4\eta_l^2\mathbb{E}\left[\left\|\frac{1}{K}\sum_{k=1}^{K}\nabla \bar{f}_k(\bar{\mathbf{x}}_{t,\tau-1}^k)\right\|^2\right]$$

$$+ 8\left(\frac{n-m}{m(n-1)} + \eta_l^2 L^2\right)\left(\frac{1}{N}\sum_{k=1}^{K}\mathbb{E}[\|X_{t,\tau-1}^{k,\perp}\|^2]\right) + \frac{2\eta_l^2\sigma^2}{N}\left(\frac{n-m}{m}\cdot\rho_{\max}^2\right). \tag{E.57}$$

$\square$

## E.5 Additional Supporting Lemmas

**Lemma E.7** (Lemma for momentum term in the update rule). The first order momentum terms $\mathbf{m}_t$ in Algorithm 1 and 2 hold the following relationship w.r.t. model difference $\Delta_t$:

$$\sum_{t=1}^{T}\mathbb{E}[\|\mathbf{m}_t\|^2] \leq \sum_{t=1}^{T}\mathbb{E}[\|\Delta_t\|^2]. \tag{E.58}$$

*Proof.* By the updating rule, we have

$$\mathbb{E}[\|\mathbf{m}_t\|^2] = \mathbb{E}\left[\left\|(1-\beta_1)\sum_{u=1}^{t}\beta_1^{t-u}\Delta_u\right\|^2\right]$$

$$\leq (1-\beta_1)^2\sum_{i=1}^{d}\mathbb{E}\left[\left(\sum_{u=1}^{t}\beta_1^{t-u}\Delta_{u,i}\right)^2\right]$$

$$\leq (1-\beta_1)^2\sum_{i=1}^{d}\mathbb{E}\left[\left(\sum_{u=1}^{t}\beta_1^{t-u}\right)\left(\sum_{u=1}^{t}\beta_1^{t-u}\Delta_{u,i}^2\right)\right]$$

$$\leq (1-\beta_1)\sum_{u=1}^{t}\beta_1^{t-u}\mathbb{E}[\|\Delta_u\|^2], \tag{E.59}$$

summing over $t = 1, ..., T$, we have

$$\sum_{t=1}^{T}\mathbb{E}[\|\mathbf{m}_t\|^2] = (1-\beta_1)\sum_{t=1}^{T}\sum_{u=1}^{t}\beta_1^{t-u}\mathbb{E}[\|\Delta_u\|^2]$$

$$= (1-\beta_1)\sum_{u=1}^{T}\sum_{t=u}^{T}\beta_1^{t-u}\mathbb{E}[\|\Delta_u\|^2]$$

$$\leq (1-\beta_1)\sum_{u=1}^{T}\frac{1}{1-\beta_1}\mathbb{E}[\|\Delta_u\|^2]$$

$$= \sum_{u=1}^{T}\mathbb{E}[\|\Delta_u\|^2]. \tag{E.60}$$

This concludes the proof. $\square$

**Lemma E.8.** Under Assumptions 5.2, for HA-Fed, we have $\|\nabla f(\mathbf{x})\| \leq G$, $\|\Delta_t\| \leq \eta_l\tau G$, $\|\mathbf{m}_t\| \leq \eta_l\tau G$ and $\|\mathbf{v}_t\| \leq \eta_l^2\tau^2 G^2$.

*Proof.* Since $f$ has $G$-bounded stochastic gradients, for any $\mathbf{x}$ and $\xi$, we have $\|\nabla f(\mathbf{x},\xi)\| \leq G$, we have

$$\|\nabla f(\mathbf{x})\| = \|\mathbb{E}_\xi \nabla f(\mathbf{x},\xi)\| \leq \mathbb{E}_\xi\|\nabla f(\mathbf{x},\xi)\| \leq G.$$

For HA-Fed, the model difference $\bar{\Delta}_t^k$ on cluster $k$ satisfies,

$$\bar{\Delta}_t^k = \bar{\mathbf{x}}_{t,\tau}^k - \mathbf{x}_t = -\eta_l \sum_{s=0}^{\tau-1} \bar{\mathbf{g}}_{t,s}^k,$$

therefore,

$$\|\bar{\Delta}_t^k\| = \left\| \eta_l \sum_{s=0}^{\tau-1} \bar{\mathbf{g}}_{t,s}^k \right\| = \left\| \eta_l \sum_{s=0}^{\tau-1} \frac{1}{K} \sum_{i \in \mathcal{V}_k} \mathbf{g}_{t,s}^i \right\| \le \eta_l \tau G,$$

for the global model difference $\Delta_t$,

$$\|\Delta_t\| = \left\| \frac{1}{K} \sum_{k \in [K]} \bar{\Delta}_t^k \right\| \le \eta_l \tau G.$$

Thus we can obtain the bound for momentum $\mathbf{m}_t$ and variance $\mathbf{v}_t$,

$$\|\mathbf{m}_t\| = \left\| (1-\beta_1) \sum_{\tau=1}^t \beta_1^{t-\tau} \Delta_t \right\| \le \eta_l \tau G, \quad \|\mathbf{v}_t\| = \left\| (1-\beta_2) \sum_{\tau=1}^t \beta_2^{t-\tau} \Delta_t^2 \right\| \le \eta_l^2 \tau^2 G^2.$$

This concludes the proof. $\qquad\square$

**Lemma E.9.** For the variance difference sequence $\widehat{\mathbf{V}}_{t-1}^{-1/2} - \widehat{\mathbf{V}}_t^{-1/2}$, we have

$$\sum_{t=1}^T \left\| \widehat{\mathbf{V}}_{t-1}^{-1/2} - \widehat{\mathbf{V}}_t^{-1/2} \right\|_1 \le \frac{d}{\sqrt{\epsilon}}, \quad \sum_{t=1}^T \left\| \widehat{\mathbf{V}}_{t-1}^{-1/2} - \widehat{\mathbf{V}}_t^{-1/2} \right\|^2 \le \frac{d}{\epsilon} \qquad (\text{E.61})$$

*Proof.* The proof of Lemma E.9 is exactly the same as the proof of Lemma C.2 in [41]. $\qquad\square$

**Lemma E.10.** For the element-wise difference, $W_t = \frac{1}{\sqrt{\mathbf{v}_t + \epsilon}} - \frac{1}{\sqrt{\beta_2 \mathbf{v}_{t-1} + \epsilon}}$, we have $\|W_t\| \le \frac{\sqrt{1-\beta_2}}{\epsilon} \|\Delta_t\|$.

*Proof.* The proof of Lemma E.9 is exactly the same as the proof of Lemma C.1 in [41]. $\qquad\square$

# F  Additional Experiments

## F.1  Simulation Study

We conduct the simulation study with synthetic data to verify the *dilemma of local steps*. The synthetic data is generated by Gaussian distribution and is heterogeneous among clients. We generate a simple set of data for 2 clients in the 2 dimensional space, in which there is assigned 10 data sample $(\mathbf{x}_i^{(k)}, y_i^{(k)})$ corresponding to client $k$. For each client $k$, $\mathbf{x}_i^{(k)}$ has mean $\boldsymbol{\mu}_k$ and covariance matrix $\boldsymbol{\Sigma}$, i.e., $x_i^{(k)} \sim \mathcal{N}(\boldsymbol{\mu}_k, \boldsymbol{\Sigma})$. The mean $\boldsymbol{\mu}_k$ varies from each client, thus data on each client are generating from different distributions. Specifically, we set $\boldsymbol{\mu}_1 = (1, 0)$ and $\boldsymbol{\mu}_2 = (0, 1)$, and labels of two clients are setup with $y_i^{(1)} = 0$ and $y_i^{(2)} = 1$. For both FedAMSGrad and HA-Fed, we simulate the full participation setting and use full batch gradients for better illustration. We perform two groups of comparison: (1) data sample $\mathbf{x}_i^{(k)}$ has covariance matrix $\mathbf{I}_2$, and we train the data on a simple multilayer-perceptron that has a hidden layer of 10 units; (2) $\mathbf{x}_i^{(k)}$ has a smaller covariance matrix $0.5 \cdot \mathbf{I}_2$, and we train the data on a simple multilayer-perceptron that has a hidden layer of 4 units; for both groups of comparison, we use ReLu activation function after the input layer.

Figure 3 shows the comparison of *local training loss* and *global training loss* for FedAMSGrad and our proposed HA-Fed algorithms. The local training loss is calculated during the training process, which reflects the average training loss on each clients with the local updated model. The global training loss is calculated in each global round after the local updates, it merges the training data among clients and calculates the training loss with all data samples. For example, suppose we have $N$ local client and each of which corresponds to a local loss function $f_i$, $i = 1, ..., N$. Denote $\mathbf{w}_i$

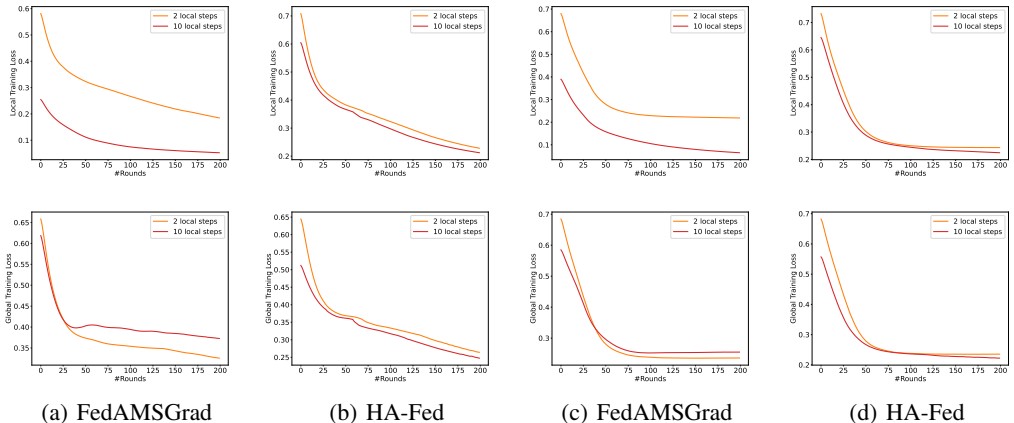



(a) FedAMSGrad     (b) HA-Fed     (c) FedAMSGrad     (d) HA-Fed



Figure 3: The comparison of local training loss and global training loss for FedAMSGrad and HA-Fed. (a)(b) is the comparison for group (1), (c)(d) is the comparison for group (2).

as the local model on client $i$, $\xi_i \sim \mathcal{D}_i$ is the data sample on client $i$ with distribution $\mathcal{D}_i$, then the *local training loss* is calculated by $f_{\text{local}} = \frac{1}{N} \sum_{i=1}^{N} f_i(\mathbf{w}_i; \xi_i)$. Moreover, after each round of local updates, we aggregate the temporary local models $\mathbf{w}' = \frac{1}{N} \sum_{i=1}^{N} \mathbf{w}_i$, together with the training data $\xi' = \bigcup_{i=1}^{N} \xi_i$ from all clients, the *global training loss* is calculated by $f_{\text{global}} = f(\mathbf{w}'; \xi')$, where $f = \frac{1}{N} \sum_{i=1}^{N} f_i$. The comparison between local and global training losses can reflect whether there exists over-fitting issues on local clients, if the local training loss decrease rapidly but the global training loss keeps a large value, it means the training with heterogeneous data causes over-fitting in clients.

From plot (a) and (c) in Figure 3, they show that FedAMSGrad with a larger number of local steps has higher global training loss compared to a smaller number of local steps. We observe that when local steps increase from $\tau = 2$ (orange) to $\tau = 10$ (red), FedAMSGrad gets worse in reducing global training loss though it improves the convergence to a locally optimal point. This shows that FedAMSGrad faces the *dilemma of local steps*, i.e., the convergence rate gets worse as the number of local steps increases. In contrast, plot (b) and (d) in Figure 3 show that our proposed HA-Fed effectively overcome this issue. For HA-Fed, the global training loss further decreases as the number of local steps increases from $\tau = 2$ to $\tau = 10$. This observation provides the empirical evidence that verifies our theory and shows that our proposed HA-fed can indeed overcome the *dilemma of local steps* in adaptive federated optimization methods.

## F.2 Non-i.i.d. Sampling and Hyperparameter Settings

**Non-i.i.d. data sampling** For the non-i.i.d. data sampling, we sort the training data by labels, and divide the data by labels. For CIFAR-10 [23] dataset, for each label, we divide the data into 20 shards size of 250. Hence we get 200 shards of size 250 in total, and each client is randomly assigned six shards. For CIFAR-100 [23] dataset, it includes 100 labels. For each label, we divide the data into 20 shards size of 25. Hence we get 2000 shards of size 25 in total, and each client is randomly assigned 60 shards. For Fashion MNIST [42] dataset, we similarly divide the data into 20 shards size of 300 for each label, and each client is randomly assigned six shards as well. The similar non-i.i.d. data sampling strategy is adopted in [29, 43, 14].

**Hyperparameter Settings** We conduct detailed hyperparameter searches to find the best hyperparameter for both FedAMSGrad and HA-Fed algorithms. We grid over the local learning rate $\eta_l \in \{0.001, 0.01, 0.1, 1.0\}$, and the global learning rate $\eta \in \{0.0001, 0.0005, 0.001, 0.01, 0.1\}$ for two methods. For the global AMSGrad optimizer, we set $\beta_1 = 0.9$, $\beta_1 = 0.99$, and we search the best $\epsilon$ from $\{10^{-10}, 10^{-8}, 10^{-6}, 10^{-4}, 10^{-2}\}$.

Specifically, for the ResNet-18 model on CIFAR-10 dataset, we set the local learning rate $\eta_l = 0.1$ and the global learning rate $\eta = 0.0005$ for FedAMSGrad and $\eta_l = 0.1$, $\eta = 0.001$ for HA-Fed, set

$\epsilon = 10^{-8}$ for both methods. For training ConvMixier-256-8 model on CIFAR-10 dataset, we set $\eta_l = 0.1, \eta = 0.001$ for FedAMSGrad, and $\eta_l = 0.1, \eta = 0.01$ for HA-Fed, set $\epsilon = 10^{-8}$ for both methods. For training ResNet-18 model on CIFAR-100 dataset, we set $\eta_l = 0.1$, $\eta = 0.001$ and $\epsilon = 10^{-8}$ for both methods. For training ConvMixer-256-8 model on CIFAR-100 dataset, we set $\eta_l = 1.0, \eta = 0.01$ and $\epsilon = 10^{-8}$ for both methods. For training ConvMixer-256-8 model on Fashion MNIST dataset, we set $\eta_l = 0.1, \eta = 0.0005$ for FedAMSGrad, $\eta_l = 0.1, \eta = 0.001$ for HA-Fed, and we set $\epsilon = 10^{-8}$ for both methods. For the CNN experiments on Fashion MNIST dataset, we set $\eta_l = 0.1, \eta = 0.01$ and $\epsilon = 10^{-8}$ for both two methods.

For training CIFAR-10 data on ConvMixer-256-8 model with Decentralized AMSGrad, we set the local learning rate $\eta_l = 0.001$ and $\epsilon = 10^{-8}$. We set $\eta_l = 0.1, \eta = 0.01$ and $\epsilon = 10^{-8}$ for FedAdam, $\eta_l = 0.01, \eta = 1.0$ for FedAvg, and $\eta_l = 0.1, \eta = 0.01$ and $\epsilon = 10^{-8}$ for FedYogi.

All the experiments are set up with 32 total clients in the network, which are equally divided into 4 clusters. The partial participation ratio is set to $p = 0.25$ except Decentralized AMSGrad, and the intra-cluster topology is ring topology by default. For both FedAMSGrad and HA-Fed, we conduct $\tau = 48$ of local training steps with a batch size of 50.

### F.3 Additional Experiments

In this section, we present additional empirical experiments for 1) our proposed HA-Fed and FedAMS-Grad algorithm in training CNN, and ConvMixer-256-8 model [37] on Fashion MNIST [42] dataset, where the CNN model [8] contains around 29 thousand trainable parameters; 2) HA-Fed and FedAMS-Grad in training ResNet-18[13] and ConvMixer-256-8 model [37] on CIFAR-10[23] dataset; 3) comparisons between our proposed HA-Fed and several federated learning baselines; 4) comparisons between HA-Fed and Decentralized AMSGrad; 5) extensions where some clients are inactive in some local iterations.

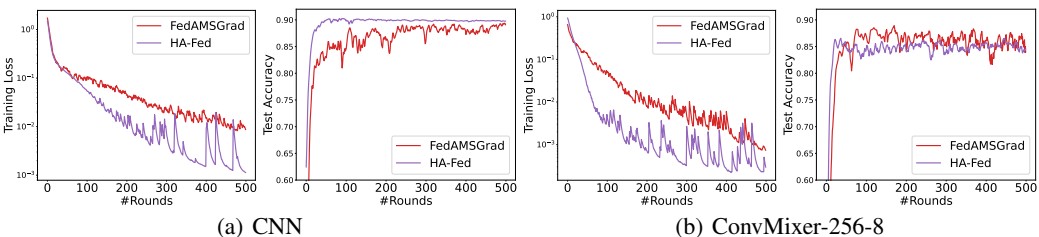

(a) CNN                                              (b) ConvMixer-256-8

Figure 4: The learning curves for HA-Fed and FedAMSGrad in training Fashion MNIST data on (a) CNN model and (b) ConvMixer-256-8 model using ring topology for intra-cluster communications.

Figure 4 shows the empirical convergence result for HA-Fed and FedAMSGrad on training Fashion-MNIST with CNN and ConvMixer-256-8 model. We compare the training loss and test accuracy against global rounds for two algorithms. Plot (a) in Figure 4 shows that HA-Fed (purple) achieves faster convergence than FedAMSGrad when training the CNN model to reduce training losses and obtain high test accuracy. For the ConvMixer-256-8 model (Figure (b)), HA-Fed again shows its faster convergence in reducing training loss, and HA-Fed maintains a similar test accuracy as FedAMSGrad.

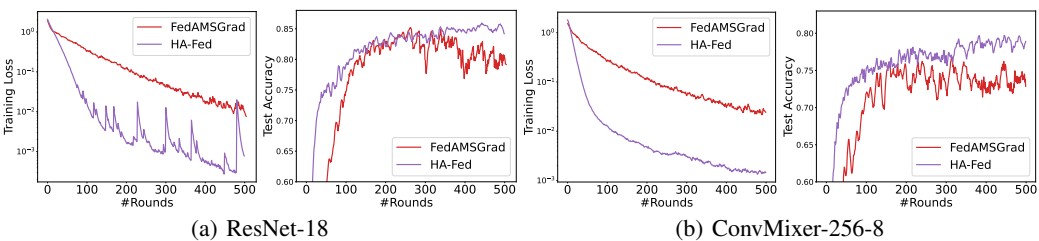

(a) ResNet-18                                        (b) ConvMixer-256-8

Figure 5: The learning curves for HA-Fed and FedAMSGrad in training CIFAR-10 data on (a) ResNet-18 model and (b) ConvMixer-256-8 model using ring topology for intra-cluster communications.

---

[8] The CNN model has two $5 \times 5$ convolution layers, where the first has 16 channels, the second has 32 channels, each followed with a $2 \times 2$ max pooling step.

Figure 5 shows the empirical convergence result for HA-Fed and FedAMSGrad on training CIFAR-10 with ResNet-18 and ConvMixer-256-8 model. We compare the training loss and test accuracy against global rounds for both models. For the ResNet-18 model, HA-Fed achieves faster convergence than FedAMSGrad in reducing training loss, and HA-Fed obtains an overall higher and stabler result in test accuracy. For the ConvMixer-256-8 model, HA-Fed again shows its faster convergence speed on training loss, in the meantime, HA-Fed still holds a higher test accuracy compared to FedAMSGrad under the same settings.

Moreover, Table 1 present the average test accuracy after three runs with different random seeds, and it shows that our proposed HA-Fed holds a higher accuracy than FedAMSGrad for all three datasets, with particular a stabler result for most of the experiments. These results empirically demonstrate the effectiveness and efficiency of our proposed HA-Fed method.

| CIFAR-10 | | | |
| ResNet-18 | Test Accuracy (%) | ConvMixer-256-8 | Test Accuracy (%) |
| --- | --- | --- | --- |
| FedAMSGrad | $79.72 \pm 3.31$ | FedAMSGrad | $73.96 \pm 3.02$ |
| HA-Fed | $\mathbf{84.38 \pm 0.33}$ | HA-Fed | $\mathbf{76.60 \pm 2.35}$ |
| CIFAR-100 | | | |
| ResNet-18 | Test Accuracy (%) | ConvMixer-256-8 | Test Accuracy (%) |
| FedAMSGrad | $56.34 \pm 0.79$ | FedAMSGrad | $61.97 \pm 0.35$ |
| HA-Fed | $\mathbf{57.12 \pm 0.47}$ | HA-Fed | $\mathbf{62.40 \pm 0.22}$ |
| Fashion MNIST | | | |
| CNN | Test Accuracy (%) | ConvMixer-256-8 | Test Accuracy (%) |
| FedAMSGrad | $88.79 \pm .16$ | FedAMSGrad | $83.54 \pm 3.36$ |
| HA-Fed | $\mathbf{89.25 \pm .22}$ | HA-Fed | $\mathbf{84.49 \pm 1.57}$ |

Table 1: The test accuracy (with mean and standard error) results with three random seeds for training CIFAR-10, CIFAR-100 and Fashion MNIST datasets.

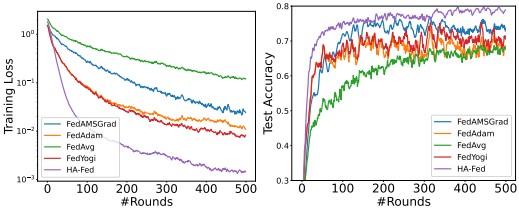

Figure 6: The learning curves for HA-Fed and several federated learning baselines in training CIFAR-10 data on ConvMixer-256-8 model.

Figure 6 shows the empirical convergence result of our HA-Fed and several federated learning baselines, including FedAvg, FedYogi, FedAdam and FedAMSgrad on training CIFAR-10 with ConvMixer-256-8 model. Our proposed HA-Fed shows its advantage in reducing training loss also with obtaining better test accuracy. Specifically, HA-Fed achieves nearly 10x smaller training loss after 500 global rounds, and HA-Fed achieves an accuracy of over 76% when training the ConvMixer-256-8 model, while FedAdam, FedYogi and FedAvg only achieve around 70%.

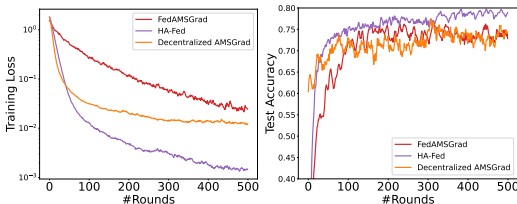

Figure 7: The learning curves for HA-Fed, FedAMSGrad and Decentralized AMSGrad in training CIFAR-10 data on ConvMixer-256-8 model.

Figure 7 shows the empirical convergence result of our HA-Fed with FedAMSgrad, one adaptive federated optimization method, and Decentralized AMSGrad, one adaptive decentralized optimization

method, on training CIFAR-10 with ConvMixer-256-8 model. Our proposed HA-Fed shows its advantage in reducing training loss also with obtaining better test accuracy. HA-Fed also achieves nearly $10\times$ smaller training loss, and achieves significantly better accuracy than Decentralized AMSGrad.

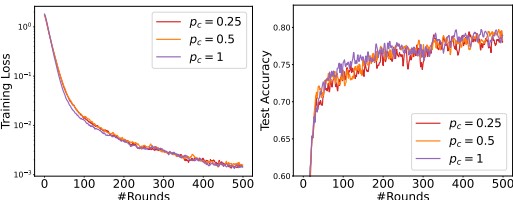

Figure 8: The learning curves for HA-Fed when cluster partial active cases in training CIFAR-10 data on ConvMixer-256-8 model.

Figure 8 shows the empirical convergence result of adaption algorithms based on HA-Fed in training CIFAR-10 with the ConvMixer-256-8 model. This adaption is to mimic the setting in that not all clients are active in each iteration. We assume that in each local iteration, each client is active (i.e., performs local model training) with probability $p_c$, and is inactive (i.e., stays idle and does not involve any computation) with probability $1 - p_c$. We simulate two cases with active rate $p_c = 1/2$ with gossip communicating in every 4 local steps and $p_c = 1/4$ with gossip communicating in every 8 local steps, and our original proposed HA-Fed can be seen as the case with $p_c = 1$ with gossip after each local step. Figure 8 shows that in such client partial active settings, HA-Fed can still achieve a similar convergence rate and test accuracy. The client who actives in each round of computing shows its advantage at an earlier stage, while all those three clients' active rates $p_c$ can achieve test accuracy over 78% after 500 global rounds.

