# OpenReview forum: "Accelerating Adaptive Federated Optimization with Local Gossip Communications"
_NeurIPS.cc/2022/Workshop/Federated_Learning — FL-NeurIPS 2022 Poster_

### Official Review · Reviewer_mpEc · 2022-10-03
**A good improvement on the state of the adaptive methods for FL**

I think this is a good paper on an important topic. It considers the problem of adaptive federated learning, where the adaptive stepsizes are used on the server side using differences of the parameter vectors. The authors prove convergence and an improvement compared to FedAMSGrad in the setting of partial participation. The authors also run experiments on Cifar10 to compare to FedAMSGrad when training convolutional neural networks.

I have a couple of suggestions for improving this work:
1. The experiments would benefit from averaging the results over multiple runs with confidence intervals.
2. The paper might be easier to read if the decentralized variant of the algorithm is presented after a simplified centralized one. This would mean two algorithms: one centralized with full participation (simplified) and one with partial participation and decentralized updates (all details).
3. The proofs are written with a lot of terms in most equations. Introducing some extra notation to make them shorter may improve their readability.

---

### Official Review · Reviewer_98Tx · 2022-10-19
**good paper but lacking some motivating example**

This is a good paper, which improves the existing upper bound for partial client participation in FL. However, the applications where having a cluster of clients would make sense, is missing.

---

### Official Review · Reviewer_UrDz · 2022-10-19
**The paper provides good ideas, but it has poor theory.**


This paper proposes a new adaptive method for federated optimization in non-convex regime. Authors suggest to use clusters of clients to utilize cheap client-to-client communications using decentarilzed techniques (gossip steps). While this idea looks promising, it is not new and it can be found in https://link.springer.com/article/10.1007/s11280-022-01046-x and https://arxiv.org/abs/2207.04338. The main problem with this approach is that authors compare the new method with FedAMSGrad method which does not have any client-to-client communication. In my opinion this comparison is not fair, since authors reformulate initial problem 3.1 and do not do the same for FedAMSGrad method. Also authors implicitly assume that client-to-client communications are avaliable and fast, while they do not allow client-to-client communications in case of FedAMSGrad method. I have to mention that compared methods should have the at least the same problem formulation and set of assumptions, which they use.
Unfortunately, the provided theory is poor. The set of assumptions is large and assumptions are strict. While assumptions C.1, C.5 and C.6 are quite general and standard in the optimization C.3 and C.4 are less general because of constant bound, but they are still standard in the literature. The most critical assumption is C.2, which says that all gradients are bounded by the same constant in R^d. This means that any function which grows faster than linear function cannot satisfy this assumption. Particularly, this means that for simple quadratic fucntion the proposed analysis cannot be applied. This is vital drawback, since quadratic function is the one of the simpliest functions to optimize.
The provided theorems are hard to understand. Theorems contain a lot of constants some of them are specified and some of them are not. For example the role of epsilon is not clear. Moreover, theorems rely of f_0 - f_*, which is also not clearly expalined. If f_* is the functional value in the minimum it is not clear which minimum is used (non-convex function can have many minimums). If the f_* is the lower bound of f, then it should be mentioned as an assumption. Not all functions can have such lower bounds.
In experiments authors used partial participation regime, when only 2 out of 8 clients participate in all 4 clusters. In this setting all clusters participate in client-server communication. While in comparison FedAMSGrad samples clients uniformly 8 out of  32 clients. I belive this is not fair comparison, since all clusters participate and sampling strategies are different. I recomend to reconsider the design of comparison between two methods.

Overall, I believe this paper needs major revision. I appreciate authors creativity and suggest to compare the new method with FedAMSGrad under the same conditions. Also it can be beneficial to avoid assumption C.2 in the analysis.

---

### Decision · Program_Chairs · 2022-10-20

Accept (Poster)